# The Perils of Optimizing Learned Reward Functions: Low Training Error Does Not Guarantee Low Regret

**Lukas Fluri** [* 1]   **Leon Lang** [* 2]   **Alessandro Abate** [3]   **Patrick Forré** [2]   **David Krueger** [4]   **Joar Skalse** [3]

## Abstract

In reinforcement learning, specifying reward functions that capture the intended task can be very challenging. Reward learning aims to address this issue by *learning* the reward function. However, a learned reward model may have a low error on the data distribution, and yet subsequently produce a policy with large regret. We say that such a reward model has an *error-regret mismatch*. The main source of an error-regret mismatch is the distributional shift that commonly occurs during policy optimization. In this paper, we mathematically show that a sufficiently low expected test error of the reward model guarantees low worst-case regret, but that for any *fixed* expected test error, there exist realistic data distributions that allow for error-regret mismatch to occur. We then show that similar problems persist even when using policy regularization techniques, commonly employed in methods such as RLHF. We hope our results stimulate the theoretical and empirical study of improved methods to learn reward models, and better ways to measure their quality reliably.

## 1. Introduction

To solve a sequential decision problem with reinforcement learning (RL), we must first formalize that decision problem using a *reward function* (Sutton & Barto, 2018). However, for complex tasks, reward functions are often hard to specify correctly (Krakovna, 2020). To solve this problem, it is increasingly popular to *learn* reward functions with *reward learning algorithms*, instead of specifying the reward functions manually. There are many different reward learning

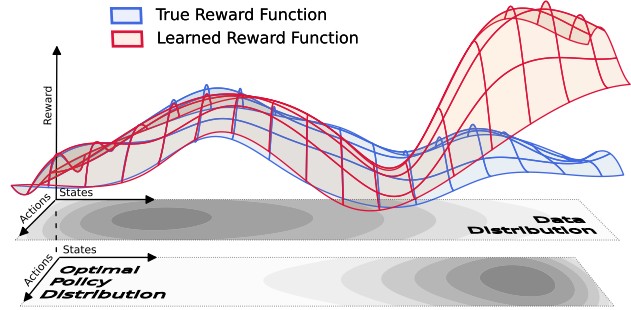

*Figure 1.* Reward models (red function) are commonly trained by supervised learning to approximate some latent, true reward (blue function). Given enough data, one can hope that the reward model is close to the true reward function on average over the training data distribution (upper gray layer) — the expected *error* is low. However, low expected error only guarantees a good approximation to the true reward function in areas with high coverage by the data distribution! Optimizing an RL policy to maximize the learned reward model then induces a distribution shift which can lead the policy to exploit uncertainties of the learned reward model in low-probability areas of the transition space (lower gray layer). In the worst case, this can lead to high *regret*. We refer to this phenomenon as *error-regret mismatch*.

algorithms (e.g. Ng & Russell (2000); Tung et al. (2018); Brown & Niekum (2019); Palan et al. (2019)), with one of the most popular being *reinforcement learning from human feedback* (RLHF) (Christiano et al., 2017; Ibarz et al., 2018).

For any learning algorithm, it is a crucial question whether or not that learning algorithm is guaranteed to converge to a "good" solution. For example, in the case of supervised learning for classification, it can be shown that a learning algorithm that produces a model with a low *empirical error* (i.e., training error) is likely to have a low *expected error* (i.e., test error), given a sufficient amount of training data and assuming that both the training data and the test data is drawn i.i.d. from a single stationary distribution (Kearns & Vazirani, 1994). In the case of supervised learning and standard assumptions, we can therefore be confident that a learning algorithm will converge to a good model, provided that it is given a sufficient amount of training data.

Since reward models are also typically learned by supervised

---

[*]Equal contribution   [1]ETH Zurich [2]University of Amsterdam [3]Oxford University [4]University of Cambridge. Correspondence to: Lukas Fluri <lukas.fluri@inf.ethz.ch>, Leon Lang <l.lang@uva.nl>.

*Proceedings of the $42^{nd}$ International Conference on Machine Learning*, Vancouver, Canada. PMLR 267, 2025. Copyright 2025 by the author(s).

learning, we might assume that classical learning-theoretic guarantees carry over. However, these guarantees only ensure that the reward model is approximately correct *relative to the training distribution*. But after reward learning, we optimize a policy to maximize the learned reward, which effectively leads to a *distributional shift*. This raises the worry that the trained policy can exploit regions of the state space with abnormally high learned rewards if those regions have a low data coverage during training. In this case, we can have reward models that have both a low error on the training distribution and an optimal policy with large regret, a phenomenon we call *error-regret mismatch*. We visualize this concern in Figure 1.

To illustrate this concern, imagine training a chatbot to be helpful, honest, and harmless (Askell et al., 2021). We know that the chatbot will face various unsafe queries during deployment (e.g. "how to build a bomb") and so on such queries we train a reward model to penalize helpful answers and highly reward refusals.

Unfortunately, all unsafe prompts can be answered in various distinct "styles" (e.g., different languages (Verma & Bharadwaj, 2025)). Consequently, at least one specific harmful answer will likely be very rare in the reward model's training data. The learned reward model can then erroneously assign a high reward to this rare, harmful answer without a significant increase in its training error (as this answer is infrequent in training). During policy optimization, the policy may exploit this flaw, choosing the harmful answer the reward model mistakenly prefers. This can result in a harmful chatbot with high true regret, despite the reward model having low error on the training data distribution, an example of error-regret mismatch. We illustrate this concern in detail in Appendix B.4.

To single out the issue of error-regret mismatch in our theoretical analysis, we take the goals of classical learning theory as a given and show that *they are not enough to ensure low regret*. More precisely, in probably approximately correct (PAC) learning (Kearns & Vazirani, 1994) the goal is to derive a sample size that guarantees a certain likelihood ("P") of an approximately correct ("AC") model on new data points sampled from the training distribution. In our results, we assume that we *already have* an approximately correct reward model on a data distribution, and then investigate what we can or can not conclude about the regret of policies trained to maximize the modeled reward.

Our theoretical analysis shows that guarantees in policy regret are very sensitive to the data distribution used to train the reward model, leading to our notions of *safe* and *unsafe data distributions*. Moreover, we find evidence that some MDPs are in a certain sense "too large" to allow for safe data distributions. We establish for general MDPs:

1. As the error of a learned reward model on a data distribution goes to zero, the worst-case regret of optimizing a policy according to that reward model also goes to zero (Theorems 3.1 and 3.2).

2. However, for any $\epsilon > 0$, whenever a data distribution has sufficiently low coverage of some bad policy, it is *unsafe*; in other words, there exists a reward model that achieves an expected error of $\epsilon$ but has a high-regret optimal policy (Theorem 3.3), a case of error-regret mismatch.

3. As a consequence, when an MDP has a large number of independent bad policies, *every* data distribution is unsafe (Theorem 3.4).

4. More precisely, we derive a set of linear constraints that precisely characterize the safe data distributions for a given MDP (Theorem 3.5).

We then investigate the case of *regularized* policy optimization (including KL-regularized policy optimization, which is commonly used in methods such as RLHF). We derive regularized versions of Theorems 3.1 and 3.3 in Theorem 4.1 and Theorem 4.2. This shows that regularization alone is no principled solution to error-regret mismatch.

We then develop several generalizations of our results for different types of data sources for reward model training, such as preferences over trajectories and trajectory scoring (Section 5). Lastly, motivated by the recent success of large language models (OpenAI, 2022; Gemini Team, 2023; Anthropic, 2023), we provide an analysis for the special case of RLHF in the contextual bandit case where we prove a stronger version (Theorem 6.1) of the failure mode already discussed in Theorem 4.2 for general MDPs.

## 1.1. Related work

*Note: We provide a more extensive related work section in Appendix A*

In offline reinforcement learning, we aim to learn low-regret policies for an MDP $\langle \mathcal{S}, \mathcal{A}, \tau, \mu_0, R, \gamma \rangle$ where the only information about the reward function $R$ (and sometimes transition distribution $\tau$ (Wang et al., 2022b; Uehara & Sun, 2021)) stems from a dataset $\{(s, a, r)_i\}_{i=1}^n$ sampled from some data distribution $D \in \Delta(\mathcal{S} \times \mathcal{A})$. A key research question is understanding which conditions on $D$ allow learning a *near-optimal policy* (i.e., a policy with regret smaller than some $L \in [0, 1]$) with an *efficient sample complexity*, where sample-efficient usually means polynomial in $L^{-1}$ and some other parameters of the MDP and $D$. Existing theoretical work primarily falls into two categories, covering both *MDPs* (Foster et al., 2021; Wang et al., 2022b; 2020; Amortila et al., 2020; Uehara & Sun, 2021; Uehara et al.,

2021) and *contextual bandits* (Nika et al., 2024; Cen et al., 2024):

**Lower bound results** prove that various data-coverage conditions are insufficient for sample-efficient offline RL by establishing a lower bound on the number of samples required to achieve low policy regret. In particular, research in this area (Foster et al., 2021; Wang et al., 2022b; 2020; Amortila et al., 2020; Nika et al., 2024) identifies adversarial MDPs that satisfy specific data-coverage conditions and yet for which achieving low regret is either computationally intractable due to excessive sample requirements (Foster et al., 2021; Wang et al., 2022b; 2020; Nika et al., 2024) or fundamentally impossible regardless of sample size (Amortila et al., 2020).

**Upper bound results**, on the other hand, establish positive guarantees under specific structural assumptions. Research in this category (Wang et al., 2022b; 2020; Uehara & Sun, 2021; Nika et al., 2024; Cen et al., 2024; Song et al., 2024) develops algorithms with provable upper-bounds on the number of required samples to achieve low policy regret. This is usually done by making structural assumptions about the MDP, reward learning process, or policy optimization approach.

In reward learning, compared to offline RL, one first learns a *reward model* from a dataset, which is typically sampled via various strategies from a data distribution $D \in \Delta(\mathcal{S} \times \mathcal{A})$ (cf. Section 2.1). Intuitively, as in offline RL, the quality of a reward model is influenced by two key factors: the dataset size $n$ and the dataset quality, specifically how well the data distribution $D$ *covers* the data space $\mathcal{S} \times \mathcal{A}$. Indeed, prior work confirms this intuition, with most works (see Nika et al. (2024) for a recent example) showing that the regret is dependent on *a)* the inverse dataset size $\frac{1}{n}$, *b)* some measure of the coverage of $D$, and *c)* some structural assumptions of the specific approach. Such structural assumptions may include: realizability of function classes (Wang et al., 2022b; Uehara & Sun, 2021; Foster et al., 2021; Nika et al., 2024), linear function approximation (Nika et al., 2024; Cen et al., 2024; Wang et al., 2022b), and various constraints on reward- or policy functions (Wang et al., 2020; Uehara & Sun, 2021; Nika et al., 2024).

Our paper differs from these works in a key aspect: We explicitly analyze how the reward modeling error is related to the final policy regret, rather than focusing on the number of samples. This also allows us to study *adversarial guarantees* of low policy regret (given low reward modeling error), whereas prior work considers probabilistic guarantees when sampling from the data distribution. The most relevant work studying a similar setup to ours is Song et al. (2024). Their setup in section 3, combined with their Assumption 4.3, perfectly recovers our safe distribution definition (see Theorem 2.1) when applied to the special case of RLHF and

when using the mean squared error metric. Their Theorem 4.2 demonstrates that $\mathrm{Regret} \in \mathcal{O}\big(\mathrm{Cov} \cdot \sqrt{\epsilon}\big)$, where $\mathrm{Cov}$ is some measure of coverage and $\epsilon$ the error in the reward function, and where the square root emerges from using the mean squared error during the reward learning step (see Appendix B.3 for how this relates to our results).

While Song et al. (2024) focus on RLHF with mean-squared error metric, we provide similar results for general classes of regularized and unregularized policy optimization (for both MDPs and contextual bandits), and show how these regret guarantees automatically generalize to a wide range of different error metrics for different reward learning methods. For our initial guarantees (Theorems 3.1, 3.2 and 4.1) we use the coverage condition $\min_{(s,a)} D(s,a) > 0$ and phrase the required reward learning error $\epsilon$ in terms of this coverage. Since we assume that all states of our MDPs are reachable, this is equivalent to a full coverage condition (see Table 1 of Uehara & Sun (2021) for an overview of different coverage conditions). We then show that for fixed $\epsilon$, a too small coverage leads to possibly high regret (Theorems 3.3, 4.2 and 6.1). Finally, we fully generalize our results from Theorems 3.1 to 3.4 into a single theorem (Theorem 3.5) which allows us to determine for *arbitrary* data distributions whether they give rise to worst-case safety for fixed error $\epsilon$ and required regret $L$. To the best of our knowledge, we are the first work to achieve such fine-grained safety results.

Several approaches have been proposed to address the issue of out-of-distribution robustness in reward learning, such as ensembles of conservative reward models (Coste et al., 2023), averaging weights of multiple reward models (Ramé et al., 2024), iteratively updating training labels (Zhu et al., 2024), on-policy reward learning (Lang et al., 2024a), and distributionally robust planning (Zhan et al., 2023). Recently, Kwa et al. (2024) show that RLHF and Conditioning can be provably safe under fairly strong structural assumptions—such as deterministic transitions, light-tailed reward errors, and independence between true and proxy rewards. Furthermore, Laidlaw et al. (2024) consider a setting where the learned and true reward functions are positively correlated under a reference policy. They prove that maximizing the proxy reward with a chi-squared divergence penalty yields regret no worse than that of the reference policy. In experiments, they approximate this regularized objective and report favorable results.

## 2. Preliminaries

A *Markov Decision Process* (MDP) is a tuple $\langle \mathcal{S}, \mathcal{A}, \tau, \mu_0, R, \gamma \rangle$ where $\mathcal{S}$ is a set of *states*, $\mathcal{A}$ is a set of *actions*, $\tau : \mathcal{S} \times \mathcal{A} \to \Delta(\mathcal{S})$ is a *transition function*, $\mu_0 \in \Delta(S)$ is an *initial state distribution*, $R : \mathcal{S} \times \mathcal{A} \to \mathbb{R}$ is a *reward function*, and $\gamma \in (0, 1)$ is a *discount rate*. We define the *range* of a reward function $R$ as $\mathrm{range}\, R :=$ $\max_{(s,a) \in \mathcal{S} \times \mathcal{A}} R(s,a) - \min_{(s,a) \in \mathcal{S} \times \mathcal{A}} R(s,a)$.

A *policy* is a function $\pi : \mathcal{S} \to \Delta(\mathcal{A})$. We denote the set of all policies by $\Pi$. A *trajectory* $\xi = \langle s_0, a_0, s_1, a_1, ... \rangle$ is a possible path in an MDP. The *return function G* gives the cumulative discounted reward of a trajectory, $G(\xi) = \sum_{t=0}^{\infty} \gamma^t R(s_t, a_t)$, and the *evaluation function J* gives the expected trajectory return given a policy, $J(\pi) = \mathbb{E}_{\xi \sim \pi} [G(\xi)]$. A policy maximizing $J$ is an *optimal policy*. We define the *regret* of a policy $\pi$ with respect to reward function $R$ as

$$\mathrm{Reg}^R(\pi) := \frac{\max_{\pi' \in \Pi} J_R(\pi') - J_R(\pi)}{\max_{\pi' \in \Pi} J_R(\pi') - \min_{\pi' \in \Pi} J_R(\pi')} \in [0, 1].$$

Here, $J_R$ is the policy evaluation function for $R$. We choose the regret $\mathrm{Reg}^R$ as our main performance metric of policies in this paper, which is justified by the fact that it is a normalized version of the policy evaluation $J_R$.

In this paper, we assume that $\mathcal{S}$ and $\mathcal{A}$ are finite, and that all states are reachable under $\tau$ and $\mu_0$. We also assume that $\max J_R - \min J_R \neq 0$ (since the reward function would otherwise be trivial). Note that this implies that range $R > 0$, and that $\mathrm{Reg}^R(\pi)$ is well-defined.

The *state-action occupancy measure* is a function $\eta : \Pi \to \mathbb{R}^{|\mathcal{S} \times \mathcal{A}|}$ mapping each policy $\pi \in \Pi$ to the corresponding "state-action occupancy measure", describing the discounted frequency that each state-action tuple is visited by a policy. Formally, $\eta(\pi)(s,a) = \eta^{\pi}(s,a) = \sum_{t=0}^{\infty} \gamma^t \cdot P(s_t = s, a_t = a \mid \xi \sim \pi)$. Note that by writing the reward function $R$ as a vector $\vec{R} \in \mathbb{R}^{|\mathcal{S} \times \mathcal{A}|}$, we can split $J$ into a function that is linear in $R$: $J(\pi) = \eta^{\pi} \cdot \vec{R}$. By normalizing a state-action occupancy measure $\eta^{\pi}$ we obtain a *policy-induced distribution* $D^{\pi} := (1 - \gamma) \cdot \eta^{\pi}$.

### 2.1. Problem formalization of RL with reward learning

In RL with reward learning, we assume that we have an MDP $\langle \mathcal{S}, \mathcal{A}, \tau, \mu_0, R, \gamma \rangle$ where the reward function $R$ is unknown. We may also assume that $\tau$ and $\mu_0$ are unknown, as long as we can sample from them (though $\mathcal{S}$, $\mathcal{A}$, and $\gamma$ must generally be known, at least implicitly). We then first learn a reward model $\hat{R}$ that approximates the true reward $R$ and then optimize a policy $\hat{\pi}$ to maximize $\hat{R}$. The aim of this two-step procedure is for $\hat{\pi}$ to achieve low regret under the true reward function $R$. We now formalize these aspects in detail for our theoretical analysis, with a visualization provided in Figure 2:

**Reward learning** We first learn a reward model $\hat{R}$ from data. There are many possible data sources for reward learning, like demonstrations (Ng & Russell, 2000), preferences over trajectories (Christiano et al., 2017), or even the initial environment state (Shah et al., 2019); a taxonomy can be found in (Jeon et al., 2020). Since we are concerned with problems that remain even when the reward model is already

*approximately correct*, we abstract away the data sources and training procedures and assume that we learn a reward model $\hat{R}$ which satisfies

$$\mathbb{E}_{(s,a) \sim D} \left[ \frac{|\hat{R}(s,a) - R(s,a)|}{\mathrm{range}\, R} \right] \leq \epsilon \qquad (1)$$

for some $\epsilon > 0$ and stationary distribution $D$ over transitions $\mathcal{S} \times \mathcal{A}$. Note that this is the true expectation under $D$, rather than an estimate of this expectation based on some finite sample. We divide by range $R$, since the absolute error $\epsilon$ is only meaningful relative to the overall scale of the reward $R$.

To be clear, most reward learning algorithms *cannot guarantee* a bound as in Equation (1) since most realistic data sources do not determine the true reward function, even for infinite data (Skalse et al., 2023). Instead, we choose Equation (1) because it serves as an *upper bound* to many common reward learning training objectives (see Appendix C.5). Thus, when we show in later sections that high regret is possible even when this inequality holds, then this problem can be expected to generalize to other data sources. We make this generalization precise for some data sources in Section 5. In particular, we will show that Equation (1) implies a low cross-entropy error between the choice distributions of the true reward function and the reward model, as is commonly used for RLHF, e.g. in the context of language models (Ziegler et al., 2019).

**Policy optimization** Given $\hat{R}$, we then learn a policy $\hat{\pi}$ by solving the MDP $\langle \mathcal{S}, \mathcal{A}, \tau, \mu_0, \hat{R}, \gamma \rangle$. In the most straightforward case, we do this by simply finding a policy that is optimal according to $\hat{R}$. However, it is also common to perform *regularized optimization*. In that case, we make use of an additional regularization function $\omega : \Pi \to \mathbb{R}$, with $\omega(\pi) \geq 0$ for all $\pi \in \Pi$. Given $\hat{R}$, a regularization function $\omega$, and a regularization weight $\lambda \in [0, \infty)$, we say that $\hat{\pi}$ is $(\lambda, \omega)$-optimal if

$$\hat{\pi} \in \arg\max_{\pi} J_{\hat{R}}(\pi) - \lambda \omega(\pi). \qquad (2)$$

Typically, $\lambda$ punishes large deviations from some reference policy $\pi_{\mathrm{ref}}$, e.g. with the regularization function given by the KL-divergence $\omega(\pi) = \mathbb{D}_{\mathrm{KL}}(\pi || \pi_{\mathrm{ref}})$. $\pi_{\mathrm{ref}}$ may also be used to collect training data for the reward learning algorithm, in which case we may assume $D = D^{\pi_{\mathrm{ref}}}$ in Equation (1). However, most of our results do not depend on these specific instantiations.

**Regret minimization** The aim of the previous two steps is for the policy $\hat{\pi}$ to have low regret $\mathrm{Reg}^R(\hat{\pi})$ under the true reward function $R$. Our question is thus if and when it is sufficient to ensure that $\hat{R}$ satisfies Equation (1), in order to guarantee that a policy $\hat{\pi}$ optimal according to Equation (2) has low regret $\mathrm{Reg}^R(\hat{\pi})$.

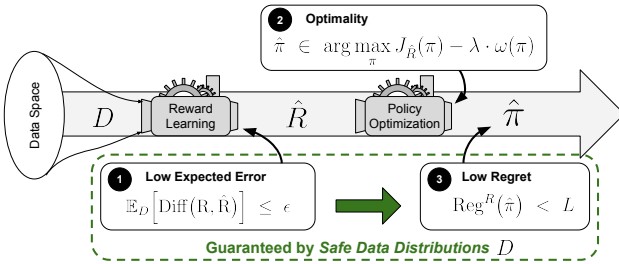

*Figure 2.* An abstract model of the classical reward learning pipeline. A reward model $\hat{R}$ is trained to approximate the true reward function $R$ under some data distribution $D$. The training process converges when $\hat{R}$ is similar to $R$ in expectation (see ❶). In the second step, a policy $\hat{\pi}$ is trained to achieve high learned reward, possibly involving a regularization (see ❷). We are interested in the question of when exactly this training process guarantees that $\hat{\pi}$ has low regret. More formally, we call a data distribution $D$ *safe* whenever the implication ❶ $\Longrightarrow$ ❸ holds for all reward models $\hat{R}$ that satisfy ❶.

## 2.2. Safe data distributions

We now make the elaborations from the previous subsections more concrete by providing a formal definition of a *safe data distribution*. In particular, we say that a data distribution $D$ is safe, whenever it holds that for every reward model $\hat{R}$ that satisfies Equation (1) for $D$, all optimal policies of $\hat{R}$ have low regret. We provide a visualization of this concept in Figure 2 and a formal definition in Theorem 2.1.

**Definition 2.1** (Safe- and unsafe data distributions). For a given MDP $\langle \mathcal{S}, \mathcal{A}, \tau, \mu_0, R, \gamma \rangle$, let $\epsilon > 0$, $L \in [0, 1]$, and $\lambda \in [0, \infty)$. Let $\omega$ be a continuous function with $\omega(\pi) \geq 0$ for all $\pi \in \Pi$. Then the set of *safe data distributions* $\mathbf{safe}(R, \epsilon, L, \lambda, \omega)$ is the set of all distributions $D \in \Delta(\mathcal{S} \times \mathcal{A})$ such that for all possible reward models $\hat{R} : \mathcal{S} \times \mathcal{A} \to \mathbb{R}$ and policies $\hat{\pi} : \mathcal{S} \to \Delta(\mathcal{A})$ that satisfy the following two properties:

1. **Low expected error:** $\hat{R}$ is $\epsilon$-close to $R$ under $D$, i.e.,
$$\mathbb{E}_{(s,a) \sim D} \left[ \frac{|\hat{R}(s,a) - R(s,a)|}{\text{range } R} \right] \leq \epsilon.$$

2. **Optimality:** $\hat{\pi}$ is $(\lambda, \omega)$-optimal with respect to $\hat{R}$, i.e.
$$\hat{\pi} \in \arg\max_\pi J_{\hat{R}}(\pi) - \lambda\omega(\pi).$$

we can guarantee that $\hat{\pi}$ has regret smaller than $L$, i.e.:

3. **Low regret:** $\hat{\pi}$ has a regret smaller than $L$ with respect to $R$, i.e., $\text{Reg}^R(\hat{\pi}) < L$.

Similarly, we define the set of *unsafe data distributions* to be the complement of $\mathbf{safe}(R, \epsilon, L, \lambda, \omega)$:

$$\mathbf{unsafe}(R, \epsilon, L, \lambda, \omega) \coloneqq \Delta(\mathcal{S} \times \mathcal{A}) \setminus \mathbf{safe}(R, \epsilon, L, \lambda, \omega).$$

Thus, $\mathbf{unsafe}(R, \epsilon, L, \lambda, \omega)$ consists of the data distributions $D$ for which there *exists* a reward model $\hat{R}$ that is $\epsilon$-close to $R$ and a policy $\hat{\pi}$ that is $(\lambda, \pi)$-optimal with respect to $\hat{R}$, but such that $\hat{\pi}$ has large regret $\text{Reg}^R(\hat{\pi}) \geq L$. In this sense, we are operating under a worst-case framework for the reward model and policy learned by our training algorithms. Note that $\epsilon$ and $L$ are free parameters in our definition; they are a measure of how well we can approximate the true reward function $R$, and what regret we find acceptable, respectively.

Whenever we consider the unregularized case ($\lambda = 0$ or $\omega = 0$), we drop the $\lambda$ and $\omega$ to ease the notation and just use $\mathbf{safe}(R, \epsilon, L)$ and $\mathbf{unsafe}(R, \epsilon, L)$ instead. Lastly, we mention that while we use the mean absolute error (MAE) in condition 1, one could in principle also work with the mean-squared error. All our results then have analogous versions. We explain this in Appendix B.3.

*Note: Throughout this paper, we will use the terminology that a data distribution $D$ "allows for error-regret mismatch" as a colloquial term to express that $D \in \mathbf{unsafe}(R, \epsilon, L, \lambda, \omega)$.*

## 3. Error-regret mismatch for unregularized policy optimization

In this section, we investigate the case where no regularization is used in the policy optimization stage. We seek to determine if it is sufficient for a reward model to be close to the true reward function on a data distribution in order to ensure low regret for the learned policy.

In our first result, we show that under certain conditions, a low expected error $\epsilon$ does indeed guarantee that policy optimization will yield a policy with low regret.

**Proposition 3.1.** *Let $\langle \mathcal{S}, \mathcal{A}, \tau, \mu_0, R, \gamma \rangle$ be an arbitrary MDP, let $L \in (0, 1]$, and let $D \in \Delta(\mathcal{S} \times \mathcal{A})$ be a positive data distribution (i.e., a distribution such that $D(s, a) > 0$ for all $(s, a) \in \mathcal{S} \times \mathcal{A}$). Then there exists an $\epsilon > 0$ such that $D \in \mathbf{safe}(R, \epsilon, L)$.*

The proof of Theorem 3.1 can be found in Appendix D.1 (see Theorem D.7) and is based on an application of Berge's maximum theorem (Berge, 1963), and the fact that the expected distance between the true reward function and the learned reward model under $D$ is induced from a norm. See Theorem D.15 for a similar result in which the expected error in rewards is replaced by an expected error in choice probabilities.

One might be inclined to conclude that the guarantee of Theorem 3.1 allows one to practically achieve low regret by ensuring a low error $\epsilon$ (as measured by Equation (1)). However, in the following result we provide a more detailed analysis that shows that low regret requires a prohibitively

low $\epsilon$:

**Proposition 3.2.** *Let the setting be as in Theorem 3.1. If $\epsilon > 0$ satisfies*

$$\epsilon < \frac{1-\gamma}{\sqrt{2}} \cdot \frac{\mathrm{range}\ J^R}{\mathrm{range}\ R} \cdot \min_{(s,a) \in \mathcal{S} \times \mathcal{A}} D(s,a) \cdot L$$

*then $D \in \mathbf{safe}(R, \epsilon, L)$.*

The proof can be found in Theorems D.11 and D.12, Appendix D.2. Theorem D.14 shows that the bound on $\epsilon$ is tight up to a factor of $\sqrt{2}$. This result is problematic in practice due to the dependence on the minimum of $D$. Realistic MDPs usually contain a massive amount of states and actions, which necessarily requires $D$ to give a very small support to at least some transitions. The dependence of the upper bound on $D$ also shows that there is no $\epsilon$ for which every distribution $D$ is guaranteed to be safe, as $\min_{(s,a) \in \mathcal{S} \times \mathcal{A}} D(s,a)$ can be arbitrarily small. We concretize this intuition by showing that in every MDP and for every $\epsilon > 0$, there exist weak assumptions for which a data distribution allows for a large error-regret mismatch.

**Proposition 3.3.** *Let $M = \langle \mathcal{S}, \mathcal{A}, \tau, \mu_0, R, \gamma \rangle$ be an MDP, $D \in \Delta(\mathcal{S} \times \mathcal{A})$ a data distribution, $\epsilon > 0$, and $L \in [0,1]$. Assume there exists a policy $\hat{\pi}$ with the property that $\mathrm{Reg}^R(\hat{\pi}) \geq L$ and $D(\mathrm{supp}\ D^{\hat{\pi}}) < \epsilon$, where $\mathrm{supp}\ D^{\hat{\pi}}$ is defined as the set of state-action pairs $(s,a) \in \mathcal{S} \times \mathcal{A}$ such that $D^{\hat{\pi}}(s,a) > 0$. In other words, there is a "bad" policy for $R$ that is not very supported by $D$. Then, $D$ allows for error-regret mismatch to occur, i.e., $D \in \mathbf{unsafe}(R, \epsilon, L)$.*

The proof of Theorem 3.3 can be found in Appendix C.2 (see Theorem C.5). The intuition is straightforward: There exists a reward model $\hat{R}$ that is very similar to the true reward function $R$ outside the support of $D^{\hat{\pi}}$ but has very large rewards for the support of $D^{\hat{\pi}}$. Because $D(\mathrm{supp}\ D^{\hat{\pi}})$ is very small, this still allows $\hat{R}$ to have a very small expected error w.r.t. to $D$, while $\hat{\pi}$, the optimal policy for $\hat{R}$, will have regret at least $L$. To avoid confusions, we show in Theorem C.7 that the assumptions on $\epsilon$ in Theorem 3.2 and Theorem 3.3 cannot hold simultaneously. This is as expected since otherwise the *conclusions* of these propositions would imply that a data distribution can be both safe and unsafe.

Note that the conditions for unsafe data distributions in Theorem 3.3 also cover positive data distributions (that we showed to be eventually safe for small enough $\epsilon$ in Theorem 3.1). Furthermore, especially in very large MDPs, it is very likely that the data distribution will not sufficiently cover large parts of the support of some policies, especially since the number of (deterministic) policies grows exponentially with the number of states. Sometimes, this can lead to *all* data distributions being unsafe, as we show in the following corollary:

**Corollary 3.4.** *Let $M = \langle \mathcal{S}, \mathcal{A}, \tau, \mu_0, R, \gamma \rangle$ be an MDP, $\epsilon > 0$, and $L \in [0,1]$. Assume there exists a set of policies $\Pi_L$ with:*

- $\mathrm{Reg}^R(\pi) \geq L$ *for all $\pi \in \Pi_L$;*

- $\mathrm{supp}\ D^\pi \cap \mathrm{supp}\ D^{\pi'} = \emptyset$ *for all $\pi, \pi' \in \Pi_L$; and*

- $|\Pi_L| \geq 1/\epsilon$.

*Then $\mathbf{unsafe}(R, \epsilon, L) = \Delta(\mathcal{S} \times \mathcal{A})$, i.e.: all distributions are unsafe.*

The proof of Theorem 3.4 can be found in Appendix C.2 (see Theorem C.6).

Theorem 3.4 outlines sufficient conditions for a scenario where all possible data distributions are unsafe for a given MDP. This happens when there exist *many* different policies with large regret and disjoint support, which requires there to be a large action space. This could for example happen in the case of a language model interacting with a user if there are many mutually distinct *styles* to answer unsafe queries. We illustrated this concern in slightly more detail in the introduction, and in full detail in Appendix B.4. More generally, we believe this intuition could be turned into a concrete theoretical result for general MDPs by assuming that for each state, there are many actions that are equally bad under the true reward function but induce the same transition-dynamics. This could be studied in the context of MDPs with symmetries (van der Pol et al., 2021) and might allow to prove the existence of the bad policy $\hat{\pi}$ from Theorem 3.3 or the set of bad policies $\Pi_L$ from Theorem 3.4. We leave such an investigation to future work.

We conclude by stating the main result of this section, which unifies all previous results and derives the most general conditions, i.e. *necessary and sufficient* conditions, for when exactly a data distribution allows for error-regret mismatch to occur:

**Theorem 3.5.** *For all MDPs $\langle \mathcal{S}, \mathcal{A}, \tau, \mu_0, R, \gamma \rangle$ and $L \in [0,1]$, there exists a matrix $M$ such that for all $\epsilon > 0$ and $D \in \Delta(\mathcal{S} \times \mathcal{A})$ we have:*

$$D \in \mathbf{safe}(R, \epsilon, L) \iff M \cdot D > \epsilon \cdot \mathrm{range}\ R \cdot \mathbf{1}, \quad (3)$$

*where we use the vector notation of $D$, and $\mathbf{1}$ is a vector containing all ones.*

The proof of Theorem 3.5 can be found in Appendix C.3 (see Theorem C.16) and largely relies on geometric arguments that arise from comparing the set of unsafe reward models and the set of reward models that are close to the true reward function. Interestingly, this means that the set of *safe* data distributions resembles a polytope, in the sense that it is a convex set and is defined by the intersection of an open

polyhedral set (defined by the system of strict inequalities $M \cdot D > \epsilon \cdot \operatorname{range} R \cdot \mathbf{1}$), and the closed data distribution simplex.

While Theorem 3.5 only proves the existence of such a matrix $M$, we provide further results and analyses in the appendix, namely:

1. In Appendix C.3.2 we derive closed-form expressions of the rows of matrix $M$, and show that its entries depend on multiple factors, such as the original reward function $R$, the state transition distribution $\tau$, and the set of deterministic policies that achieve regret at least $L$.

2. In Appendix C.3.3 we provide an algorithm to compute matrix $M$.

3. In Appendix C.3.4 we provide a worked example of computing and visualizing the set of safe distributions for a toy example.

Lastly, we note that $M$ does not depend on $\epsilon$, and $M$ only contains non-negative entries (see Appendix C.3.2). This allows us to recover Theorem 3.1, since by letting $\epsilon$ approach zero, the set of data distributions that fulfill the conditions in Equation (3) approaches the entire data distribution simplex. On the other hand, the dependence of $M$ on the true reward function and the underlying MDP implies that computing $M$ is infeasible in practice since many of these components are not known, restricting the use of $M$ to theoretical analysis.

## 4. Error-regret mismatch for regularized policy optimization

In this section, we investigate the error-regret mismatch for regularized policy optimization.

First, we prove that for almost any reference policy $\pi_{\mathrm{ref}}$ that achieves regret $L$ and minimizes the regularization term $\omega$, there exists a sufficiently small $\epsilon$ such that reward learning within $\epsilon$ of the true reward function preserves the regret bound $L$.

**Proposition 4.1.** *Let $\lambda \in (0, \infty)$, let $\langle \mathcal{S}, \mathcal{A}, \tau, \mu_0, R, \gamma \rangle$ be any MDP, and let $D \in \Delta(\mathcal{S} \times \mathcal{A})$ be any data distribution that assigns positive probability to all transitions. Let $\omega : \Pi \to \mathbb{R}$ be a continuous regularization function that has a reference policy $\pi_{\mathrm{ref}}$ as a minimum.[1] Assume that $\pi_{\mathrm{ref}}$ is not $(\lambda, \omega)$-optimal for $R$ and let $L = \operatorname{Reg}^R(\pi_{\mathrm{ref}})$. Then there exists $\epsilon > 0$ such that $D \in \mathbf{safe}(R, \epsilon, L, \lambda, \omega)$.*

The proof of Theorem 4.1 can be found in Appendix D.4 (see Theorem D.22) and is again an application of Berge's

---

[1] E.g., if $\pi_{\mathrm{ref}}(a \mid s) > 0$ for all $(s, a) \in \mathcal{S} \times \mathcal{A}$ and $\omega(\pi) := \mathbb{D}_{\mathrm{KL}}(\pi || \pi_{\mathrm{ref}})$, then the minimum is $\pi_{\mathrm{ref}}$.

theorem (Berge, 1963). Note that the regret bound $L$ is defined as the regret of the reference policy. This intuitively makes sense, as regularized policy optimization constrains the policy under optimization $\hat{\pi}$ to not deviate too strongly from the reference policy $\pi_{\mathrm{ref}}$, which will also constrain the regret of $\hat{\pi}$ to stay close to the regret of $\pi_{\mathrm{ref}}$. Under the conditions of Theorem 4.1, the regret of $\pi_{\mathrm{ref}}$ serves as an upper regret bound because for small enough $\epsilon$ the learned reward $\hat{R}$ and the true reward $R$ are so close that maximizing $\hat{R}$ also improves reward with respect to $R$. Furthermore, we note that it is also possible to derive a version in which the expected error in rewards is replaced by a KL divergence of choice probabilities, similar to Proposition D.15, by combining the arguments in that proposition with the arguments in Berge's theorem — see Theorem D.23.

Similar to Theorem 3.1, Theorem 4.1 does not guarantee the existence of a universal $\epsilon$ such that all data distributions $D$ are in $\mathbf{safe}(R, \epsilon, L, \lambda, \omega)$. In our next result, we show that such an $\epsilon$ does not exist, since for each $\epsilon$, there is a nontrivial set of data distributions that allows for error-regret mismatch to occur:

**Theorem 4.2.** *Let $\mathcal{M} = \langle \mathcal{S}, \mathcal{A}, \tau, \mu_0, R, \gamma \rangle$ be an arbitrary MDP, $\lambda \in (0, \infty)$, $L \in (0, 1)$, and $\omega : \Pi \to \mathcal{R}$ be a regularization function. Furthermore, let $\pi_*$ be a determinstic worst-case policy for $R$, meaning that $\operatorname{Reg}^R(\pi_*) = 1$. Let $C := C(\mathcal{M}, \pi_*, L, \lambda, \omega) < \infty$ be the constant defined in Equation (106) in the appendix. Let $\epsilon > 0$. Then for all data distributions $D \in \Delta(\mathcal{S} \times \mathcal{A})$ with*

$$D(\operatorname{supp} D^{\pi_*}) \leq \frac{\epsilon}{1 + C}, \tag{4}$$

*we have $D \in \mathbf{unsafe}(R, \epsilon, L, \lambda, \omega)$.*

The proof of Theorem 4.2 can be found in Appendix C.5 (see Theorem C.38). The general idea is as follows: To prove that $D$ is unsafe, define $\hat{R}$ to be equal to $R$ outside of $\operatorname{supp} D^{\pi_*}$, and very large in $\operatorname{supp} D^{\pi_*}$. If it is sufficiently large in this region, then regularized optimization leads to a policy $\hat{\pi}$ with $\operatorname{Reg}^R(\hat{\pi}) \geq L$. Finally, the condition that $D(\operatorname{supp} D^{\pi_*}) \leq \frac{\epsilon}{1+C}$ ensures that $\hat{R}$ has a reward error bounded by $\epsilon$.

Note that Theorem 4.2 is very general and covers a large class of different regularization methods. In Theorem C.40 we provide a specialized result for the case of $KL$-regularized policy optimization, and in Section 6 we investigate error-regret mismatch in the RLHF framework. At the end of our conceptual example described in the introduction and in detail in Appendix B.4, we also discuss the simple intuition that simply giving a low enough training probability to *some* unsafe actions can be enough to lead to unsafe reward inference and policy optimization even in the regularized case. This is in accordance with Theorem 4.2.

## 5. Generalization of the error measurement

Our results have so far expressed the error of the learned reward $\hat{R}$ in terms of Equation (1), i.e., in terms of the expected error of individual transitions. In this section, we show that many common reward learning training objectives can be upper-bounded in terms of the expected error metric defined in Equation (1). This in turn means that our negative results generalize to reward learning algorithms that use these other training objectives. We state all upper bounds for MDPs with finite time horizon $T$ (but note that these results directly generalize to MDPs with infinite time horizon by taking the limit of $T \to \infty$).

In the finite horizon setting, trajectories are defined as a finite list of states and actions: $\xi = s_0, a_0, s_1, ..., a_{T-1}$. We use $\Xi$ for the set of all trajectories of length $T$. As in the previous sections, $G : \Xi \to \mathbb{R}$ denotes the trajectory return function, defined as $G(\xi) = \sum_{t=0}^{T-1} \gamma^t \cdot R(s_t, a_t)$. We start by showing that low expected error in transitions implies low expected error in trajectory returns:

**Proposition 5.1.** *Given an MDP $\langle \mathcal{S}, \mathcal{A}, \tau, \mu_0, R, \gamma \rangle$, a data sampling policy $\pi : \mathcal{S} \to \Delta(\mathcal{A})$ and its resulting data distribution $D^\pi = \frac{1-\gamma}{1-\gamma^T} \cdot \eta^\pi$ and a second reward function $\hat{R} : \mathcal{S} \times \mathcal{A} \to \mathbb{R}$, we can upper bound the expected difference in trajectory evaluation as follows:*

$$\mathbb{E}_{\xi \sim \pi}\Big[ \big| G_R(\xi) - G_{\hat{R}}(\xi) \big| \Big] \leq \frac{1-\gamma^T}{1-\gamma} \cdot \mathbb{E}_{(s,a) \sim D^\pi} \Big[ \big| R(s,a) - \hat{R}(s,a) \big| \Big].$$

The proof of Theorem 5.1 can be found in Appendix C.4.1 (see Theorem C.24). Furthermore, a low expected error of trajectory returns implies a low expected error of choice distributions (a distance metric commonly used as the loss in RLHF (Christiano et al., 2017)). Namely, given a reward function $R$, define the probability of trajectory $\xi_1$ being preferred over $\xi_2$ to be:

$$p_R(\xi_1 \succ \xi_2) = \sigma(G_R(\xi_1) - G_R(\xi_2))$$
$$= \frac{\exp(G_R(\xi_1))}{\exp(G_R(\xi_1)) + \exp(G_R(\xi_2))}.$$

We then have:

**Proposition 5.2.** *Given an MDP $\langle \mathcal{S}, \mathcal{A}, \tau, \mu_0, R, \gamma \rangle$, a data sampling policy $\pi : \mathcal{S} \to \Delta(\mathcal{A})$ and a second reward function $\hat{R} : \mathcal{S} \times \mathcal{A} \to \mathbb{R}$, we can upper bound the expected KL divergence over trajectory preference distributions as follows:*

$$\mathbb{E}_{\xi_1, \xi_2 \sim \pi \times \pi} \Big[ \mathbb{D}_{KL}\big(p_R(\cdot|\xi_1, \xi_2) || p_{\hat{R}}(\cdot|\xi_1, \xi_2)\big) \Big]$$
$$\leq 2 \cdot \mathbb{E}_{\xi \sim \pi} \big[ |G_R(\xi) - G_{\hat{R}}(\xi)| \big].$$

The proof of Theorem 5.2 can be found in Appendix C.4.1 (see Theorem C.25).

Finally, in some RLHF scenarios, for example in RLHF with prompt-response pairs, one prefers to only compare trajectories with a common starting state. In the following proposition, we upper bound the expected error of choice distributions with trajectories that share a common starting state by the expected error of choice distributions with arbitrary trajectories:

**Proposition 5.3.** *Given an MDP $\langle \mathcal{S}, \mathcal{A}, \tau, \mu_0, R, \gamma \rangle$, a data sampling policy $\pi : \mathcal{S} \to \Delta(\mathcal{A})$ and a second reward function $\hat{R} : \mathcal{S} \times \mathcal{A} \to \mathbb{R}$, we can upper bound the expected KL divergence of preference distributions over trajectories with a common starting state as follows:*

$$\mathbb{E}_{\substack{s_0 \sim \mu_0, \\ \xi_1, \xi_2 \sim \pi(s_0)}} \Big[ \mathbb{D}_{KL}\big(p_R(\cdot|\xi_1, \xi_2) || p_{\hat{R}}(\cdot|\xi_1, \xi_2)\big) \Big]$$
$$\leq \frac{\mathbb{E}_{\xi_1, \xi_2 \sim \pi \times \pi} \big[ \mathbb{D}_{KL}\big(p_R(\cdot|\xi_1, \xi_2) || p_{\hat{R}}(\cdot|\xi_1, \xi_2)\big) \big]}{\min_{s' \in \mathcal{S}, \mu_0(s') > 0} \mu_0(s')}.$$

The proof of Theorem 5.3 can be found in Appendix C.4.1 (see Theorem C.26).

## 6. Error-regret mismatch in RLHF

In this section we extend our results to reinforcement learning from human feedback (RLHF). We provide more general results for the class of KL-regularized policy optimization methods in Appendix C.4.5.

RLHF, especially in the context of large language models, is usually modeled in a *contextual bandit* setting (Ziegler et al., 2019; Stiennon et al., 2020; Bai et al., 2022; Ouyang et al., 2022; Rafailov et al., 2023). A *contextual bandit* $\langle \mathcal{S}, \mathcal{A}, \mu_0, R \rangle$ is defined by a set of states $\mathcal{S}$, a set of actions $\mathcal{A}$, a data distribution $\mu_0 \in \Delta(\mathcal{S})$, and a reward function $R : \mathcal{S} \times \mathcal{A} \to \mathbb{R}$. The goal is to learn a policy $\pi : \mathcal{S} \to \Delta(\mathcal{A})$ that maximizes the expected return $J(\pi) = \mathbb{E}_{s \sim \mu_0, a \sim \pi(\cdot|s)} [R(s,a)]$. In the context of language models, $\mathcal{S}$ is usually called the set of *prompts* or *contexts*, and $\mathcal{A}$ the set of *responses*.

We state the following theorem using a version of Theorem 2.1 tailored to the RLHF setting. In particular, we replace the similarity metric (property 1 of Theorem 2.1) with the expected similarity in choice probabilities. A precise mathematical definition can be found in Appendix C.4.3. We denote the resulting sets of safe- and unsafe data distributions by $\mathbf{safe}^{\mathrm{RLHF}}\big(R, \epsilon, L, \lambda, \mathbb{D}_{\mathrm{KL}}(\cdot||\pi_{\mathrm{ref}})\big)$ and $\mathbf{unsafe}^{\mathrm{RLHF}}\big(R, \epsilon, L, \lambda, \mathbb{D}_{\mathrm{KL}}(\cdot||\pi_{\mathrm{ref}})\big)$.

By making use of the specifics of this setting, we can derive more interpretable and stronger results. In particular, we specify a set of reference distributions for which performing KL-regularized policy optimization allows for error-regret

mismatch to occur.

**Theorem 6.1.** *Let* $\langle S, A, \mu_0, R \rangle$ *be a contextual bandit. Given* $L \in [0, 1)$, *we define for every state* $s \in S$ *the reward threshold:* $R_L(s) := (1 - L) \cdot \max_{a \in A} R(s, a) + L \cdot \min_{a \in A} R(s, a)$. *Lastly, let* $\pi_{\text{ref}} : S \to A$ *be an arbitrary reference policy for which it holds that for every* $(s, a) \in S \times A$, $\pi_{\text{ref}}(a|s) > 0$, *and there exists at least one action* $a_s \in A$ *such that* $R(s, a_s) < R_L(s)$ *and* $\pi_{\text{ref}}(a_s|s)$ *satisfies the following inequality:*

$$\pi_{\text{ref}}(a_s|s) \leq \frac{(R_L(s) - R(s, a_s)) \cdot \text{range } R}{L \cdot \exp\left(\frac{1}{\lambda} \cdot \text{range } R\right)} \cdot \frac{\epsilon^2}{4 \cdot \lambda^2}.$$

*Let* $D_\mu^{\text{ref}}(s, a) := \mu(s) \cdot \pi_{\text{ref}}(a|s)$ *for some* $\mu \in \Delta(S)$. *Then* $D_\mu^{\text{ref}} \in \textbf{unsafe}^{\text{RLHF}}\left(R, \epsilon, L, \lambda, \mathbb{D}_{KL}\left(\cdot || \pi_{\text{ref}}\right)\right)$

Intuitively, the theorem shows that even if we learn a reward model $\hat{R}$ that induces $\epsilon$-correct choice probabilities according to the data distribution generated from a reference policy $\pi_{\text{ref}}$, a policy that maximizes $\hat{R}$ with KL-penalty can still have regret $\geq L$ if $\pi_{\text{ref}}$ gives sufficiently low probability to bad actions. The proof of Theorem 6.1 can be found in Appendix C.4.4 (see Theorem C.32). We expect the conditions on the reference policy $\pi_{\text{ref}}$ to likely hold in real-world cases as the number of potential actions (or responses) is usually very large, and language models typically assign a large portion of their probability mass to only a tiny fraction of all responses. Hence, for every state/prompt $s$, a large majority of actions/responses $a$ have a very small probability $\pi_{\text{ref}}(a \mid s)$. See our conceptual example in the introduction and Appendix B.4 to make this intuition concrete.

# 7. Discussion

In this paper, we contributed to the foundations of reward learning theory by studying the relationship between the training error of the learned reward function and the regret of policies that then result from policy optimization. We showed that as the expected error of a reward model $\hat{R}$ goes to zero, the regret of the resulting policy (with or without regularization) also goes to zero (Theorem 3.1) or is bounded by the regret of a reference policy (Theorem 4.1). However, in Theorem 3.2 we showed that the training error needed to ensure a certain regret is proportional to the minimum of the data distribution $D$. Consequently, there exists no training error that can universally ensure low regret.

More specifically, low expected error of $\hat{R}$ does *not* ensure low regret for all realistic data distributions (Theorems 3.3, 4.2 and 6.1). We refer to this phenomenon as *error-regret mismatch*. This is due to policy optimization involving a *distributional shift*. Moreover, for some MDPs with very large state-action spaces there does not exist *any* safe data distribution relative to a reasonable reward model error and desired regret bound (Theorem 3.4). We also showed that

our results generalize to other data sources, such as preferences over trajectories and trajectory scores (Theorems 5.1 to 5.3), supporting the conclusion that this issue is a fundamental problem of reward learning.

Lastly, for unregularized optimization, we derive *necessary and sufficient* conditions that allow us to determine the set of safe data distributions for arbitrary MDPs, thereby fully answering when a data distribution is safe (Theorem 3.5).

## 7.1. Limitations and future work

Our work focuses on a worst-case setting regarding the learned reward function and optimal policy. Future work could account for the inductive biases of common optimization procedures and consider non-optimal policies.

It is also important to theoretically analyze *improved* reward learning and policy optimization procedures. Empirical work has explored reward model ensembles (Coste et al., 2023), weight-averaged reward models (Ramé et al., 2024), and iterated data-smoothing for multi-armed bandits (Zhu et al., 2024). Recent efforts address learning reward models on online data to mitigate distribution shifts (Lang et al., 2024a) and even provide theoretical insights for linear reward functions (Song et al., 2024). We hope a careful theoretical analysis of these settings, in similar generality to our work, can improve upon the "theoretical baseline" we establish.

Another important direction to explore is whether optimizing an implicit reward model (as used by direct preference optimization (Rafailov et al., 2023) and its many derivatives) in place of an explicitly learned reward model improves the robustness to error-regret mismatch.

Lastly, there are other ways to improve safety. For example, one could research evaluation methods for learned reward functions that go beyond looking at the training error, e.g. by using interpretability methods (Michaud et al., 2020; Jenner & Gleave, 2022) or finding better ways to quantify reward function distance (Gleave et al., 2020; Skalse et al., 2024).

## Impact statement

Reward learning methods such as RLHF are widely used to steer the behavior of frontier models. Thus, it is important that reward models are robust and reliable. We point out a theoretical challenge to the robustness of reward models to policy optimization. We hope that this stimulates further research in overcoming this challenge. Since our work is purely theoretical, we do not foresee negative societal consequences.

## Author contributions

**Lukas Fluri** and **Leon Lang** are the core contributors who developed the technical results and wrote a large part of the main paper and all of the appendix. While many results arose from strong contributions by both together, Lukas had a particular focus and impact on the general unregularized optimization result (Theorem 3.5) and the generalization of the error measurement (Section 5), whereas Leon had a particular focus and impact on the results showing that in the limit, a data distribution becomes safe (Theorems 3.1 and 4.1) and the general regularized optimization results (Section 4).

**Joar Skalse** developed the project idea and provided close supervision during the project's duration by providing feedback and ideas and editing the paper.

**Alessandro Abate**, **Patrick Forré**, and **David Krueger** advised on the project by providing helpful feedback on the project idea, as well as reviewing and improving drafts of the paper. Furthermore, Patrick had the initial idea of using Berge's theorem to prove our positive results (Theorems 3.1 and 4.1).

## Acknowledgements

Lukas Fluri and Leon Lang are grateful for financial support provided by the Berkeley Existential Risk Initiative for this project. Leon Lang furthermore thanks Open Philanthropy for financial support.

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

# APPENDIX

This appendix develops the theory outlined in the main paper in a self-contained and complete way, including all proofs. In Appendix A we include an extended related work section. In Appendix B, we present the setup of all concepts and the problem formulation, as was already contained in the main paper. In Appendix C, we present all "negative results". Conditional on an error threshold in the reward model, these results present conditions for the data distribution that allow reward models to be learned that allow for error-regret mismatch. That section also contains Theorem C.16 which is an equivalent condition for the absence of error-regret mismatch but could be considered a statement about error-regret mismatch by negation. In Appendix D, we present sufficient conditions for *safe optimization* in several settings. Typically, this boils down to showing that given a data distribution, a *sufficiently small* error in the reward model guarantees that its optimal policies have low regret.

## Contents of the Appendix

## A. Extended related work

**Reward Learning**    Reward learning is a key concept in reinforcement learning that involves learning the reward function for complex tasks with latent and difficult-to-specify reward functions. Many methods have been developed to incorporate various types of human feedback into the reward learning process (Wirth et al., 2017; Ng et al., 2000; Bajcsy et al., 2017; Jeon et al., 2020).

**Challenges in Reward Learning**    Reward learning presents several challenges (Casper et al., 2023; Lang et al., 2024b; Skalse & Abate, 2023; 2024), such as *reward misgeneralization*, where the reward model learns a different reward function that performs well on in-distribution data but differs strongly on out-of-distribution data (Skalse et al., 2023). This can lead to unintended consequences in real-world applications.

Reward misgeneralization can also result in *reward hacking* (Krakovna, 2020), a consequence of Goodhart's law (Goodhart, 1984; Zhuang & Hadfield-Menell, 2020; Hennessy & Goodhart, 2023; Strathern, 1997; Karwowski et al., 2023). Reward hacking has been extensively studied both theoretically (Skalse et al., 2022; 2024; Zhuang & Hadfield-Menell, 2020) and empirically (Zhang et al., 2018; Farebrother et al., 2018; Cobbe et al., 2019; Krakovna, 2020; Gao et al., 2023; Tien et al., 2022).

**Offline RL**    In offline reinforcement learning, we aim to learn low-regret policies for an MDP $\langle \mathcal{S}, \mathcal{A}, \tau, \mu_0, R, \gamma \rangle$ where the reward function (and sometimes transition distribution (Wang et al., 2022b; Uehara & Sun, 2021)) is unknown and must be learned from an offline dataset $\{(s, a, r)_i\}_{i=1}^n$ sampled from a data distribution $D \in \Delta(\mathcal{S} \times \mathcal{A})$. A key research question is understanding what data coverage conditions ensure learning a near-optimal policy with an *efficient* sample complexity. Existing theoretical work primarily falls into two categories, covering both *MDPs* (Foster et al., 2021; Wang et al., 2022b; 2020; Amortila et al., 2020; Uehara & Sun, 2021; Uehara et al., 2021) and *contextual bandits* (Nika et al., 2024; Cen et al., 2024):

*Lower bound results* prove that various data-coverage conditions are insufficient for sample-efficient offline RL by establishing worst-case sample complexity bounds. Research in this area (Foster et al., 2021; Wang et al., 2022b; 2020; Amortila et al., 2020; Nika et al., 2024) identifies adversarial MDPs that satisfy specific data-coverage conditions where achieving low regret is either computationally intractable due to excessive sample requirements (Foster et al., 2021; Wang et al., 2022b; 2020; Nika et al., 2024) or fundamentally impossible regardless of sample size (Amortila et al., 2020).

*Upper bound results*, on the other hand, establish positive guarantees under specific structural assumptions. Works in this category (Wang et al., 2022b; 2020; Uehara & Sun, 2021; Nika et al., 2024; Cen et al., 2024; Song et al., 2024) develop algorithms with provable sample-efficiency bounds by making structural assumptions about the MDP structure, reward learning process, or policy optimization approach.

Intuitively, the quality of a reward model that is being approximated from a finite dataset is influenced by two key factors: the dataset size $n$ and the dataset quality, specifically how well the data distribution $D$ *covers* the data space $\mathcal{S} \times \mathcal{A}$. Prior work confirms this intuition, with most works deriving variants of the following template (see for example recent work (Nika et al., 2024)): Regret $\in \mathcal{O}\left(\text{poly}\left(\frac{\text{Cov} \cdot \text{Struct}}{n}\right)\right)$. Here, *Cov* represents some measure of the coverage of $D$, while *Struct* captures the structural assumptions of the specific approach. Such structural assumptions may include: realizability of function classes (Wang et al., 2022b; Uehara & Sun, 2021; Foster et al., 2021; Nika et al., 2024), linear function approximation (Nika et al., 2024; Cen et al., 2024; Wang et al., 2022b), and various constraints on reward- or policy functions (Wang et al., 2020;

Uehara & Sun, 2021; Nika et al., 2024).

Our paper differs from these works in two key aspects: a) we explicitly analyze how the reward modeling error $\epsilon$ affects the final policy regret, rather than focusing on the number of samples (prior works only implicitly consider $\epsilon$), and b) we examine worst-case scenarios instead of probabilistic guarantees. The most relevant work in this area is (Song et al., 2024), which analyzes RLHF specifically. Their setup in section 3, combined with their Assumption 3, perfectly recovers our safe distribution definition (see Theorem 2.1) when applied to the special case of RLHF and when using the mean squared error metric. Their Theorem 4.2 demonstrates that $\text{Regret} \in \mathcal{O}\big(\text{Cov} \cdot \sqrt{\epsilon}\big)$, where the square root emerges from using the mean squared error during the reward learning step.

While Song et al. (2024) focus on RLHF with mean-squared error metric, we provide similar results for general classes of regularized and unregularized policy optimization (for both MDPs and contextual bandits), as well as a wide range of different error metrics. Similar to prior sample-complexity results, we investigate the influence of different coverage constraints on regret guarantees. For our initial results (Theorems 3.1, 3.2 and 4.1) we use the condition $\min_{(s,a)} D(s,a) > 0$. Since we assume that all states of our MDPs are reachable, this is equivalent to a full coverage condition (see Table 1 of (Uehara & Sun, 2021) for an overview of different coverage conditions). We then relax the constraints to partial coverage constraints and prove several negative results (Theorems 3.3, 4.2 and 6.1). Finally, we fully generalize our results from Theorems 3.1 to 3.4 into a single theorem (Theorem 3.5) which allows us to determine the worst-case safety of *arbitrary* data distributions. To the best of our knowledge, we are the first work to achieve such a level of generality.

**Advancements in Addressing Distribution Shifts**    Several approaches have been proposed to address the issue of out-of-distribution robustness in reward learning, such as ensembles of conservative reward models (Coste et al., 2023), averaging weights of multiple reward models (Ramé et al., 2024), iteratively updating training labels (Zhu et al., 2024), on-policy reward learning (Lang et al., 2024a), and distributionally robust planning (Zhan et al., 2023). Recently, Kwa et al. (2024) show that RLHF and Conditioning can be provably safe under fairly strong structural assumptions—such as deterministic transitions, light-tailed reward errors, and independence between true and proxy rewards. Furthermore, Laidlaw et al. (2024) consider a setting where the learned and true reward functions are positively correlated under a reference policy. They prove that maximizing the proxy reward with a chi-squared divergence penalty yields regret no worse than that of the reference policy. In experiments, they approximate this regularized objective and report favorable results.

Our work further emphasizes the usefulness of exploring additional assumptions or methods to mitigate the perils of distribution shift, as we show that without any additional assumptions, there are next to no guarantees. We therefore hope that our work can serve as a theoretical baseline, that people can use to express and analyze their new assumptions or methods.

In classical machine learning, research in out-of-distribution generalization has a long history, and a rich literature of methods exists (Li et al., 2022; Zhou et al., 2022; Wang et al., 2022a; Liu et al., 2021; Li et al., 2023; Yoon et al., 2023). These methods could potentially be adapted to address distribution shift challenges in reinforcement learning.

**Contextual Bandits**    In Section 6 we work in the contextual bandit setting and derive variants of our results for RLHF. Several theoretical results have been developed that investigate the challenge of RLHF (Xiong et al., 2024; Zhu et al., 2023; Ji et al., 2023; Mehta et al., 2023) and reward learning in general, (Agarwal et al., 2012; Foster et al., 2020) in the contextual bandit setting. In our Theorem 6.1 we show that in the worst-case setting, and without any additional assumptions, many common data distributions are unsafe. On the other hand, these works develop safety guarantees in settings with more structural assumptions such as various restrictions on the reward functions (Xiong et al., 2024; Zhu et al., 2023), focusing on the probability and efficiency of safety guarantees (Zhu et al., 2023; Ji et al., 2023; Agarwal et al., 2012), and developing active learning algorithms (Mehta et al., 2023).

**Direct Preference Optimization**    Direct preference optimization (DPO) (Rafailov et al., 2023) is a recent technique that allows to directly optimize a policy via an implicitly defined reward model which promises to mitigate some of the common issues with classic reward model training like such as training stability. DPO's empirical performance is promising and many recent works are trying to further improve and extend upon the base idea. *RPO* (Liu et al., 2024) adds an imitation loss from a baseline policy to regularize DPO and provably mitigate overoptimization. *SimPO* (Meng et al., 2024) simplifies DPO by removing the need for a reference model and using the average log-likelihood of tokens as the reward. *IPO* (Garg et al., 2025) avoids explicit external reward models by letting LLMs select samples for DPO themselves. Lastly, $\chi^{PO}$ replaces the KL-regularization of DPO with $\chi^2$ regularization, thereby improving robustness against overoptimization and achieving

sample efficiency guarantees.

Since our work explicitly analyzes the conditions under which a reward model might misgeneralize during training, our work does not directly translate to this family of DPO algorithms. However, we do consider it important future work to explore if and how DPO might be more robust to the issue of error-regret mismatch.

**Alternative Formalizations of Reward and Utility**   We would like to note that throughout this work, we study the classical MDP setting, where utility of a policy is measured by an expected value of a (potentially discounted) cumulative sum of a scalar reward signal (see the definition of $J$ in Section 2). This definition is based on the *reward hypothesis* (Sutton, 2004; Littman, 2019; Sutton & Barto, 2018) which states that:

> *That all of what we mean by goals and purposes can be well thought of as maximization of the expected value of the cumulative sum of a received scalar signal (reward).*

While this is the default setting, several works investigate if, or under which conditions this formalization is sufficient to express the wide variety of goals and purposes that one might be interested in. Toward that end, Shakerinava & Ravanbakhsh (2022) propose axioms that are necessary and sufficient for Markovian rewards to model preference relations. These axioms are later generalized by Bowling et al. (2023) to accommodate more general settings such as discounted reward and episodic settings. Pitis (2019) show how utility functions with fixed discount factors fail to model some types of preferences and then argue for a state-action-dependent discount factor. Similarly, Abel et al. (2021) investigate the expressivity of Markovian rewards and identify tasks that cannot be modeled with Markovian rewards.

# B. Introduction

## B.1. Preliminaries

A *Markov Decision Process* (MDP) is a tuple $\langle \mathcal{S}, \mathcal{A}, \tau, \mu_0, R, \gamma \rangle$ where $\mathcal{S}$ is a set of *states*, $\mathcal{A}$ is a set of *actions*, $\tau : \mathcal{S} \times \mathcal{A} \to \Delta(\mathcal{A})$ is a *transition function*, $\mu_0 \in \Delta(S)$ is an *initial state distribution*, $R : \mathcal{S} \times \mathcal{A} \to \mathbb{R}$ is a *reward function*, and $\gamma \in (0, 1)$ is a *discount rate*. A *policy* is a function $\pi : \mathcal{S} \to \Delta(\mathcal{A})$. A *trajectory* $\xi = \langle s_0, a_0, s_1, a_1, ... \rangle$ is a possible path in an MDP. The *return function* $G$ gives the cumulative discounted reward of a trajectory, $G(\xi) = \sum_{t=0}^{\infty} \gamma^t R(s_t, a_t, s_{t+1})$, and the *evaluation function* $J$ gives the expected trajectory return given a policy, $J(\pi) = \mathbb{E}_{\xi \sim \pi} [G(\xi)]$. A policy maximizing $J$ is an *optimal policy*. The *state-action occupancy measure* is a function $\eta : \Pi \to \mathbb{R}^{|\mathcal{S} \times \mathcal{A}|}$ which assigns each policy $\pi \in \Pi$ a vector of occupancy measure describing the discounted frequency that a policy takes each action in each state. Formally, $\eta(\pi)(s, a) = \eta^\pi(s, a) = \sum_{t=0}^{\infty} \gamma^t \cdot P(s_t = s, a_t = a \mid \xi \sim \pi)$. Note that by writing the reward function $R$ as a vector $\vec{R} \in \mathbb{R}^{|\mathcal{S} \times \mathcal{A}|}$, we can split $J$ into a linear function of $\pi$: $J(\pi) = \eta^\pi \cdot \vec{R}$. The *value function* $V$ of a policy encodes the expected future discounted reward from each state when following that policy. We use $\mathcal{R}$ to refer to the set of all reward functions. When talking about multiple rewards, we give each reward a subscript $R_i$, and use $J_i$, $G_i$, and $V_i^\pi$, to denote $R_i$'s evaluation function, return function, and $\pi$-value function.

## B.2. Problem formalization

The standard RL process using reward learning works roughly like this:

1. You are given a dataset of transition-reward tuples $\{(s_i, a_i, r_i)\}_{i=0}^n$. Here, each $(s_i, a_i) \in \mathcal{S} \times \mathcal{A}$ is a transition from some (not necessarily known) MDP $\langle \mathcal{S}, \mathcal{A}, \tau, \mu_0, R, \gamma \rangle$ that has been sampled using some distribution $D \in \Delta(\mathcal{S} \times \mathcal{A})$, and $r_i = R(s_i, a_i)$. The goal of the process is to find a policy $\hat{\pi}$ which performs roughly optimally for the unknown true reward function $R$. More formally: $J_R(\hat{\pi}) \approx \max_{\pi \in \Pi} J_R(\pi)$.

2. Given some error tolerance $\epsilon \in \mathbb{R}$, a reward model $\hat{R} : \mathcal{S} \times \mathcal{A} \to \mathbb{R}$ is learned using the provided dataset. At the end of the learning process $\hat{R}$ satisfies some optimality criterion such as: $\mathbb{E}_{(s,a) \sim D} \left[ |\hat{R}(s, a) - R(s, a)| \right] < \epsilon$

3. The learned reward model $\hat{R}$ is used to train a policy $\hat{\pi}$ that fulfills the following optimality criterion: $\hat{\pi} = \arg\max_{\pi \in \Pi} J_{\hat{R}}(\pi)$.

The problem is that training $\hat{\pi}$ to optimize $\hat{R}$ effectively leads to a distribution shift, as the transitions are no longer sampled from the original data distribution $D$ but some other distribution $\hat{D}$ (induced by the policy $\hat{\pi}$). Depending on the definition

of $D$, this could mean that there are no guarantees about how close the expected error of $\hat{R}$ to the true reward function $R$ is (i.e., $\mathbb{E}_{(s,a)\sim\hat{D}}\left[|\hat{R}(s,a) - R(s,a)|\right]$ could not be upper-bounded).

This means that we have no guarantee about the performance of $\hat{\pi}$ with respect to the original reward function $R$, so it might happen that $\hat{\pi}$ performs arbitrarily bad under the true reward $R$: $J_R(\hat{\pi}) \ll \max_\pi J_R(\pi)$.

If for a given data distribution $D$ there exists a reward model $\hat{R}$ such that $\hat{R}$ is close in expectation to the true reward function $R$ but it is possible to learn a policy that performs badly under $J_R$ despite being optimal for $\hat{R}$, we say that $D$ *allows for error-regret mismatch* and that $\hat{R}$ *has an error-regret mismatch*.

## B.3. The mean-squared error as an alternative distance measure

In the main paper, particular in Theorem 2.1, we use the mean absolute error (MAE) as our error measure in the reward function. In this appendix section, we explain what changes in the results if one were to use the mean-squared error (MSE) instead.

We define the mean-squared error by

$$d_D^{\mathrm{MSE}}(R, \hat{R}) := \mathbb{E}_{(s,a)\sim D}\left[\left(\frac{\hat{R}(s,a) - R(s,a)}{\operatorname{range} R}\right)^2\right].$$

This is like the usual MSE, with the difference that we divide by $\operatorname{range} R$ since the distance is only meaningful relative to the range of the true reward function $R$. In the main paper, we work with the following mean absolute error instead:

$$d_D^{\mathrm{MAE}}(R, \hat{R}) = \mathbb{E}_{(s,a)}\left[\frac{|\hat{R}(s,a) - R(s,a)|}{\operatorname{range} R}\right].$$

Then for any distance measure $d^X$ (with $X = \mathrm{MSE}$ or $X = \mathrm{MAE}$) involving a data distribution D, we can define the set of safe data distributions $\mathrm{safe}^X(R, \epsilon, L, \lambda, \omega)$, slightly generalizing Theorem 2.1: $\mathrm{safe}(R, \epsilon, L, \lambda, \omega)$ is the set of all distributions $D$ such that for all $\hat{R}$ that are $\epsilon$-close to $R$ according to $d_D^X$ and all $\hat{\pi}$ that are $(\lambda, \omega)$-optimal with respect to $\hat{R}$, we have $\mathrm{Reg}^R(\hat{\pi}) < L$. The complement of this set is $\mathrm{unsafe}^X(R, \epsilon, L, \lambda, \omega)$.

We now explain that for all of our results where in the main paper we talk about $\mathrm{safe}^{\mathrm{MAE}}$, there is a corresponding result for $\mathrm{safe}^{\mathrm{MSE}}$, and the same for $\mathrm{unsafe}^{\mathrm{MAE}}$ and $\mathrm{unsafe}^{\mathrm{MSE}}$.

### B.3.1. TRANSFER OF POSITIVE RESULTS

**Proposition B.1.** *If $D \in \mathrm{safe}^{\mathrm{MAE}}(R, \epsilon, L, \lambda, \omega)$, then $D \in \mathrm{safe}^{\mathrm{MSE}}(R, \epsilon^2, L, \lambda, \omega)$.*

*Proof.* Assume the condition. Let $\hat{R}, \hat{\pi}$ be such that $d_D^{\mathrm{MSE}}(R, \hat{R}) \leq \epsilon^2$ and $\hat{\pi}$ is $(\lambda, \omega)$-optimal with respect to $\hat{R}$. Due to Jensen's inequality, we have

$$
\begin{aligned}
d_D^{\mathrm{MAE}}(R, \hat{R})^2 &= \mathbb{E}_{(s,a)\sim D}\left[\frac{|\hat{R}(s,a) - R(s,a)|}{\operatorname{range} R}\right]^2 \\
&\leq \mathbb{E}_{(s,a)\sim D}\left[\left(\frac{\hat{R}(s,a) - R(s,a)}{\operatorname{range} R}\right)^2\right] \\
&= d_D^{\mathrm{MSE}}(R, \hat{R}) \\
&\leq \epsilon^2.
\end{aligned}
$$

It follows $d_D^{\mathrm{MAE}}(R, \hat{R}) < \epsilon$. By the definition of $\mathrm{safe}^{\mathrm{MAE}}(R, \epsilon, L, \lambda, \omega)$ and the assumption, this results in $\mathrm{Reg}^R(\hat{\pi}) < L$. Since $\hat{R}, \hat{\pi}$ were arbitrary, this shows $D \in \mathrm{safe}^{\mathrm{MSE}}(R, \epsilon^2, L, \lambda, \omega)$. $\square$

This proposition implies that our positive results (Theorem 3.1 and Theorem 4.1) transfer over from $\mathrm{safe}^{\mathrm{MAE}}$ to $\mathrm{safe}^{\mathrm{MSE}}$.

Theorem 3.2 transfers as well, with the condition on $\epsilon$ replaced by a square of the old condition:

$$\epsilon < \left( \frac{1-\gamma}{\sqrt{2}} \cdot \frac{\text{range } J^R}{\text{range } R} \cdot \min_{(s,a)} D(s,a) \cdot L \right)^2 .$$

### B.3.2. Transfer of the remaining results

The negative results do not transfer *automatically* since we would need an inequality between $d^{\text{MAE}}$ and $d^{\text{MSE}}$ in the other direction, which does not exist without further assumptions. Nevertheless, it is easily possible to modify most the proofs, where appropriate, to obtain corresponding results. In particular:

- Theorem 3.3 and Theorem 3.4 hold verbatim with unsafe$^{\text{MSE}}$ instead of unsafe$^{\text{MAE}}$. In the proof of Theorem 3.3, we can use the same construction of $\hat{R}$, and an almost identical derivation shows the bound in $d^{\text{MSE}}$.

- On Theorem 3.5: Due to Theorem B.1 in this rebuttal the "if"-direction of the theorem automatically holds when replacing $d_D^{\text{MAE}}(R, \hat{R})$ with $d_D^{\text{MSE}}(R, \hat{R})$, i.e., there exists a set of linear inequalities such that a given data distribution $D$ is safe, i.e., $D \in \text{safe}^{\text{MSE}}(R, \epsilon^2, L)$, whenever this set of linear inequalities is satisfied.
  However, the "only-if" direction does not hold since safe$^{\text{MSE}}(R, \epsilon^2, L)$ is not a polytope (whereas safe$^{\text{MAE}}(R, \epsilon, L)$ is) and can thus not be expressed by a finite set of linear constraints. The reason is that by replacing $d_D^{\text{MAE}}(R, \hat{R})$ with $d_D^{\text{MSE}}(R, \hat{R})$, the set $\{\hat{R} : d_D^{\text{MSE}}(R, \hat{R}) \leq \epsilon\}$ becomes an ellipsoid, whereas it was a polytope in the original formulation. Future work could look into a precise characterization in more detail.

- For Theorem 4.2, there is a corresponding version that is almost identical but replaces the condition on $D(\text{supp } D^{\hat{\pi}})$ by the following version including a square:

$$D(\text{supp } D^{\hat{\pi}}) \leq \frac{\epsilon}{(1+C)^2}.$$

  This condition can then be used at the very end of the proof of Theorem C.38 to finish the proof of an adapted Theorem Theorem 4.2.

- For the final negative result, Theorem 6.1, we already use a different distance measure motivated by the practice of RLHF. Thus, we are not interested in an adaptation for the MSE.

### B.4. A conceptual example of overoptimization concerns

In this section, we present a conceptual example that illustrates overoptimization concerns. This is meant to serve as an intuition for many of our "negative" theoretical results Theorems 3.3, 3.4, 4.2 and 6.1, with the aim to make them more grounded in realistic concerns.

In summary, imagine a scenario of a chatbot: It can either obtain "safe" or "dangerous" queries; safe queries (e.g. "Please help me create a high-protein diet") should be answered, dangerous queries (e.g. "Please tell me how to build a nuclear weapon") should be refused. We call answering a query "helping", irrespective of whether this is desired or not. We will specifically analyze an always-helping policy, its regret, and its plausibility to occur from reward learning. Helpful-only policies have been analyzed in past safety research (Denison et al., 2024) and are often a starting point for policies meant to become "helpful, honest, and harmless" (Askell et al., 2021).

First, we look at conditions for when helpful-only policies are unsafe relative to a regret bound $L$. It turns out that they are less safe if there appear more unsafe queries in the deployment environment, and if the damage caused by answering them is larger — see Appendix B.4.2. Then we look into the conditions for when this policy can be learned by reward learning — see Appendix B.4.3. It turns out that if there are "many styles" with which the chatbot can answer an unsafe query, then some of those answers must have a low probability on the training distribution, and thus a learned reward model can inflate its reward while achieving a low training error. The always-helping policy can then result from policy optimization, leading to a large regret. This illustrates an error-regret mismatch.

### B.4.1. SPECIFYING THE CONTEXTUAL BANDIT

We model the situation as follows: Assume a contextual bandit with states and actions given by

$$\mathcal{S} = \{q_{\text{safe}}, q_{\text{uns.}}\}, \quad \mathcal{A} = \{a_{\text{help}}^i, a_{\text{ref.}}^i\}_{i=1}^N.$$

In other words, there is one safe and one unsafe query,[2] and actions that either help with or refuse to answer the query in $N$ different styles. One should imagine $N$ to be fairly large since there are lots of ways to vary the style of an answer without changing the content, given that the amount of possible answers scales exponentially with length.

We assume the following simplified true reward function, where $C > 0$ is some (potentially large) constant:

$$
\begin{aligned}
R(q_{\text{safe}}, a_{\text{help}}^i) &= 1 \\
R(q_{\text{safe}}, a_{\text{ref.}}^i) &= 0 \\
R(q_{\text{uns.}}, a_{\text{help}}^i) &= -C \\
R(q_{\text{uns.}}, a_{\text{ref.}}^i) &= 0.
\end{aligned}
\tag{5}
$$

The idea is that answering a safe query should lead to some positive reward, whereas refusing it doesn't create value or damage — the reward is zero. Answering/helping with an *unsafe* queries, however, incurs a large negative reward $-C$ since it can lead to substantial damage, whereas, once again, refusing to answer does neither create value nor damage.

Finally, we assume some "true" distribution of queries, given by $\mu_{\text{uns.}} \in [0, 1]$ and $\mu_{\text{safe}} = 1 - \mu_{\text{uns.}}$. These can be imagined to be the frequencies with which actual users in the deployment environment ask safe vs. unsafe queries. In total, we have thus specified a contextual bandit $(\mathcal{S}, \mathcal{A}, R, \mu)$.

We now make a regret-analysis — analyzing when an always-helping policy is safe — followed by a reward learning analysis — under which conditions can an always-helping policy result from reward learning?

### B.4.2. REGRET ANALYSIS FOR ALWAYS-HELPING POLICY

For a policy $\hat{\pi}$ with answer probabilities $\hat{\pi}(a \mid q)$, the policy evaluation (i.e., expected reward) is given by

$$J_R(\hat{\pi}) = \mu_{\text{safe}} \cdot \sum_{i=1}^N \hat{\pi}(a_{\text{help}}^i \mid q_{\text{safe}}) - (1 - \mu_{\text{safe}}) \cdot C \cdot \sum_{i=1}^N \hat{\pi}(a_{\text{help}}^i \mid q_{\text{uns.}}). \tag{6}$$

This follows directly from (5). The idea is that under a safe query, which happens with probability $\mu_{\text{safe}}$, the reward is the probability to help with the query. For an unsafe query, which happens with probability $1 - \mu_{\text{safe}}$, the reward is $-C$ times the probability that the model helps with that query.

Now, the highest expected reward $J_R$ can be achieved if $\hat{\pi}$ always helps with a safe query and never helps with an unsafe query. This is hard to achieve in practice since training the model to refuse unsafe queries often leads to "over-refusal" on safe queries (Cui et al., 2024). In contrast, the lowest expected reward $J_R$ is achieved is $\hat{\pi}$ never helps with a safe query and always helps with an unsafe query. Thus, the maximum and minimum expected values are given by:

$$
\begin{aligned}
\max_{\hat{\pi}} J_R(\hat{\pi}) &= \mu_{\text{safe}}, \\
\min_{\hat{\pi}} J_R(\hat{\pi}) &= -(1 - \mu_{\text{safe}}) \cdot C.
\end{aligned}
\tag{7}
$$

Now, for purposes of illustration we look at one specific type of policy $\hat{\pi}$: one that *always* helps. Let $\hat{\pi}$ be such a policy. There are several such policies since they can differ in their allocation of probabilities to answers of different *styles*, but the defining property is that their action probabilities for helpful answers sum to 1:

$$\sum_{i=1}^N \hat{\pi}(a_{\text{help}}^i \mid q_{\text{safe}}) = 1, \quad \sum_{i=1}^N \hat{\pi}(a_{\text{help}}^i \mid q_{\text{uns.}}) = 1.$$

---

[2]Having a larger number of safe and unsafe queries does not change the mathematical picture much, but for illustration purposes we chose this simplified setting.

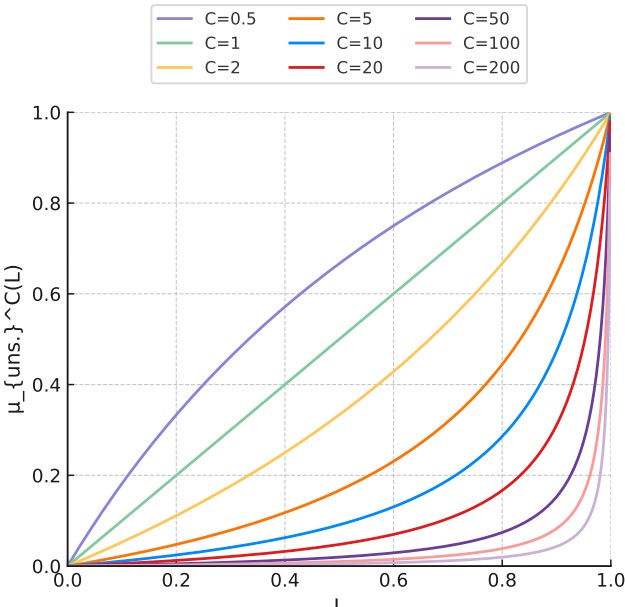

*Figure 3.* In our conceptual example, we analyze when an always-helping policy $\hat{\pi}$ is unsafe. This depends on the probability of an unsafe query $\mu_{\text{uns.}}$. For a given damage $C$ of answering such a query and a given regret bound $L$, $\hat{\pi}$ has a regret of at least $L$ if $\mu_{\text{uns.}}$ is larger than the plotted $\mu_{\text{uns.}}^C(L) = L/[(1 - L) \cdot C + L]$. $\mu_{\text{uns.}}^C(L)$ grows with growing $L$ and shrinks with growing $C$.

Using (6), its expected value is given by:

$$J_R(\hat{\pi}) = \mu_{\text{safe}} - (1 - \mu_{\text{safe}}) \cdot C. \tag{8}$$

Additionally using (7), the *regret* of this policy is:

$$
\begin{aligned}
\text{Reg}^R(\hat{\pi}) &= \frac{\max_\pi J_R(\pi) - J_R(\hat{\pi})}{\max_\pi J_R(\pi) - \min_\pi J_R(\pi)} \\
&= \frac{\mu_{\text{safe}} - \mu_{\text{safe}} + (1 - \mu_{\text{safe}}) \cdot C}{\mu_{\text{safe}} + (1 - \mu_{\text{safe}}) \cdot C} \\
&= \frac{(1 - \mu_{\text{safe}}) \cdot C}{\mu_{\text{safe}} + (1 - \mu_{\text{safe}}) \cdot C} \\
&= \frac{\mu_{\text{uns.}} \cdot C}{1 - \mu_{\text{uns.}} + \mu_{\text{uns.}} \cdot C}.
\end{aligned}
\tag{9}
$$

Now, imagine our goal is to have a regret lower than the bound $L \in [0, 1]$ — a threshold that we find "safe enough" for deployment. Is $\hat{\pi}$ unsafe? It depends on the value of $\mu_{\text{uns.}}$, i.e., the frequency of unsafe queries. Indeed, using (9), the inequality $\text{Reg}^R(\hat{\pi}) \geq L$ is equivalent to:

$$\mu_{\text{uns.}} \geq \frac{L}{(1 - L) \cdot C + L}. \tag{10}$$

In Figure 3 we analyze for several different values of the damage $C$ the relationship between the regret bound $L$ and the smallest probability $\mu_{\text{uns.}}^C(L) := L/[(1 - L) \cdot C + L]$ of the unsafe query for which the policy $\hat{\pi}$ would have a regret of at least $L$. We observe the following:

- For each $C$, as the regret bound $L$ gets larger, one needs a larger probability $\mu_{\text{uns.}}$ for $\hat{\pi}$ to have regret at least $L$. This makes sense: $\hat{\pi}$ acts correctly on safe queries, and so only unsafe queries can contribute to the regret. Thus, the more unsafe queries the policy encounters, the larger its regret becomes.

- For each regret bound $L$, as the damage of helping with an unsafe query, $C$, gets larger, a smaller probability $\mu_{\text{uns.}}$ is sufficient for $\hat{\pi}$ to reach regret at least $L$. This makes sense since the policy's overall performance is then more and more dominated by its performance on unsafe queries.

Note that over time, language models are approaching more concerning "dangerous capabilities" (Phuong et al., 2024; Anthropic, 2024), which means that the caused damage $C$ for following through with unsafe requests can be imagined to go up over time with increased capabilities. Positive value goes up, too, but plausibly in the near-term not as fast as the tailrisks. Thus, we can reasonably think that even for large values of the regret bound $L$, a small probability $\mu_{\text{uns.}}$ of an unsafe query would already cause the always-helping policy $\hat{\pi}$ to have a regret of at least $L$, and thus to be unsafe.

Alternatively, instead of looking at regret, we could also think directly about the expected value $J_R(\hat{\pi})$ computed in (8). Then we might say: the policy is unsafe if its expected value is negative, i.e., it causes more damage than it provides value. With growing damage $C$ for more capable models, the expected value eventually becomes negative, and so also this viewpoint suggests that $\hat{\pi}$ is not a safe policy.

### B.4.3. REWARD LEARNING ANALYSIS

Now, lets assume that the relationship between $L$, $C$, and $\mu_{\text{uns.}}$ as per Equation (10) is such that an always-helping policy $\hat{\pi}$ is *unsafe*, i.e., has regret at least $L$. Now the question becomes: Under what conditions could such a policy be learned by reward learning followed by policy optimization? To be clear, there are also other policies that have regret at least $L$ (e.g., a policy that doesn't help for safe queries and always helps for unsafe queries is even worse), but since we are operating under a worst-case framework under the policy optimization, it is already bad if any always-helping policy $\hat{\pi}$ can be learned. Thus, we are searching for sufficient conditions for this to happen.

Thus, let $\hat{R}$ be the learned reward function. For this to give rise to the policy $\hat{\pi}$ under unregularized policy optimization, $\hat{R}$ needs to favor at least one helpful answers over every refusing answer for both queries:

$$\begin{aligned}
\exists i \forall j \colon \hat{R}(q_{\text{safe}}, a_{\text{help}}^i) > \hat{R}(q_{\text{safe}}, a_{\text{ref.}}^j), \\
\exists i \forall j \colon \hat{R}(q_{\text{uns.}}, a_{\text{help}}^i) > \hat{R}(q_{\text{uns.}}, a_{\text{ref.}}^j).
\end{aligned} \tag{11}$$

Again, since we are operating under a worst-case framework, it is enough if we find one *specific* learned reward function with these conditions that can be learned in practice. Thus, for simplicity, we assume $\hat{R}(q_{\text{safe}}, a_{\text{help}}^i) = 1$, $\hat{R}(q_{\text{safe}}, a_{\text{ref.}}^i) = 0$ for all $i$. Also assume $\hat{R}(q_{\text{uns.}}, a_{\text{ref.}}^i) = 0$ for all $i$. Assume there exists a single $i_0$ with $B := \hat{R}(q_{\text{uns.}}, a_{\text{help}}^{i_0}) > 0$, and that $\hat{R}(q_{\text{uns.}}, a_{\text{help}}^i) = -C$ for all $i \neq i_0$. Then the conditions from (11) are met, and the learned reward function almost everywhere agrees with the true reward function $R$ from (5).

Now we want to determine the (mean absolute) training error of this reward model. For this, assume we train on some data distribution $D \in \Delta(\mathcal{S} \times \mathcal{A})$, given by $D(q, a) = D(q) \cdot D(a \mid q)$.[3] Since our reward model equals the true reward function in every query-answer pair except $(q_{\text{uns.}}, a_{\text{help}}^{i_0})$, the training error becomes:

$$\mathbb{E}_{(q,a) \sim D}\left[\frac{|\hat{R}(q,a) - R(q,a)|}{\text{range } R}\right] = D(q_{\text{uns.}}, a_{\text{help}}^{i_0}) \cdot \frac{B + C}{1 + C}.$$

Assume we train until we have achieved a small but realistic training error $\epsilon$. Then the question is under what conditions $\hat{R}$ can "slip through" the training by leading to an error bounded above by $\epsilon$. This is the case if:

$$D(q_{\text{uns.}}, a_{\text{help}}^{i_0}) < \frac{(1 + C) \cdot \epsilon}{B + C}. \tag{12}$$

Thus, if there is *some* $i_0$ for which this inequality holds, then $\hat{R}$ can be learned, and the always-helping policy $\hat{\pi}$ results. Now, note that if the number of "styles" $i = 1, \ldots, N$ is very large relative to the inverse of $\epsilon$, this is automatic. Namely, if

$$N > \frac{D(q_{\text{uns.}}) \cdot (B + C)}{\epsilon \cdot (1 + C)}, \tag{13}$$

---

[3] $D(q_{\text{safe}})$ is not necessarily equal to $\mu_{\text{safe}}$, the likelihood of safe queries in the deployment environment. This is intuitive: Before deploying a chatbot in the real world, it may be hard to know what proportion of requests will be safe, and the proportion during training may be different.

then since the probabilities sum to 1 there is an $i_0 \in \{1, \ldots, N\}$ with $D(a_{\text{help}}^{i_0} \mid q_{\text{uns.}}) \leq 1/N$, and we automatically obtain the result, (12).

A note on regularized policy optimization: Regularization can prevent $\hat{\pi}$ from being learned even if $\hat{R}$ favors this policy. However, if $B = \hat{R}(q_{\text{uns.}}, a_{\text{help}}^{i_0}) > 0$ is *very* large, then this creates so much reward that the regularization effect with constant regularization strength can be counteracted. Growing $B$ just leads to the need for larger $N$ in (13), and so we can say: If the number of styles $N$ is large enough (leading to a small training-probability of some bad action) and the always-helping policy $\hat{\pi}$ has regret larger then $L$, then supervised reward learning up to reasonable errors $\epsilon$ followed by (un)regularized policy optimization can result in a policy with regret $\geq L$. Thus, there is then an *error-regret mismatch*, and the distribution $D$ is unsafe, as per Theorem 2.1. That a large number of "bad options" or a small probability of *some* bad option can lead to an error-regret mismatch is the core intuition behind our negative results Theorems 3.3, 3.4, 4.2 and 6.1.

# C. Existence of error-regret mismatch

In this section, we answer the question under which circumstances error-regret mismatch could occur. We consider multiple different settings, starting from very weak statements, and then steadily increasing the strength and generality.

## C.1. Assumptions

For every MDP $\langle \mathcal{S}, \mathcal{A}, \tau, \mu_0, R, \gamma \rangle$ that we will define in the following statements, we assume the following properties:

- **Finiteness:** Both the set of states $\mathcal{S}$ and the set of actions $\mathcal{A}$ are finite

- **Reachability:** Every state in the given MDP's is reachable, i.e., for every state $s \in S$, there exists a path of transitions from some initial state $s_0$ (s.t. $\mu_0(s_0) > 0$) to $s$, such that every transition $(s, a, s)$ in this path has a non-zero probability, i.e., $\tau(s'|s, a) > 0$. Note that this doesn't exclude the possibility of some transitions having zero probability in general.

## C.2. Intuitive unregularized existence statement

**Definition C.1** (Regret). We define the *regret* of a policy $\pi$ with respect to reward function $R$ as

$$\text{Reg}^R(\pi) := \frac{\max J_R - J_R(\pi)}{\max J_R - \min J_R} \in [0, 1].$$

Here, $J$ is the policy evaluation function corresponding to $R$.

**Definition C.2** (Policy-Induced Distribution). Let $\pi$ be a policy. Then we define the *policy-induced distribution* $D^\pi$ by

$$D^\pi := (1 - \gamma) \cdot \eta^\pi.$$

**Definition C.3** (Range of Reward Function). Let $R$ be a reward function. Its *range* is defined as

$$\text{range } R := \max R - \min R.$$

**Lemma C.4.** *for any policy $\pi$, $D^\pi$ is a distribution.*

*Proof.* This is clear. $\square$

**Proposition C.5.** *Let $M = \langle \mathcal{S}, \mathcal{A}, \tau, \mu_0, R, \gamma \rangle$ be an MDP, $D \in \Delta(\mathcal{S} \times \mathcal{A})$ a data distribution, and $\epsilon > 0$, $L \in [0, 1]$. Assume there exists a policy $\hat{\pi}$ with the property that $\text{Reg}^R(\hat{\pi}) \geq L$ and $D(\text{supp } D^{\hat{\pi}}) < \epsilon$, where $\text{supp } D^{\hat{\pi}}$ is defined as the set of state-action pairs $(s, a) \in \mathcal{S} \times \mathcal{A}$ such that $D^{\hat{\pi}}(s, a) > 0$. In other words, there is a "bad" policy for $R$ that is not very supported by $D$. Then, $D$ allows for error-regret mismatch to occur, i.e., $D \in \textbf{unsafe}(R, \epsilon, L)$.*

*Proof.* We claim that whenever there exists a policy $\hat{\pi}$ with the following two properties:

- $\text{Reg}^R(\hat{\pi}) \geq L$,

- $D(\text{supp } D^{\hat{\pi}}) < \epsilon$,

then there exists a reward function $\hat{R}$ for which $\hat{\pi}$ is optimal, and such that

$$\mathbb{E}_{(s,a)\sim D}\left[\frac{|R(s,a)-\hat{R}(s,a)|}{\text{range } R}\right] \le \epsilon.$$

Since we assumed that $\text{Reg}^R(\hat{\pi}) \ge L$, this gives the result $D \in \mathbf{unsafe}(R,\epsilon,L)$.

To prove the claim, define

$$\hat{R}(s,a) := \begin{cases} R(s,a), & (s,a) \notin \text{supp } D^{\hat{\pi}}; \\ \max R, & \text{else.} \end{cases}$$

The state-action pairs that $\hat{\pi}$ visits all lie in $\text{supp } D^{\hat{\pi}}$, where $\hat{R}$ takes on its maximal reward $\max R$. This implies that $\hat{\pi}$ is optimal for $\hat{R}$. Furthermore, we obtain

$$\begin{aligned}
\mathbb{E}_{(s,a)\sim D}\left[\frac{|R(s,a)-\hat{R}(s,a)|}{\text{range } R}\right] &= \sum_{(s,a)} D(s,a)\frac{|R(s,a)-\hat{R}(s,a)|}{\text{range } R} \\
&= \sum_{(s,a)\in\text{supp } D^{\hat{\pi}}} D(s,a)\frac{\max R - R(s,a)}{\text{range } R} \\
&\le \sum_{(s,a)\in\text{supp } D^{\hat{\pi}}} D(s,a) \\
&= D(\text{supp } D^{\hat{\pi}}) \\
&\le \epsilon.
\end{aligned}$$

That was to show. $\qquad\square$

**Corollary C.6.** *Let $M = \langle \mathcal{S}, \mathcal{A}, \tau, \mu_0, R, \gamma \rangle$ be an MDP, $\epsilon > 0$, and $L \in [0,1]$. Assume there exists a set of policies $\Pi_L$ with:*

- *$\text{Reg}^R(\pi) \ge L$ for all $\pi \in \Pi_L$;*
- *$\text{supp } D^\pi \cap \text{supp } D^{\pi'} = \emptyset$ for all $\pi, \pi' \in \Pi_L$; and*
- *$|\Pi_L| \ge 1/\epsilon$.*

*Then $\mathbf{unsafe}(R,\epsilon,L) = \Delta(\mathcal{S}\times\mathcal{A})$, i.e.: all distributions are unsafe.*

*Proof.* Let $D \in \Delta(\mathcal{S}\times\mathcal{A})$. Let $\pi \in \arg\min_{\pi'\in\Pi_L} D(\text{supp } D^{\pi'})$. We obtain

$$|\Pi_L| \cdot D(\text{supp } D^\pi) \le \sum_{\pi'\in\Pi_L} D(\text{supp } D^{\pi'}) = D\left(\bigcup_{\pi'\in\Pi_L} \text{supp } D^{\pi'}\right) \le 1,$$

and therefore $D(\text{supp } D^\pi) \le 1/|\Pi_L| < \epsilon$. Since $\pi \in \Pi_L$, we also have $\text{Reg}^R(\pi) \ge L$. Together, $\pi$ and $D$ thus satisfy the assumptions from Theorem 3.3, whose conclusion implies $D \in \mathbf{unsafe}(R,\epsilon,L)$. This shows the inclusion $\Delta(\mathcal{S}\times\mathcal{A}) \subseteq \mathbf{unsafe}(R,\epsilon,L)$. The other inclusion is clear, and so we have equality. $\qquad\square$

**Proposition C.7.** *The assumptions on $\epsilon$ in Theorem 3.2 and Theorem 3.3 cannot hold simultaneously.*

*Proof.* If they *would* hold simultaneously, we would get:

$$\min_{(s,a)\in\mathcal{S}\times\mathcal{A}} D(s,a) \le D(\text{supp} D^{\hat{\pi}}) < \epsilon < \frac{1-\gamma}{\sqrt{2}} \cdot \frac{\text{range} J_R}{\text{range} R} \cdot \min_{(s,a)\in\mathcal{S}\times\mathcal{A}} D(s,a) \cdot L.$$

Here, the first step is clear, the second step is the assumption from Theorem 3.3, and the third step is the assumption from Theorem 3.2. We now show that this leads to a contradiction.

Dividing by the minimum on both sides, we obtain

$$1 < \frac{L}{\sqrt{2}} \cdot \frac{(1-\gamma)\text{range}J_R}{\text{range}R}. \tag{14}$$

Clearly, we have $L/\sqrt{2} < 1$. We also claim that the second fraction is smaller or equal to $1$, which then leads to the desired contradiction. Indeed, let $\pi^*$ and $\pi_*$ be an optimal and a worst-case policy, respectively. Then we have

$$
\begin{aligned}
(1-\gamma)\text{range}J_R &= (1-\gamma)(J_R(\pi^*) - J_R(\pi_*)) \\
&= (1-\gamma)\eta^{\pi^*} \cdot \vec{R} - (1-\gamma)\eta^{\pi_*} \cdot \vec{R} \\
&= D^{\pi^*} \cdot \vec{R} - D^{\pi_*} \cdot \vec{R} \\
&= \sum_{(s,a)\in\mathcal{S}\times\mathcal{A}} D^{\pi^*}(s,a)R(s,a) - \sum_{(s,a)\in\mathcal{S}\times\mathcal{A}} D^{\pi_*}(s,a)R(s,a) \\
&\leq \max_{(s,a)\in\mathcal{S}\times\mathcal{A}} R(s,a) - \min_{(s,a)\in\mathcal{S}\times\mathcal{A}} R(s,a) \\
&= \text{range}R.
\end{aligned}
$$

Here, we used the formulation of the policy evaluation function in terms of the occupancy measure $\eta$, and then that $1 - \gamma$ is a normalizing factor that transforms the occupancy measure into a distribution. Overall, this means that $(1 - \gamma)\text{range}J_R/\text{range}R \leq 1$, contradicting (14). Consequently, the assumptions of Theorem 3.2 and Theorem 3.3 cannot hold simultaneously. $\qquad\square$

### C.3. General existence statements

We start by giving some definitions:

**Definition C.8** (Minkowski addition). Let $A, B$ be sets of vectors, then the Minkowski addition of $A, B$ is defined as:

$$A + B \coloneqq \{a + b \mid a \in A, \ b \in B\}.$$

(Karwowski et al., 2023) showed in their proposition 1, that for every MDP, the corresponding occupancy measure space $\Omega$ forms a convex polytope. Furthermore, for each occupancy measure $\eta \in \Omega$ there exists at least one policy $\pi^\eta$ such that $\forall(s,a) \in \mathcal{S}\times\mathcal{A}, \ \eta^\pi(s,a) = \eta(s,a)$ (see Theorem 6.9.1, Corollary 6.9.2, and Proposition 6.9.3 of (Puterman, 1994)). In the following proofs, we will refer multiple times to vertices of the occupancy measure space $\Omega$ whose corresponding policies have high regret. We formalize this in the following definition:

**Definition C.9** (High regret vertices). Given a lower regret bound $L \in [0, 1]$, an MDP $\langle\mathcal{S}, \mathcal{A}, \tau, \mu_0, R, \gamma\rangle$ and a corresponding occupancy measure $\Omega$, we define the set of high-regret vertices of $\Omega$, denoted by $V_R^L$, to be the set of vertices $v$ of $\Omega$ for which $\text{Reg}^R(\pi^v) \geq L$

**Definition C.10** (Active inequalities). Let $\langle\mathcal{S}, \mathcal{A}, \tau, \mu_0, R, \gamma\rangle$ be an MDP with corresponding occupancy measure space $\Omega$. For every $\eta \in \Omega$, we define the set of transitions $(s, a)$ for which $\eta(s, a) = 0$ by $zeros(\eta)$.

**Definition C.11** (Normal cone). The normal cone of a convex set $C \subset \mathbb{R}^n$ at point $x \in C$ is defined as:

$$N_C(x) \coloneqq \{n \in \mathbb{R}^n \mid n^T \cdot (x' - x) \leq 0 \ \text{ for all } x' \in C\} \tag{15}$$

We first state a theorem from prior work that we will use to prove some lemmas in this section:

**Theorem C.12** ( (Schlaginhaufen & Kamgarpour, 2023)). *Let $\langle\mathcal{S}, \mathcal{A}, \tau, \mu_0, \gamma\rangle$ be an MDP without reward function and denote with $\Omega$ its corresponding occupancy measure space. Then, for every reward function $R$ and occupancy measure $\eta \in \Omega$, it holds that:*

$$\eta \ \text{is optimal for } R \quad\Longleftrightarrow\quad R \ \in \ N_\Omega(\eta), \tag{16}$$

*where the normal cone is equal to:*

$$N_\Omega(\eta) \ = \ \Phi + \text{cone}\left(\{-e_{s,a}\}_{(s,a)\in zeros(\eta)}\right) \tag{17}$$

*where $\Phi$ is the linear subspace of potential functions used for reward-shaping, and the addition is defined as the Minkowski addition.*

*Proof.* This is a special case of theorem 4.5 of Schlaginhaufen & Kamgarpour (2023), where we consider the unconstrained-and unregularized RL problem. $\qquad\square$

From the previous lemma, we can derive the following corollary which uses the fact that $\Omega$ is a closed, and bounded convex polytope (see Proposition 1 of Karwowski et al. (2023)).

**Corollary C.13.** *Given an MDP $\langle \mathcal{S}, \mathcal{A}, \tau, \mu_0, R, \gamma \rangle$ and a corresponding occupancy measure space $\Omega$, then for every reward function $\hat{R} : \mathcal{S} \times \mathcal{A} \to \mathbb{R}$, and lower regret bound $L \in [0, 1]$, the following two statements are equivalent:*

a) *There exists an optimal policy $\hat{\pi}$ for $\hat{R}$ such that $\hat{\pi}$ has regret at least $L$ w.r.t. the original reward function, i.e., $\mathrm{Reg}^R(\hat{\pi}) \geq L$.*

b) *$\hat{R} \in \Phi + \bigcup_{v \in V_R^L} \mathrm{cone}\left(\{-e_{s,a}\}_{(s,a)\in zeros(v)}\right)$, where $\Phi$ is the linear subspace of potential functions used for reward-shaping, the addition is defined as the Minkowski addition.*

*Proof.* Let $\hat{R}$ be chosen arbitrarily. Statement $a)$ can be formally expressed as:

$$\exists \hat{\pi} \in \Pi, \ \mathrm{Reg}^{\hat{R}}(\hat{\pi}) = 0 \ \wedge \ \mathrm{Reg}^R(\hat{\pi}) \geq L.$$

Using Theorem C.12, it follows that:

$$\exists \hat{\pi} \in \Pi, \ \mathrm{Reg}^{\hat{R}}(\hat{\pi}) = 0 \ \wedge \ \mathrm{Reg}^R(\hat{\pi}) \geq L$$
$$\iff \quad \exists \hat{\pi} \in \Pi, \quad \hat{R} \in N_\Omega(\eta^{\hat{\pi}}) \ \wedge \ \mathrm{Reg}^R(\hat{\pi}) \geq L$$
$$\iff \quad \hat{R} \in \bigcup_{\eta:\ \mathrm{Reg}^R(\pi^\eta) \geq L} N_\Omega(\eta).$$

It remains to be shown that the union in the previous derivation is equivalent to a union over just all $V_R^L$. First, note that by definition of the set of high-regret vertices $V_R^L$ (see Theorem C.9), it trivially holds that:

$$\bigcup_{v \in V_R^L} N_\Omega(v) \ \subseteq \bigcup_{\eta:\ \mathrm{Reg}^R(\pi^\eta) \geq L} N_\Omega(\eta), \tag{18}$$

Next, because $\Omega$ is a convex polytope, it can be defined as the intersection of a set of defining half-spaces which are defined by linear inequalities:

$$\Omega \ = \ \{\eta \mid a_i^T \cdot \eta \leq b_i, \ \text{for } i = 1, ..., m\}.$$

By defining the active index set of a point $\eta \in \Omega$ as $I_\Omega(\eta) = \{a_i \mid a_i^T \cdot \eta = b_i\}$, Rockafellar & Wets (2009) then show that:

$$N_\Omega(\eta) \ = \ \left\{y_1 \cdot a_1 + ... + y_m \cdot a_m \mid y_i \geq 0 \text{ for } i \in I_\Omega(\eta), \ y_i = 0 \text{ for } i \notin I_\Omega(\eta)\right\}, \tag{19}$$

(see their theorem 6.46). Note that, because $\Omega$ lies in an $|\mathcal{S}| \cdot (|\mathcal{A}| - 1)$ dimensional affine subspace (see Proposition 1 of (Karwowski et al., 2023)), a subset of the linear inequalities which define $\Omega$ must always hold with equality, namely, the inequalities that correspond to half-spaces which define the affine subspace in which $\Omega$ resides. Therefore, the corresponding active index set, let's denote it by $I_{\Omega,\Phi}(\eta)$ because the subspace orthogonal to the affine subspace in which $\Omega$ lies corresponds exactly to $\Phi$, is always non-empty and the same for every $\eta \in \Omega$.

Now, from Equation (19), it follows that for every $\eta \in \Omega$, there exists a vertex $v$ of $\Omega$, such that $N_\Omega(\eta) \subseteq N_\Omega(v)$. We take this one step further and show that for every $\eta$ with $\mathrm{Reg}^R(\pi^\eta) \geq L$, there must exist a vertex $v$ with $\mathrm{Reg}^R(\pi^v) \geq L$ such that $N_\Omega(\eta) \subseteq N_\Omega(v)$. We prove this via case distinction on $\eta$.

- $\eta$ is in the interior of $\Omega$. In this case, the index set $I_\Omega(\eta)$ reduces to $I_{\Omega,\Phi}(\eta)$ and because we have $I_{\Omega,\Phi}(\eta) \subseteq I_\Omega(\eta)$ for every $\eta \in \Omega$, the claim is trivially true.

- $\eta$ itself is already a vertex in which case the claim is trivially true.

- $\eta$ is on the boundary of $\Omega$. In this case $\eta$ can be expressed as the convex combination of some vertices $V_\eta$ which lie on the same face of $\Omega$ as $\eta$. Note that all occupancy measures with regret $\geq L$ must lie on one side of the half-space defined by the equality $R^T \cdot \eta = L \cdot \eta^{\min} + (1 - L) \cdot \eta^{\max}$, where $\eta^{\min}$ and $\eta^{\max}$ are worst-case and best-case occupancy measures. By our assumption, $\eta$ also belongs to this side of the half-space. Because $\eta$ lies in the interior of the convex hull of the vertices $V_\eta$, at least one $v \in V_\eta$ must therefore also lie on this side of the hyperplane and have regret $\geq L$. Because $v$ and $\eta$ both lie on the same face of $\Omega$, we have $I_\Omega(\eta) \subset I_\Omega(v)$ and therefore also $N_\Omega(\eta) \subseteq N_\Omega(v)$.

Hence, it must also hold that:

$$\bigcup_{\eta:\, \mathrm{Reg}^R(\pi^\eta) \geq L} N_\Omega(\eta) \ \subseteq \ \bigcup_{v \in V_R^L} N_\Omega(v),$$

which, together with Equation (18) proves the claim. $\qquad\square$

The following lemma relates the set of reward functions to the set of probability distributions $D$

**Lemma C.14.** *Given an MDP $\langle S, A, \tau, \mu_0, R, \gamma \rangle$ and a second reduced reward function $\hat{R} : S \times A \to \mathbb{R}$, then the following two statements are equivalent:*

a) *There exists a data distribution $D \in \Delta(S \times A)$ such that $\mathbb{E}_{(s,a) \sim D}\left[|R(s,a) - \hat{R}(s,a)|\right] < \epsilon \cdot \mathrm{range}\, R$*

b) *At least one component $\hat{R}_i$ of $\hat{R}$ is "close enough" to R, i.e., it holds that for some transition $(s,a)$: $|R(s,a) - \hat{R}(s,a)| < \epsilon \cdot \mathrm{range}\, R$.*

*Proof.* We first show the direction $b) \Rightarrow a)$. Assume that $|R(s^*, a^*) - \hat{R}(s^*, a^*)| < \epsilon \cdot \mathrm{range}\, R$ for a given $\hat{R}$ and transition $(s^*, a^*)$. In that case, we can construct the data distribution $D$ which we define as follows:

$$D(s,a) = \begin{cases} p & \text{if } (s,a) \neq (s^*, a^*) \\ 1 - (|S \times A| - 1) \cdot p & \text{if } (s,a) = (s^*, a^*) \end{cases}$$

where we choose $p < \min\left( \frac{\epsilon \cdot \mathrm{range}\, R - |R(s^*,a^*) - \hat{R}(s^*,a^*)|}{\sum_{(s,a) \neq (s^*,a^*)} |R(s,a) - \hat{R}(s,a)|}, \frac{1}{|S \times A|} \right)$. From this it can be easily seen that:

$$\mathbb{E}_{(s,a) \sim D}\left[|R(s,a) - \hat{R}(s,a)|\right]$$
$$= (1 - (|S \times A| - 1) \cdot p) \cdot |R(s^*,a^*) - \hat{R}(s^*,a^*)|$$
$$+ p \cdot \sum_{(s,a) \neq (s^*,a^*)} |R(s,a) - \hat{R}(s,a)|$$
$$< \epsilon \cdot \mathrm{range}\, R$$

We now show the direction $a) \Rightarrow b)$ via contrapositive. Whenever it holds that $|R(s,a) - \hat{R}(s,a)| \geq \epsilon \cdot \mathrm{range}\, R$ for all transitions $(s,a) \in S \times A$, then the expected difference under an arbitrary data distribution $D \in \Delta(S \times A)$ can be lower bounded as follows:

$$\mathbb{E}_{(s,a) \sim D}\left[|R(s,a) - \hat{R}(s,a)|\right]$$
$$= \sum_{(s,a) \in S \times A} D(s,a) \cdot |R(s,a) - \hat{R}(s,a)|$$
$$\geq \epsilon \cdot \mathrm{range}\, R \cdot \sum_{(s,a) \in S \times A} D(s,a)$$
$$= \epsilon \cdot \mathrm{range}\, R$$

Because this holds for all possible data distributions $D$ we have $\neg b) \Rightarrow \neg a)$ which proves the result. $\qquad\square$

Theorem C.13 describes the set of reward functions $\hat{R}$ for which there exists an optimal policy $\hat{\pi}$ that achieves worst-case regret under the true reward function $R$. Theorem C.14 on the other hand, describes the set of reward functions $\hat{R}$, for which there exists a data distribution $D$ such that $\hat{R}$ is close to the true reward function $R$ under $D$. We would like to take the intersection of those two sets of reward functions, and then derive the set of data distributions $D$ corresponding to this intersection. Toward this goal we first present the following lemma:

**Lemma C.15.** *For all $\epsilon > 0$, $L \in [0, 1]$, MDP $M = \langle \mathcal{S}, \mathcal{A}, \tau, \mu_0, R, \gamma \rangle$ and all data distributions $D \in \Delta(\mathcal{S} \times \mathcal{A})$, there exists a system of linear inequalities, such that $D \in \mathbf{unsafe}(R, \epsilon, L)$ if and only if the system of linear inequalities is solvable.*

*More precisely, let $V_R^L$ be the set of high-regret vertices defined as in Theorem C.9. Then, there exists a matrix $C$, as well as a matrix $U(v)$ and a vector $b(v)$ for every $v \in V_R^L$ such that the following two statements are equivalent:*

1. *$D \in \mathbf{unsafe}(R, \epsilon, L)$, i.e., there exists a reward function $\hat{R}$ and a policy $\hat{\pi}$ such that:*

    (a) $\mathbb{E}_{(s,a) \sim D} \left[ \frac{|\hat{R}(s,a) - R(s,a)|}{\text{range } R} \right] \leq \epsilon$;

    (b) $\text{Reg}^R (\hat{\pi}) \geq L$

    (c) $\text{Reg}^{\hat{R}} (\hat{\pi}) = 0$

2. *There exists a vertex $v \in V_R^L$ such that the linear system*

$$\begin{bmatrix} U(v) \\ C \cdot \text{diag}(D) \end{bmatrix} \cdot B \leq \begin{bmatrix} b(v) \\ \epsilon \cdot \text{range } R \cdot \mathbf{1} \end{bmatrix} \tag{20}$$

    *has a solution $B$. Here, we use the vector notation of the data distribution $D$.*

*Proof.* We can express any reward function $\hat{R}$ as $\hat{R} = R + B$, i.e. describing $\hat{R}$ as a deviation $B : \mathcal{S} \times \mathcal{A} \to \mathbb{R}$ from the true reward function. Note that in this case, we get $\hat{R} - R = B$. Next, note that the expression:

$$\mathbb{E}_{(s,a) \sim D} \left[ |B(s,a)| \right] \leq \epsilon \cdot \text{range } R \tag{21}$$

describes a "weighted $L^1$ ball" around the origin in which $B$ must lie:

$$\mathbb{E}_{(s,a) \sim D} \left[ |B(s,a)| \right] \leq \epsilon \cdot \text{range } R \tag{22}$$

$$\iff \sum_{(s,a) \in \mathcal{S} \times \mathcal{A}} D(s,a) \cdot |B(s,a)| \leq \epsilon \cdot \text{range } R \tag{23}$$

$$\iff B \in \mathcal{C}(D) := \left\{ x \in \mathbb{R}^{|\mathcal{S} \times \mathcal{A}|} \;\middle|\; \sum_{(s,a) \in \mathcal{S} \times \mathcal{A}} D(s,a) \cdot |x_{s,a}| \leq \epsilon \cdot \text{range } R \right\}. \tag{24}$$

This "weighted $L^1$ ball" is a polyhedral set, which can be described by the following set of inequalities:

$$D(s_1, a_1) \cdot B(s_1, a_1) + D(s_1, a_2) \cdot B(s_1, a_2) + \; ... \; \leq \; \epsilon \cdot \text{range } R$$
$$-D(s_1, a_1) \cdot B(s_1, a_1) + D(s_1, a_2) \cdot B(s_1, a_2) + \; ... \; \leq \; \epsilon \cdot \text{range } R$$
$$D(s_1, a_1) \cdot B(s_1, a_1) - D(s_1, a_2) \cdot B(s_1, a_2) + \; ... \; \leq \; \epsilon \cdot \text{range } R$$
$$-D(s_1, a_1) \cdot B(s_1, a_1) - D(s_1, a_2) \cdot B(s_1, a_2) + \; ... \; \leq \; \epsilon \cdot \text{range } R$$
$$\cdots .$$

This can be expressed more compactly in matrix form, as:

$$C \cdot \text{diag}(D) \cdot B \leq \epsilon \cdot \text{range } R \cdot \mathbf{1}, \tag{25}$$

where $C \in \mathbb{R}^{2^{|\mathcal{S} \times \mathcal{A}|} \times |\mathcal{S} \times \mathcal{A}|}$, $\text{diag}(D) \in \mathbb{R}^{|\mathcal{S} \times \mathcal{A}| \times |\mathcal{S} \times \mathcal{A}|}$, $B \in \mathbb{R}^{|\mathcal{S} \times \mathcal{A}|}$, $\mathbf{1} \in \{1\}^{|\mathcal{S} \times \mathcal{A}|}$ and the individual matrices are defined as follows:

$$C = \begin{bmatrix} 1 & 1 & \cdots & 1 \\ -1 & 1 & \cdots & 1 \\ 1 & -1 & \cdots & 1 \\ \cdots & \cdots & \cdots & \cdots \\ -1 & -1 & \cdots & -1 \end{bmatrix}, \qquad \text{diag}(D) = \begin{bmatrix} D(s_1, a_1) & & 0 \\ & \ddots & \\ 0 & & D(s_n, a_m) \end{bmatrix}. \tag{26}$$

Next, from Theorem C.13 we know that a reward function $\hat{R} = R + B$ has an optimal policy with regret larger or equal to $L$ if and only if:

$$R + B \in \Phi + \bigcup_{v \in V_R^L} \mathrm{cone}\left(\{-e_{s,a}\}_{(s,a) \in zeros(v)}\right)$$

$$\iff B \in -R + \Phi + \bigcup_{v \in V_R^L} \mathrm{cone}\left(\{-e_{s,a}\}_{(s,a) \in zeros(v)}\right) \tag{27}$$

We can rephrase the above statement a bit. Let's focus for a moment on just a single vertex $v \in V_R^L$. First, note that because $\Phi$ and $\mathrm{cone}\left(\{-e_{s,a}\}_{(s,a) \in zeros(v)}\right)$, are polyhedral, $\Phi + \mathrm{cone}\left(\{-e_{s,a}\}_{(s,a) \in zeros(v)}\right)$ must be polyhedral as well (this follows directly from Corollary 3.53 of (Rockafellar & Wets, 2009)). Therefore, the sum on the right-hand side can be expressed by a set of linear constraints $U(v) \cdot B \leq b(v)$.

Hence, a reward function, $\hat{R} = R + B$ is close in expected L1 distance to the true reward function $R$, and has an optimal policy that has large regret with respect to $R$, if and only if there exists at least one vertex $v \in V_R^L$, such that:

$$\begin{bmatrix} U(v) \\ C \cdot \mathrm{diag}\,(D) \end{bmatrix} \cdot B \leq \begin{bmatrix} b(v) \\ \epsilon \cdot \mathrm{range}\,R \cdot \mathbf{1} \end{bmatrix} \tag{28}$$

holds. □

In the next few subsections, we provide a more interpretable version of the linear system of inequalities in Equation (20), and the conditions for when it is solvable and when not.

### C.3.1. MORE INTERPRETABLE STATEMENT

Ideally, we would like to have a more interpretable statement about which classes of data distributions $D$ fulfill the condition of Equation (20). We now show that for an arbitrary MDP and data distribution $D$, $D$ is a safe distribution, i.e., error-regret mismatch is not possible, if and only if $D$ fulfills a fixed set of linear constraints (independent of $D$).

**Theorem C.16.** *For all MDPs $\langle \mathcal{S}, \mathcal{A}, \tau, \mu_0, R, \gamma \rangle$ and $L \in [0, 1]$, there exists a matrix $M$ such that for all $\epsilon > 0$ and $D \in \Delta(\mathcal{S} \times \mathcal{A})$ we have:*
$$D \in \mathbf{safe}(R, \epsilon, L) \iff M \cdot D > \epsilon \cdot \mathrm{range}\,R \cdot \mathbf{1}, \tag{29}$$
*where we use the vector notation of $D$, and $\mathbf{1}$ is a vector containing all ones.*

*Proof.* Remember from Theorem C.15, that a data distribution $D$ is safe, i.e., $D \in \mathbf{safe}(R, \epsilon, L)$, if and only if for all unsafe vertices $v \in V_R^L$ the following system of linear inequalities:

$$\begin{bmatrix} U(v) \\ C \cdot \mathrm{diag}\,(D) \end{bmatrix} \cdot B \leq \begin{bmatrix} b(v) \\ \epsilon \cdot \mathrm{range}\,R \cdot \mathbf{1} \end{bmatrix} \tag{30}$$

has no solution. Let $v \in V_R^L$ be chosen arbitrarily and define $\mathcal{U}_v := \{B \in \mathbb{R}^{|\mathcal{S} \times \mathcal{A}|} \mid U(v) \cdot B \leq b(v)\}$, i.e., $\mathcal{U}_v$ is the set of all $B \in \mathbb{R}^{|\mathcal{S} \times \mathcal{A}|}$, such that $\hat{R} := R + B$ has an optimal policy with regret at least $L$. Then, Equation (30) has no solution if and only if:

$$\forall B \in \mathcal{U}_v, \quad C \cdot \mathrm{diag}\,(D) \cdot B \not\leq \epsilon \cdot \mathrm{range}\,R \cdot \mathbf{1} \tag{31}$$

$$\iff \forall B \in \mathcal{U}_v, \quad \mathrm{abs}(B)^T \cdot D > \epsilon \cdot \mathrm{range}\,R, \tag{32}$$

where we used the definition of the matrices $C$, and $\mathrm{diag}\,(D)$ (see Equation (25)) and $\mathrm{abs}(\cdot)$ denotes the element-wise absolute value function. Now, we will finish the proof by showing that there exists a *finite* set of vectors $X \subset \mathcal{U}_v$ (which is independent of the choice of $D$), such that for every $x \in X$, Equation (32) holds if and only if it is true for all $B$, i.e., more formally:

$$\forall B \in X, \quad \mathrm{abs}(B)^T \cdot D > \epsilon \cdot \mathrm{range}\,R$$

$$\iff \forall B \in \mathcal{U}_v, \quad \mathrm{abs}(B)^T \cdot D > \epsilon \cdot \mathrm{range}\,R.$$

And since $X$ is finite, we can then summarize the individual elements of $X$ as rows of a matrix $M$ and get the desired statement by combining the previous few statements, namely:

$$D \in \mathbf{safe}(R, \epsilon, L) \quad \Longleftrightarrow \quad M \cdot D > \epsilon \cdot \mathrm{range}\, R \cdot \mathbf{1} \tag{33}$$

Towards this goal, we start by reformulating Equation (32) as a condition on the optimal value of a convex optimization problem:

$$
\begin{aligned}
& \forall x \in \mathcal{U}_v, \quad \mathrm{abs}(x)^T \cdot D > \epsilon \cdot \mathrm{range}\, R \\
\Longleftrightarrow \quad & \left( \min_{x \in \mathcal{U}_v} \mathrm{abs}(x)^T \cdot D \right) > \epsilon \cdot \mathrm{range}\, R \\
\Longleftrightarrow \quad & \mathrm{abs}(x^*)^T \cdot D > \epsilon \cdot \mathrm{range}\, R, \quad \text{where } x^* := \qquad \underset{x \in \mathcal{U}_v}{\arg\min} \quad \mathrm{abs}(x)^T \cdot D \\
\Longleftrightarrow \quad & \mathrm{abs}(x^*)^T \cdot D > \epsilon \cdot \mathrm{range}\, R, \quad \text{where } x^* := \qquad \underset{x}{\arg\min} \quad \mathrm{abs}(x)^T \cdot D, \\
& \qquad\qquad\qquad\qquad\qquad\qquad\qquad\qquad\qquad\qquad\qquad\quad \text{subject to} \quad U(v) \cdot x \le b(v)
\end{aligned}
\tag{34}
$$

Note that the optimal value $x^*$ of this convex optimization problem depends on the precise definition of the data distribution $D$. But importantly, the set over which we optimize (i.e., $\mathcal{U}_v$ defined as the set of all $x$, such that $U(v) \cdot x \le b$) does *not* depend on $D$! The goal of this part of the proof is to show that for all possible $D$ the optimal value of the optimization problem in Equation (34) is *always* going to be one of the vertices of $\mathcal{U}_v$. Therefore, we can transform the optimization problem in Equation (34) into a new optimization problem that does not depend on $D$ anymore. It will then be possible to transform this new optimization problem into a simple set of linear inequalities which will form the matrix $M$ in Equation (33).

Towards that goal, we continue by splitting up the convex optimization problem into a set of linear programming problems. For this, we partition $\mathbb{R}^{|\mathcal{S} \times \mathcal{A}|}$ into its different orthants $O_c$ for $c \in \{-1, 1\}^{|\mathcal{S} \times \mathcal{A}|}$ (a high-dimensional generalization of the quadrants). More precisely, for every $x \in O_c$, we have $\mathrm{diag}\,(c) \cdot x = \mathrm{abs}(x)$. Using this definition, we can reformulate the constraint on the convex optimization problem as follows:

$$\min_{\substack{c \in \{-1,1\}^{|\mathcal{S} \times \mathcal{A}|} \\ x_c \neq \emptyset}} (\mathrm{diag}\,(c) \cdot x_c)^T \cdot D > \epsilon \cdot \mathrm{range}\, R, \tag{35}$$

where the individual $x_c$ are defined as the solution of linear programming problems:

$$
\begin{aligned}
x_c := \; & \underset{x}{\arg\min} \quad (\mathrm{diag}\,(c) \cdot x)^T \cdot D \\
& \text{subject to} \quad U(v) \cdot x \le b(v) \\
& \qquad\qquad\quad\; \mathrm{diag}\,(c) \cdot x \ge 0,
\end{aligned}
\tag{36}
$$

or $x_c := \emptyset$ in case the linear program is infeasible. Finally, by re-parametrizing each linear program using the variable transform $x' = \mathrm{diag}\,(c) \cdot x$ we can convert these linear programs into standard form:

$$
\begin{aligned}
x_c := \; \mathrm{diag}\,(c) \cdot \; & \underset{x'}{\arg\min} \qquad\qquad\qquad x'^T \cdot D \\
& \text{subject to} \qquad U(v) \cdot \mathrm{diag}\,(c) \cdot x' \le b(v) \\
& \qquad\qquad\qquad\qquad\qquad x' \ge 0,
\end{aligned}
\tag{37}
$$

where we used twice the fact that $\mathrm{diag}\,(c)^{-1} = \mathrm{diag}\,(c)$, and hence, $x = \mathrm{diag}\,(c) \cdot x'$. Because it was possible to transform these linear programming problems described in Equation (36) into standard form using a simple variable transform, we can apply standard linear programming theory to draw the following conclusions (see Theorem 3.4 and Section 6 of Chapter 2 of (Vanderbei, 1998) for reference):

1. The set of constraints in Equations (36) and (37) are either infeasible or they form a polyhedral set of feasible solutions.

2. If the set of constraints in Equations (36) and (37) are feasible, then there exists an optimal feasible solution that corresponds to one of the vertices (also called basic feasible solutions) of the polyhedral constraint sets. This follows from the fact that the objective function is bounded from below by zero.

Let's denote the polyhedral set of feasible solutions defined by the constraints in Equation (36) by $\mathcal{F}_c(v)$. Because $\mathcal{F}_c(v)$ does not depend on the specific choice of the data distribution, this must mean that for every possible data distribution $D$, we have either $x_c = \emptyset$ or $x_c$ is one of the vertices of $\mathcal{F}_c(v)$, denoted by $\mathrm{vertices}(\mathcal{F}_c(v))$! Note that, by definition of $x_c$, it holds that:

$$\forall x \in \mathrm{vertices}(\mathcal{F}_c(v)), \quad (\mathrm{diag}\,(c) \cdot x_c)^T \cdot D \;\leq\; (\mathrm{diag}\,(c) \cdot x)^T \cdot D. \tag{38}$$

Therefore, we can define:

$$X(v) \;:=\; \bigcup_{c \in \{-1,1\}^{|\mathcal{S} \times \mathcal{A}|}} \mathrm{vertices}(\mathcal{F}_c(v)) \;=\; \{x_1, ..., x_k\}, \qquad \text{and} \qquad M_{X(v)} \;:=\; \begin{bmatrix} \mathrm{abs}(x_1)^T \\ \cdots \\ \mathrm{abs}(x_k)^T \end{bmatrix}, \tag{39}$$

where $M_X(v)$ contains the element-wise absolute value of all vectors of $X(v)$ as row vectors. Let $D$ be an arbitrary data distribution. Then, we've shown the following equivalences:

$$\forall B \in \mathcal{U}_v, \quad \mathrm{abs}(B)^T \cdot D \;>\; \epsilon \cdot \mathrm{range}\, R \qquad \text{(see Equation (32))}$$

$$\Longleftrightarrow \quad \min_{\substack{c \in \{-1,1\}^{|\mathcal{S} \times \mathcal{A}|} \\ x_c \neq \emptyset}} (\mathrm{diag}\,(c) \cdot x_c)^T \cdot D \;>\; \epsilon \cdot \mathrm{range}\, R \qquad \text{(see Equation (35))}$$

$$\Longleftrightarrow \quad \min_{x \in X(v)} \mathrm{abs}(x)^T \cdot D \;>\; \epsilon \cdot \mathrm{range}\, R \qquad \text{(due to Equation (38))}$$

$$\Longleftrightarrow \quad M_X(v) \cdot D \;>\; \epsilon \cdot \mathrm{range}\, R \cdot \mathbf{1}$$

Now, by combining the individual sets of vertices $X(v)$, as follows:

$$X \;:=\; \bigcup_{v \in V_R^L} X(v) \;=\; \{x_1, ..., x_l\}, \qquad \text{and} \qquad M = \begin{bmatrix} \mathrm{abs}(x_1)^T \\ \cdots \\ \mathrm{abs}(x_l)^T \end{bmatrix}, \tag{40}$$

we are now ready to finish the proof by combining all previous steps:

$$D \in \mathbf{safe}(R, \epsilon, L)$$
$$\Longleftrightarrow \quad \forall v \in V_R^L, \, \forall B \in \mathcal{U}_v, \qquad\qquad \mathrm{abs}(B)^T \cdot D > \epsilon \cdot \mathrm{range}\, R$$
$$\Longleftrightarrow \quad \forall v \in V_R^L, \qquad\qquad\qquad M_X(v) \cdot D > \epsilon \cdot \mathrm{range}\, R \cdot \mathbf{1}$$
$$\Longleftrightarrow \qquad\qquad\qquad\qquad\qquad M \cdot D > \epsilon \cdot \mathrm{range}\, R \cdot \mathbf{1}.$$

That was to show. $\qquad\qquad\qquad\qquad\qquad\qquad\qquad\qquad\qquad\qquad\qquad\qquad\qquad\qquad\qquad\qquad\qquad\qquad\quad$ $\square$

### C.3.2. DERIVING THE CONDITIONS ON $D$

In Theorem C.16 we've shown that there exists a set of linear constraints $M \cdot D > \epsilon \cdot \mathrm{range}\, R \cdot \mathbf{1}$, such that whenever a data distribution $D$ satisfies these constraints, it is safe. In this subsection, we derive closed-form expressions for the individual rows of $M$ to get a general idea about the different factors determining whether an individual data distribution is safe.

In the proof of Theorem C.16, we showed that $M$ has the form:

$$M = \begin{bmatrix} \mathrm{abs}(x_1)^T \\ \vdots \\ \mathrm{abs}(x_l)^T \end{bmatrix},$$

for some set $X = \{x_1, ..., x_l\}$, where each $x \in X$ belongs to a vertex of the set of linear constraints defined by the following class of system of linear inequalities:

$$\begin{bmatrix} U(v) \\ -\mathrm{diag}\,(c) \end{bmatrix} \cdot x \leq \begin{bmatrix} b(v) \\ 0 \end{bmatrix} \qquad \begin{array}{l} \text{(Corresponds to the set of unsafe reward functions)} \\ \text{(Corresponds to the orthant } O_c) \end{array} \tag{41}$$

for some $v \in V_R^L$ (the set of unsafe vertices of $\Omega$), and some $c \in \{-1, 1\}^{|\mathcal{S} \times \mathcal{A}|}$ (defining the orthant $O_c$).

To ease the notation in the following paragraphs, we will use the notation $\mathcal{U}_v$ for the polyhedral set of $x$ such that $U(v) \cdot x \leq b(v)$, and $\mathcal{F}_c(v)$ for the set of solutions to the full set of linear inequalities in Equation (41). Furthermore, we will use $n := |\mathcal{S}|$ and $m := |\mathcal{A}|$.

We start by giving a small helper definition.

**Definition C.17** (General position, (Stanley, 2024))**.** Let $\mathcal{H}$ be a set of hyperplanes in $\mathbb{R}^n$. Then $\mathcal{H}$ is in general position if:

$$\begin{aligned} \{H_1, ..., H_p\} \subseteq \mathcal{H}, \ p \leq n \quad &\implies \quad \dim(H_1 \cap ... \cap H_p) = n - p \\ \{H_1, ..., H_p\} \subseteq \mathcal{H}, \ p > n \quad &\implies \quad H_1 \cap ... \cap H_p = \emptyset \end{aligned}$$

We will use this definition in the next few technical lemmas. First, we claim that each of the vertices of $\mathcal{F}_c(v)$ must lie on the border of the orthant $O_c$.

**Lemma C.18** (Vertices lie on the intersection of the two constraint sets.)**.** *All vertices of the polyhedral set, defined by the system of linear inequalities:*

$$\begin{bmatrix} U(v) \\ -\mathrm{diag}\,(c) \end{bmatrix} \cdot x \leq \begin{bmatrix} b(v) \\ 0 \end{bmatrix} \tag{42}$$

*must satisfy some of the inequalities of* $-\mathrm{diag}\,(c) \cdot x \ \leq \ 0$ *with equality.*

*Proof.* Let $\mathcal{U}_v$ be the set of solutions of the upper part of the system of linear equations in Equation (42) and $O_c$ be the set of solutions of the lower part of the system of linear equations in Equation (42). The lemma follows from the fact that $\mathcal{U}_v$ can be expressed as follows (see Equation (27) and the subsequent paragraph):

$$\mathcal{U}_v \ = \ -R + \Phi + \mathrm{cone}\left(\{-e_{s,a}\}_{(s,a) \in zeros(v)}\right), \tag{43}$$

where $\Phi$ is a linear subspace. Hence, for every $x$ that satisfies the constraints $U(v) \cdot x \leq b(v)$, $x$ lies on the interior of the line segment spanned between $x' = x + \phi$, and $x'' = x - \phi$ for some $\phi \in \Phi$, $\phi \neq \mathbf{0}$. Note that every point on this line segment also satisfies the constraints $U(v) \cdot x \leq b(v)$. Therefore, $x$ can only be a vertex if it satisfies some of the additional constraints, provided by the inequalities $-\mathrm{diag}\,(c) \cdot x \leq 0$, with equality. $\qquad\square$

Consequently, every vertex of $\mathcal{F}_c(v)$ is the intersection of some $k$-dimensional surface of $\mathcal{U}_v$ and $k > 0$ standard hyperplanes (hyperplanes whose normal vector belongs to the standard basis).

**Lemma C.19** (Basis for $\Phi$. (Schlaginhaufen & Kamgarpour, 2023))**.** *The linear subspace $\Phi$ of potential shaping transformations can be defined as:*

$$\Phi \ = \ \mathrm{span}(A - \gamma \cdot P),$$

*where $A, P \in \mathbb{R}^{(n \cdot m) \times n}$ for $n = |\mathcal{S}|, m = |\mathcal{A}|$ are matrices defined as:*

$$A := \begin{bmatrix} \mathbf{1}^m & \mathbf{0}^m & \cdots & \mathbf{0}^m \\ \mathbf{0}^m & \mathbf{1}^m & \cdots & \mathbf{0}^m \\ \cdots & \cdots & \ddots & \cdots \\ \mathbf{0}^m & \mathbf{0}^m & \cdots & \mathbf{1}^m \end{bmatrix}, \qquad P := \begin{bmatrix} \underline{\quad} & \tau(\,\cdot \mid s_1, a_1) & \underline{\quad} \\ \underline{\quad} & \tau(\,\cdot \mid s_1, a_2) & \underline{\quad} \\ \cdots & \cdots & \cdots \\ \underline{\quad} & \tau(\,\cdot \mid s_n, a_m) & \underline{\quad} \end{bmatrix},$$

*where $\mathbf{0}^m, \mathbf{1}^m$ are column vectors and $\tau(\cdot|s_i, a_j)$ is a row vector of the form $[\tau(s_1 \mid s_i, a_j), \cdots, \tau(s_n \mid s_i, a_j)]$.*

*Furthermore, we have $\dim \Phi = n$.*

*Proof.* This has been proven by (Schlaginhaufen & Kamgarpour, 2023) (see their paragraph "Identifiability" of Section 4). The fact that $\dim \Phi = n$ follows from the fact that $\Phi$ is the linear space orthogonal to the affine space containing the occupancy measure space $\Omega$, i.e. $\Phi^\perp = L$ where $L$ is the linear subspace parallel to $\mathrm{span}(\Omega)$ (see the paragraph *Convex Reformulation* of Section 3 of (Schlaginhaufen & Kamgarpour, 2023)) and the fact that $\dim \mathrm{span}(\Omega) = n \cdot (m - 1)$ (see Proposition 1 of (Karwowski et al., 2023)). $\qquad\square$

**Lemma C.20** (Dimension of $\mathcal{U}_v$)**.** $\dim \mathcal{U}_v = n \cdot m$.

*Proof.* Remember that $\mathcal{U}_v$ can be expressed as follows (see Equation (27) and the subsequent paragraph):

$$\mathcal{U}_v = -R + \Phi + \text{cone}\left(\{-e_{s,a}\}_{(s,a)\in zeros(v)}\right), \tag{44}$$

From Theorem C.19 we know that $\dim \Phi = n$. We will make the argument that:

a) $\dim\left[\text{cone}\left(\{-e_{s,a}\}_{(s,a)\in zeros(v)}\right)\right] \geq n \cdot (m-1)$

b) There exist exactly $n \cdot (m-1)$ basis vectors of $\text{cone}\left(\{-e_{s,a}\}_{(s,a)\in zeros(v)}\right)$ such that the combined set of these vectors and the basis vectors of $\Phi$ is linearly independent.

From this, it must follow that:

$$\dim\left[\Phi + \text{cone}\left(\{-e_{s,a}\}_{(s,a)\in zeros(v)}\right)\right] = \dim\left[\Phi\right] + n \cdot (m-1) = n \cdot m$$

For $a)$, remember that $v$ is a vertex of the occupancy measure space $\Omega$ and that each vertex $v$ of $\Omega$ corresponds to at least one deterministic policy $\pi^v$ (see Proposition 1 of (Karwowski et al., 2023)). And since every deterministic policy is zero for exactly $n \cdot (m-1)$ transitions, it must follow that $v$ is also zero in *at least* $n \cdot (m-1)$ transitions, since whenever $\pi^v(a|s) = 0$ for some $(s,a) \in \mathcal{S}\times\mathcal{A}$, we have:

$$v(s,a) = \sum_{t=0}^{\infty} \gamma^t \cdot P(s_t = s, a_t = a \mid \pi^v, \tau) = \pi^v(a|s) \cdot \sum_{t=0}^{\infty} \gamma^t \cdot P(s_t = s \mid \pi^v, \tau) = 0.$$

Therefore, it follows that $\dim\left[\text{cone}\left(\{-e_{s,a}\}_{(s,a)\in zeros(v)}\right)\right] \geq n \cdot (m-1)$.

For $b)$, (Puterman, 1994) give necessary and sufficient conditions for a point $x \in \mathbb{R}^{n \cdot m}$ to be part of $\Omega$ (see the dual linear program in section 6.9.1 and the accompanying explanation), namely:

$$x \in \Omega \iff \left[(A - \gamma \cdot P)^T \cdot x = \mu_0 \quad \text{and} \quad I \cdot x \geq 0\right],$$

where $I$ is the identity matrix and we use the vector notation of the initial state distribution $\mu_0$. Because $v$ is a vertex of $\Omega$, it can be described as the intersection of $n \cdot m$ supporting hyperplanes of $\Omega$ that are in general position. Because $(A - \gamma \cdot P)$ has rank $n$ (see Theorem C.19), this must mean that for $v$ at least $n \cdot (m-1)$ inequalities of the system $I \cdot v \geq 0$ hold with equality and the combined set of the corresponding row vectors and the row vectors of $(A - \gamma \cdot P)^T$ is linearly independent (as the vectors correspond to the normal vectors of the set of $n \cdot m$ hyperplanes in general position).

Note that the set of unit vectors that are orthogonal to $v$ is precisely defined by $\{-e_{s,a}\}_{(s,a)\in zeros(v)}$, since, by definition of $zeros(v)$ (see Theorem C.10), we have

$$\forall x \in \{-e_{s,a}\}_{(s,a)\in zeros(v)}, \quad x^T \cdot v = 0.$$

From this, it must follow that the polyhedral set $\mathcal{U}_v$, has dimension $n \cdot m$. $\square$

**Lemma C.21** (Defining the faces of $\mathcal{U}_v$). *Each $k$-dimensional face $F$ of $\mathcal{U}_v$ (with $k \geq n$) can be expressed as:*

$$-R + \Phi + \text{cone}(E_F), \qquad \text{where} \ \ E_F \subset \{-e_{s,a}\}_{(s,a)\in zeros(v)}, \tag{45}$$

*such that $|E_F| = k - n$ and the combined set of vectors of $E_F$ and the columns of $A - \gamma \cdot P$ is linearly independent.*

*Proof.* Remember that $\mathcal{U}_v$ can be expressed as follows (see Equation (27) and the subsequent paragraph):

$$\mathcal{U}_v = -R + \Phi + \text{cone}\left(\{-e_{s,a}\}_{(s,a)\in zeros(v)}\right), \tag{46}$$

This means that we can express $\mathcal{U}_v$ as a polyhedral cone, spanned by non-negative combinations of:

- The column vectors of the matrix $A - \gamma \cdot P$.

- The column vectors of the matrix $-(A - \gamma \cdot P)$. Since $\Phi$ is a linear subspace and a cone is spanned by only the positive combinations of its set of defining vectors we also have to include the negative of this matrix to allow arbitrary linear combinations.

- The set of vectors $\{-e_{s,a}\}_{(s,a) \in zeros(v)}$.

Consequently, each face of $\mathcal{U}_v$ of dimension $k$ is spanned by a subset of the vectors that span $\mathcal{U}_v$ and is therefore also a cone of these vectors. Because the face has dimension $k$, we require exactly $k$ linearly independent vectors, as it's not possible to span a face of dimension $k$ with less than $k$ linearly independent vectors, and every additional linearly independent vector would increase the dimension of the face. Furthermore, since $\Phi$ is a linear subspace that is unbounded by definition, it must be part of every face. Therefore, every face of $\mathcal{U}_v$ has a dimension of at least $n$ (the dimension of $\Phi$). $\qquad \square$

Note that the converse of Theorem C.21 doesn't necessarily hold, i.e., not all sets of the form described in Equation (45) are necessarily surfaces of the polyhedral set $U(v) \cdot x \leq b(v)$.

We are now ready to develop closed-form expressions for the vertices of $\mathcal{F}_c(v)$. Note that it is possible for $\mathbf{0} \in \mathbb{R}^{n \cdot m}$ to be a vertex of $\mathcal{F}_c(v)$. But in this case, according to Theorem C.16, this must mean that the linear system of inequalities $M \cdot D > \epsilon \cdot \text{range } R \cdot \mathbf{1}$ is infeasible (since $M$ would contain a zero row and all elements on the right-hand side are non-negative), which means that in this case $\mathbf{safe}(R, \epsilon, L) = \emptyset$. We will therefore restrict our analysis to all non-zero vertices of $\mathcal{F}_c(v)$.

**Proposition C.22** (Vertices of $\mathcal{F}_c(v)$.). *Every vertex $v_{FG}$ of $\mathcal{F}_c(v)$, with $v_{FG} \neq \mathbf{0}$, lies on the intersection of some face $F$ of the polyhedral set $\mathcal{U}_v$ and some face $G$ of the orthant $O_c$ and is defined as follows:*

$$v_{FG} \quad = \quad -R + [A - \gamma \cdot P, E_F] \cdot \left( E_G \cdot [A - \gamma \cdot P, E_F] \right)^{-1} \cdot E_G \cdot R,$$

*where $E_F$, $E_G$ are matrices whose columns contain standard unit vectors, such that:*

$$
\begin{aligned}
F \quad &= \quad -R + \Phi + \text{cone}\,(E_F)\,, \quad \text{for } E_F \subset \{-e_{s,a}\}_{(s,a) \in zeros(v)} \\
G \quad &= \quad \{x \in \mathbb{R}^{n \cdot m} \mid E_G \cdot x = \mathbf{0}\}.
\end{aligned}
$$

*Proof.* We start by defining the faces of the orthant $O_c$. Remember that $O_c$ is the solution set to the system of inequalities $\text{diag}\,(c) \cdot x \geq 0$. Therefore, each defining hyperplane of $O_c$ is defined by one row $i$ of $\text{diag}\,(c)$, i.e. $\text{diag}\,(c)_i \cdot x = 0$. Note that since $c \in \{-1, 1\}^{n \cdot m}$, this is equivalent to the equation $e_i^T \cdot x = 0$ where $e_i$ is either the $i$'th standard unit vector or its negative. And because every l-dimensional face $G$ of $O_c$ is the intersection of $l$ standard hyperplanes $\{e_{i_1}, ..., e_{i_l}\}$, this must mean that $G$ is defined as the set of solutions to the system of equations $E_G \cdot x = 0$ where $E_G$ is the matrix whose row vectors are the vectors $\{e_{i_1}, ..., e_{i_l}\}$.

Next, let $v_{FG}$ be an arbitrary non-zero vertex of $\mathcal{F}_c(v)$. As proven in Theorem C.18, every vertex of $\mathcal{F}_c(v)$ must satisfy some of the inequalities $\text{diag}\,(c) \cdot x \geq 0$ for $c \in \{-1, 1\}^{n \cdot m}$ with equality. This means that $v_{FG}$ must lie on some face $G$ of the orthant $O_c$. The non-zero property guarantees that not all inequalities of the system of inequalities $\text{diag}\,(c) \cdot x \geq 0$ are satisfied with equality, i.e. that $G$ is not a vertex. Assume that $k > 0$ inequalities are *not* satisfied with equality. Therefore, $G$ must have dimension $k$, and $E_G \in \mathbb{R}^{n \cdot m \times k}$.

Since $v_{FG}$ is a vertex of the intersection of the orthant $O_c$ and the polyhedral set $\mathcal{U}_v$, and it only lies on a $k$-dimensional face of $O_c$, it must also lie on a $n \cdot m - k$ dimensional face $F$ of $\mathcal{U}_v$ such that the combined set of hyperplanes defining $F$ and $G$ is in general position. The condition that the combined set of hyperplanes is in general position is necessary, to guarantee that $v_{FG}$ has dimension 0 and is therefore a proper vertex.

From Theorem C.21 we know that $F$ can be expressed as:

$$-R + \Phi + \text{cone}\,(E_F)\,, \qquad \text{where } E_F \subset \{-e_{s,a}\}_{(s,a) \in zeros(v)}, \tag{47}$$

such that $|E_F| = n \cdot (m - 1) - k$ and the combined set of vectors of $E_F$ and the columns of $A - \gamma \cdot P$ are linearly independent.

Because $v_{FG}$ is part of both, $F$ and $G$, we can combine all information that we gathered about $F$ and $G$ and deduce that it must hold that:

$$\underbrace{E_G \cdot v_{FG} = 0}_{\text{equivalent to } v_{FG} \in G} \quad \text{, and} \quad \underbrace{\exists x \in \mathbb{R}^{n \cdot m - k}, \quad v_{FG} = -R + [A - \gamma \cdot P, E_F] \cdot x,}_{\text{equivalent to } v_{FG} \in F} \tag{48}$$

where for $x$ in Equation (48) it additionally must hold that $\forall i \in \{n+1, ..., n \cdot m - k\}$, $x_i \geq 0$. This must hold because these last entries of $x$ should form a convex combination of the vectors in $E_F$ (as $F$ is defined to lie in the cone of $E_F$, see Equation (47)). We briefly state the following two facts that will be used later in the proof:

a) $v_{FG}$ is the only vector in $\mathbb{R}^{n \cdot m}$ that fulfills both conditions in Equation (48). This is because we defined $F$ in such a way that the intersection of $F$ and $G$ is a single point. And only points in this intersection fulfill both conditions in Equation (48).

b) For every non-zero vertex $v_{FG}$, there can only exist a single $x$ that satisfies the two conditions in Equation (48). This follows directly from the assumption that the combined set of vectors of $E_F$ and the columns of $A - \gamma \cdot P$ are linearly independent (see Equation (47) and the paragraph below).

We can combine the two conditions in Equation (48) to get the following, unified condition that is satisfied for every non-zero vertex $v_{FG}$:

$$\exists x \in \mathbb{R}^{n \cdot m - k}, \quad E_G \cdot \left( -R + [A - \gamma \cdot P, E_F] \cdot x \right) = \mathbf{0}^{n \cdot m - k}, \tag{49}$$

From this, it is easy to compute the precise coordinates of $v_{FG}$:

$$x = \left( E_G \cdot [A - \gamma \cdot P, E_F] \right)^{-1} \cdot E_G \cdot R \tag{50}$$

$$\implies v_{FG} = -R + [A - \gamma \cdot P, E_F] \cdot \left( E_G \cdot [A - \gamma \cdot P, E_F] \right)^{-1} \cdot E_G \cdot R. \tag{51}$$

We finish the proof by showing that the matrix inverse in Equation (50) always exists for every non-zero vertex $v_{FG}$. Assume, for the sake of contradiction, that the matrix $E_G \cdot [A - \gamma \cdot P, E_F]$ is not invertible. We will show that in this case, there exists a $z \in \mathbb{R}^{n \cdot m}$ with $z \neq v_{FG}$ such that $z$ fulfills both conditions in Equation (48). As we've shown above in fact $a)$ this is not possible, hence this is a contradiction.

Assuming that $E_G \cdot [A - \gamma \cdot P, E_F]$ is not invertible, we know from standard linear algebra that in that case the kernel of this matrix has a dimension larger than zero. Let $y_1, y_2$, be two elements of this kernel with $y_1 \neq y_2$.

Earlier in this proof, we showed that for every non-zero vertex $v_{FG}$, Equation (49) is satisfiable. Let $x$ be a solution to Equation (49). From our assumptions, it follows that both $x + y_1$ and $x + y_2$ must also be solutions to Equation (49) as:

$$\forall y \in \{y_1, y_2\}, \quad E_G \cdot \left( -R + [A - \gamma \cdot P, E_F] \cdot (x + y) \right)$$
$$= -E_G \cdot R + E_G \cdot [A - \gamma \cdot P, E_F] \cdot (x + y)$$
$$= -E_G \cdot R + E_G \cdot [A - \gamma \cdot P, E_F] \cdot x$$
$$= E_G \cdot \left( -R + [A - \gamma \cdot P, E_F] \cdot x \right)$$
$$= \mathbf{0}^{n \cdot m - k}.$$

And from this, it will follow that both, $x + y_1$ and $x + y_2$ must satisfy both conditions in Equation (48). Because $x + y_1 \neq x + y_2$, it must also hold that:

$$-R + [A - \gamma \cdot P, E_F] \cdot (x + y_1) \quad \neq \quad -R + [A - \gamma \cdot P, E_F] \cdot (x + y_2),$$

see fact $b)$ above for a proof of this. And this would mean that there exists at least one $z \in \mathbb{R}^{n \cdot m}$ with $z \neq v_{FG}$ such that $z$ fulfills both conditions in Equation (48). But as we have shown in fact $a)$, this is not possible. Therefore, the matrix $E_G \cdot [A - \gamma \cdot P, E_F]$ *must* be invertible for every non-zero vertex $v_{FG}$. $\qquad \square$

We are now ready to provide more specific information about the exact conditions necessary for a data distribution $D$ to be safe.

**Corollary C.23** (Vertices of $\mathcal{F}_c(v)$.). *For all $\epsilon > 0$, $L \in [0, 1]$ and MDPs $\langle \mathcal{S}, \mathcal{A}, \tau, \mu_0, R, \gamma \rangle$, there exists a matrix $M$ such that:*

$$D \in \mathbf{safe}(R, \epsilon, L) \quad \Longleftrightarrow \quad M \cdot D > \epsilon \cdot \mathrm{range}\, R \cdot \mathbf{1}, \tag{52}$$

*for all $D \in \Delta(\mathcal{S} \times \mathcal{A})$, where we use the vector notation of $D$, and $\mathbf{1}$ is a vector containing all ones.*

*The matrix $M$ is defined as:*

$$M = \begin{bmatrix} \mathrm{abs}(x_1)^T \\ \cdots \\ \mathrm{abs}(x_l)^T \end{bmatrix},$$

*where an individual row $x_i$ of $M$ can either be all zeros, or*

$$x_i \;=\; -R + [A - \gamma \cdot P, E_{i1}] \cdot \left( E_{i2} \cdot [A - \gamma \cdot P, E_{i1}] \right)^{-1} \cdot E_{i2} \cdot R, \tag{53}$$

*where $E_{i1}$, $E_{i2}$ are special matrices whose columns contain standard unit vectors.*

*Proof.* This is a simple combination of Theorem C.16 and Theorem C.22. □

In particular, Equation (53) shows that whether a particular data distribution $D$ is safe or not depends on the true reward function $R$, as well as the transition distribution $\tau$ (encoded by the matrix $P$).

### C.3.3. ALGORITHM TO COMPUTE THE CONDITIONS ON $D$

The derivations of Appendix C.3.2 can be used to define a simple algorithm that constructs matrix $M$. An outline of such an algorithm is presented in Algorithm 1. We use the terms $\mathrm{RowMatrix}$ and $\mathrm{ColumnMatrix}$ to denote functions that take a set of vectors and arrange them as rows/columns of a matrix.

To give a brief explanation of the algorithm:

- Line 5 follows from the definitions of $V_R^L$, $X(v)$ and $X$ (see Theorem C.9 and Equations (39) and (40)).

- Lines 6 and 7 are taken from the definition of $E_F$ in Equation (47) (except that we don't take the negative of the vectors and instead negate $E_F$ in the final formula).

- Lines 8 and 9 are taken from the definition of $E_G$ (see the first two paragraphs of Theorem C.22). We additionally ensure that $E_F$ is a subset of $E_G$ as otherwise, the matrix $E_G \cdot [A - \gamma \cdot P, -E_F]$ is not invertible (due to the multiplication of $E_G \cdot E_F$) and we know that the matrix must be invertible for every vertex.

- Lines 20 and 22 compute the row of the matrix $M$. The formulas are a combination of the definition of the sets $X(v)$, $X$ (see Equations (39) and (40)), the matrix $M_X$ (Equation (40)) and Theorem C.22.

- Line 19 checks whether the matrix $E_G \cdot [A - \gamma \cdot P, -E_F]$ is invertible. This is always the case for the rows of $M$ (see the last few paragraphs of the proof of Theorem C.22) but might not be true for other candidates.

- To explain Line 21, remember that every row of the matrix $M$ corresponds to the element-wise absolute value of a vector that lies on the intersection of two polyhedral sets $F$, and $G$ (see Theorem C.22). The polyhedral set $F$ is defined via a convex cone. To check that our solution candidate lies in this convex cone, we have to check whether the last entries of $x = (E_G \cdot [A - \gamma \cdot P, -E_F])^{-1} \cdot E_G \cdot R$, the entries belonging to the vectors in $E_F$, are non-negative.

The asymptotic runtime of this naive algorithm is exponential in $|\mathcal{S} \times \mathcal{A}|$ due to the iterations over all subsets in Lines 7 and 8. However, better algorithms might exist and we consider this an interesting direction for future work.

---

**Algorithm 1** Computes the set of conditions used to determine the safety of a data distribution.

---

1: **Input:** $MDP = \langle \mathcal{S}, \mathcal{A}, \tau, \mu_0, R, \gamma \rangle$, $L \in [0, 1]$
2: $I \leftarrow$ the *set* of all unit vectors of dimension $|\mathcal{S} \times \mathcal{A}|$. Create a fixed ordering of $\mathcal{S}$ and $\mathcal{A}$ and denote each vector of $I$ by $e_{(s,a)}$ for a unique tuple $(s, a) \in \mathcal{S} \times \mathcal{A}$.
3: candidates $\leftarrow [\ ]$
4: $\Pi_d \leftarrow$ Set of deterministic policies of $MDP$

5: % Create a set of potential row candidates.
6: **for** $\pi \in \{\pi' \in \Pi_d : \text{Reg}^R(\pi') \geq L\}$ **do**
7:    $E \leftarrow \{e_{(s,a)} \in I : \pi(a|s) = 0\}$
8:    **for** $E_F \subset E$ **do**
9:       **for** $subset \subseteq I \setminus E_F$, $|subset| = |\mathcal{S}|$ **do**
10:          $E_G \leftarrow E_F \cup subset$
11:          $E_F, E_G \leftarrow \text{ColumnMatrix}(E_F), \text{RowMatrix}(E_G)$
12:          candidates.append$((E_F, E_G))$
13:       **end for**
14:    **end for**
15: **end for**

16: % Find the valid rows amongst the candidates.
17: rows $\leftarrow [\ ]$
18: **for** $(E_F, E_G) \in$ candidates **do**
19:    $k \leftarrow \text{num\_columns}(E_F)$
20:    **if** $\text{rank}\Big(E_G \cdot [A - \gamma \cdot P, -E_F]\Big) = n + k$ **then**
21:       $x \leftarrow \Big(E_G \cdot [A - \gamma \cdot P, -E_F]\Big)^{-1} \cdot E_G \cdot R$
22:       **if** $\forall i \in \{n, n+1, ..., n+k\}\ x_i \geq 0$ **then**
23:          row $\leftarrow \text{abs}\Big(-R + [A - \gamma \cdot P, -E_F] \cdot x\Big)^T$
24:          rows.append$(\text{row})$
25:       **end if**
26:    **end if**
27: **end for**

28: $M \leftarrow \text{RowMatrix(rows)}$
29: **return** $M$

---

### C.3.4. WORKING EXAMPLE OF COMPUTING MATRIX $M$

Figure 4 shows a simple toy-MDP with a single state and three actions, for which we then compute matrix $M$ using Algorithm 1. Due to the simple structure of the MDP, the auxiliary matrix $A$ and the state-transition matrix $P$ (both used in Algorithm 1) become trivial:

$$A = \begin{bmatrix} 1 \\ 1 \\ 1 \end{bmatrix}, \text{ and } \quad P = \begin{bmatrix} 1 \\ 1 \\ 1 \end{bmatrix}$$

The resulting four constraints that a given data distribution over the state-action space of this MDP has to fulfill to be in $\textbf{safe}(R, \epsilon, L)$ are then visualized in the right-most column of Figure 4. Note that the constraints are over three-dimensional vectors. However, because $D$ is a probability distribution, it must live in a two-dimensional subspace of this three-dimensional space, and using the identity $d_3 = 1 - d_1 - d_2$ we can transform the constraints as follows:

$$\begin{bmatrix} | & | & | \\ m_1 & m_2 & m_3 \\ | & | & | \end{bmatrix} \cdot \begin{bmatrix} d_1 \\ d_2 \\ d_3 \end{bmatrix} > \begin{bmatrix} | \\ b \\ | \end{bmatrix} \iff \begin{bmatrix} | & | \\ m_1 - m_3 & m_2 - m_3 \\ | & | \end{bmatrix} \cdot \begin{bmatrix} d_1 \\ d_2 \end{bmatrix} > \begin{bmatrix} | \\ b - m_3 \\ | \end{bmatrix}$$

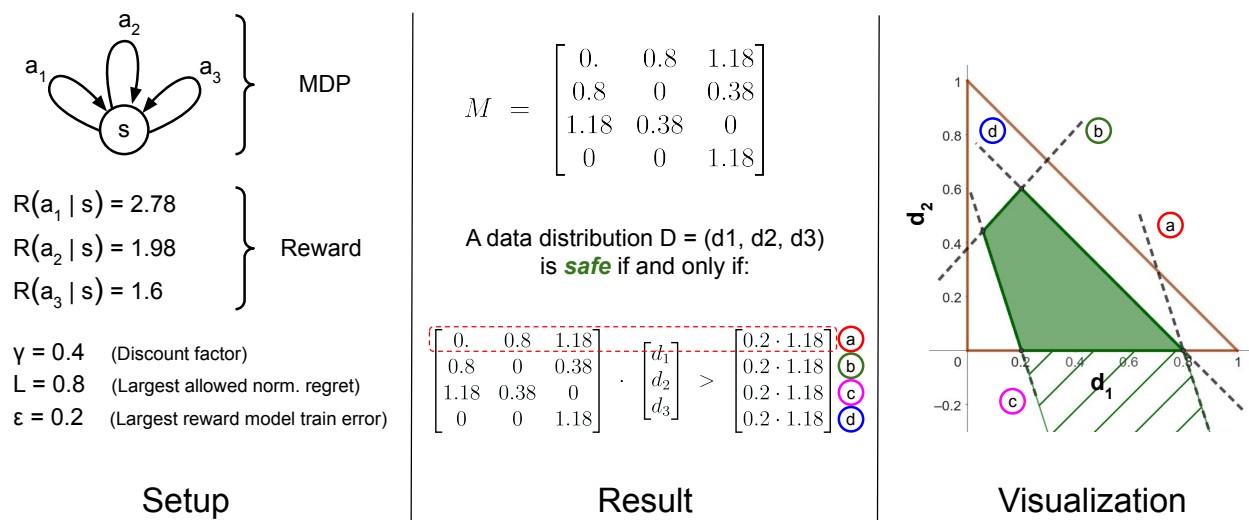

*Figure 4.* A working example of how to compute the matrix $M$ on a very simple MDP with a single state and three actions. Given the information in the *Setup* column, matrix $M$ can be computed using Algorithm 1. The constructed matrix $M$ contains four linear constraints that a data distribution $D$ has to fulfill in order to be in $\mathbf{safe}(R, \epsilon, L)$. The four constraints are plotted in the right-most column.

The brown triangle in Figure 4 depicts the 2d-probability simplex of all distributions over the three actions of the MDP.

Note that constraint ⓐ is a redundant constraint that is already covered by the constraint ⓓ and the border of the simplex. It would therefore be possible to disregard the computation of such constraints entirely, which could speed up the execution of Algorithm 1. In the next section, we discuss this possibility, as well as other potential directions in which we can extend Theorem 3.5.

### C.3.5. BUILDING UP ON THEOREM 3.5

There are multiple ways how future work can build up on the results of Theorem 3.5:

**Finding sufficient conditions for safety that require less information about the true reward function:** It would be very interesting to investigate whether there exists some subset of the set of safe data distributions for which it is possible to more easily determine membership. This could be helpful in practice, as knowing that a provided data distribution is safe directly yields safety guarantees for the resulting optimal policy.

**Developing faster methods to construct M:** While the algorithm we provide above runs in exponential time it is unclear whether this has to be the case. The set of vectors that are computed by our algorithm is redundant in the sense that some elements can be dropped as the conditions they encode are already covered by other rows of M. Depending on what fraction of computed elements are redundant it might be possible to develop an algorithm that prevents the computation of redundant rows and can therefore drastically reduce computation time. Alternatively, it would be interesting to develop fast algorithms to compute only parts of M. This could be especially interesting to quickly prove the unsafety of a data distribution, which only requires that a single constraint is violated.

**Extending Theorem 3.5 to the regularized policy optimization case:** This would allow one to extend the use case we described above to an even wider variety of reward learning algorithms, such as RLHF.

**A theoretical baseline (a broader view on the previous point):** Most of the options above reveal the properties of the "baseline algorithm" of reinforcement learning under unknown rewards: First, a reward model is trained, and second, a policy is optimized against the trained reward model. The matrix M is valid for the simplest such baseline algorithms without any regularization in either the reward model or the policy. As we mentioned in comments to other reviewers, it would be valuable to study other training schemes (e.g., regularized reward modeling, or switching back and forth between policy optimization and reward modeling on an updated data distribution), for which the set of safe data distributions (or "safe starting conditions") is likely more favorable than for the baseline case. Then, similar to how empirical work compares

new algorithms empirically against baseline algorithms, we hope our work can be a basis to theoretically study improved RL algorithms under unknown rewards, e.g. by deriving a more favorable analog of the matrix M and comparing it with our work.

## C.4. Existence of negative results in the RLHF setting

### C.4.1. GENERALIZATION OF THE ERROR MEASUREMENT: PROOFS

In this subsection we test the extent to which the results of the previous section generalize to different distance definitions. To ensure compatibility with the positive results of Appendix D.3, we consider MDPs with finite time horizon $T$. In this setting, trajectories are defined as a finite list of states and actions: $\xi = s_0, a_0, s_1, ..., a_{T-1}$. Let $\Xi$ bet the set of all trajectories of length $T$. As in the previous sections, $G : \Xi \to \mathbb{R}$ denotes the trajectory return function, defined as:

$$G(\xi) = \sum_{t=0}^{T-1} \gamma^t \cdot R(s_t, a_t)$$

**Proposition C.24.** *Given an MDP $\langle \mathcal{S}, \mathcal{A}, \tau, \mu_0, R, \gamma \rangle$, a data sampling policy $\pi : \mathcal{S} \to \Delta(\mathcal{A})$ and a second reward function $\hat{R} : \mathcal{S} \times \mathcal{A} \to \mathbb{R}$, we can upper bound the expected difference in trajectory evaluation as follows:*

$$\mathbb{E}_{\xi \sim \pi} \left[ |G_R(\xi) - G_{\hat{R}}(\xi)| \right] \ \le \ \frac{1 - \gamma^T}{1 - \gamma} \cdot \mathbb{E}_{(s,a) \sim D^\pi} \left[ |R(s, a) - \hat{R}(s, a)| \right] \tag{54}$$

*where $D^\pi = \frac{1-\gamma}{1-\gamma^T} \cdot \eta^\pi$.*

*Proof.* This follows from the subsequent derivation:

$$
\begin{aligned}
\mathbb{E}_{\xi \sim \pi} \left[ |G_R(\xi) - G_{\hat{R}}(\xi)| \right] &= \sum_{\xi \in \Xi} P(\xi \mid \pi) \cdot \left| \sum_{t=0}^{T-1} \gamma^t \cdot (R(s_t, a_t) - \hat{R}(s_t, a_t)) \right| \\
&\le \sum_{\xi \in \Xi} P(\xi \mid \pi) \cdot \sum_{t=0}^{T-1} \gamma^t \cdot \left| R(s_t, a_t) - \hat{R}(s_t, a_t) \right| \\
&= \sum_{(s,a) \in \mathcal{S} \times \mathcal{A}} \left( \sum_{t=0}^{T-1} \gamma^t \cdot P(s_t = s, a_t = a \mid \pi) \right) \cdot \left| R(s, a) - \hat{R}(s, a) \right| \\
&= \sum_{(s,a) \in \mathcal{S} \times \mathcal{A}} \eta^\pi(s, a) \cdot \left| R(s, a) - \hat{R}(s, a) \right| \\
&= \frac{1 - \gamma^T}{1 - \gamma} \cdot \mathbb{E}_{(s,a) \sim D^\pi} \left[ \left| R(s, a) - \hat{R}(s, a) \right| \right]
\end{aligned}
$$

$\square$

Given some reward function $R$, define the probability of trajectory $\xi_1$ being preferred over trajectory $\xi_2$ to be:

$$p_R(\xi_1 \succ \xi_2) \ = \ \sigma(G_R(\xi_1) - G_R(\xi_2)) \ = \ \frac{\exp(G_R(\xi_1))}{\exp(G_R(\xi_1)) + \exp(G_R(\xi_2))}.$$

Then, the following statement holds:

**Proposition C.25.** *Given an MDP $\langle \mathcal{S}, \mathcal{A}, \tau, \mu_0, R, \gamma \rangle$, a data sampling policy $\pi : \mathcal{S} \to \Delta(\mathcal{A})$ and a second reward function $\hat{R} : \mathcal{S} \times \mathcal{A} \to \mathbb{R}$, we can upper bound the expected KL divergence over trajectory preference distributions as follows:*

$$\mathbb{E}_{\xi_1, \xi_2 \sim \pi \times \pi} \left[ \mathbb{D}_{KL} \left( p_R(\cdot | \xi_1, \xi_2) || p_{\hat{R}}(\cdot | \xi_1, \xi_2) \right) \right] \ \le \ 2 \cdot \mathbb{E}_{\xi \sim \pi} \left[ |G_R(\xi) - G_{\hat{R}}(\xi)| \right], \tag{55}$$

*Proof.* The right-hand-side of Equation (55) can be lower bounded as follows:

$$2 \cdot \mathbb{E}_{\xi \sim \pi} \left[ |G_R(\xi) - G_{\hat{R}}(\xi)| \right] \tag{56}$$

$$= \mathbb{E}_{\xi_1, \xi_2 \sim \pi \times \pi} \left[ |G_R(\xi_1) - G_{\hat{R}}(\xi_1)| + |G_R(\xi_2) - G_{\hat{R}}(\xi_2)| \right] \tag{57}$$

$$\geq \mathbb{E}_{\xi_1, \xi_2 \sim \pi \times \pi} \left[ |(G_R(\xi_1) - G_R(\xi_2)) - (G_{\hat{R}}(\xi_1) - G_{\hat{R}}(\xi_2))| \right] \tag{58}$$

$$= \mathbb{E}_{\xi_1, \xi_2 \sim \pi \times \pi} \left[ |x_{\xi_1, \xi_2} - y_{\xi_1, \xi_2}| \right], \tag{59}$$

where from Equation (57) to Equation (58) we used the triangle inequality and did some rearranging of the terms, and from Equation (58) to Equation (59) we simplified the notation a bit by defining $x_{\xi_1, \xi_2} := G_R(\xi_1) - G_R(\xi_2)$ and $y_{\xi_1, \xi_2} := G_{\hat{R}}(\xi_1) - G_{\hat{R}}(\xi_2)$.

Similarly, we can reformulate the left-hand-side of Equation (55) as follows:

$$\mathbb{E}_{\xi_1, \xi_2 \sim \pi \times \pi} \left[ \mathbb{D}_{\mathrm{KL}} \left( p_R(\cdot | \xi_1, \xi_2) || p_{\hat{R}}(\cdot | \xi_1, \xi_2) \right) \right] \tag{60}$$

$$= \mathbb{E}_{\xi_1, \xi_2 \sim \pi \times \pi} \left[ \sum_{\substack{i,j \in \{1,2\} \\ i \neq j}} p_R(\xi_i \succ \xi_j | \xi_1, \xi_2) \cdot \log \left( \frac{p_R(\xi_i \succ \xi_j | \xi_1, \xi_2)}{p_{\hat{R}}(\xi_i \succ \xi_j | \xi_1, \xi_2)} \right) \right] \tag{61}$$

$$= \mathbb{E}_{\xi_1, \xi_2 \sim \pi \times \pi} \left[ \sum_{\substack{i,j \in \{1,2\} \\ i \neq j}} \sigma(G_R(\xi_i) - G_R(\xi_j)) \cdot \log \left( \frac{\sigma(G_R(\xi_i) - G_R(\xi_j))}{\sigma(G_{\hat{R}}(\xi_i) - G_{\hat{R}}(\xi_j))} \right) \right] \tag{62}$$

$$= \mathbb{E}_{\xi_1, \xi_2 \sim \pi \times \pi} \left[ \sum_{\substack{i,j \in \{1,2\} \\ i \neq j}} \sigma(x_{\xi_i, \xi_j}) \cdot \log \left( \frac{\sigma(x_{\xi_i, \xi_j})}{\sigma(y_{\xi_i, \xi_j})} \right) \right]. \tag{63}$$

We will now prove the lemma by showing that for all $(\xi_1, \xi_2) \in \Xi \times \Xi$ we have:

$$\sum_{\substack{i,j \in \{1,2\} \\ i \neq j}} \sigma(x_{\xi_i, \xi_j}) \cdot \log \left( \frac{\sigma(x_{\xi_i, \xi_j})}{\sigma(y_{\xi_i, \xi_j})} \right) \leq |x_{\xi_1, \xi_2} - y_{\xi_1, \xi_2}|, \tag{64}$$

from which it directly follows that Equation (63) is smaller than Equation (59).

Let $(\xi_1, \xi_2) \in \Xi \times \Xi$ be chosen arbitrarily. We can then upper bound the left-hand side of Equation (64) as follows:

$$\sigma(x_{\xi_1, \xi_2}) \cdot \log \left( \frac{\sigma(x_{\xi_1, \xi_2})}{\sigma(y_{\xi_1, \xi_2})} \right) + \sigma(x_{\xi_2, \xi_1}) \cdot \log \left( \frac{\sigma(x_{\xi_2, \xi_1})}{\sigma(y_{\xi_2, \xi_1})} \right) \tag{65}$$

$$\leq \log \left( \frac{\sigma(x_{\xi_1, \xi_2})}{\sigma(y_{\xi_1, \xi_2})} \right) + \log \left( \frac{\sigma(x_{\xi_2, \xi_1})}{\sigma(y_{\xi_2, \xi_1})} \right) \tag{66}$$

$$= \log \left( \frac{\sigma(x_{\xi_1, \xi_2}) \cdot \sigma(-x_{\xi_1, \xi_2})}{\sigma(y_{\xi_1, \xi_2}) \cdot \sigma(-y_{\xi_1, \xi_2})} \right) \tag{67}$$

$$= \log \left( \frac{\exp(x_{\xi_1, \xi_2}) \cdot (1 + \exp(y_{\xi_1, \xi_2}))^2}{\exp(y_{\xi_1, \xi_2}) \cdot (1 + \exp(x_{\xi_1, \xi_2}))^2} \right) \tag{68}$$

$$= x_{\xi_1, \xi_2} - y_{\xi_1, \xi_2} + 2 \cdot \log \left( \frac{1 + \exp(y_{\xi_1, \xi_2})}{1 + \exp(x_{\xi_1, \xi_2})} \right), \tag{69}$$

where we used the fact that $x_{\xi_1, \xi_2} = G_R(\xi_1) - G_R(\xi_2)$ and therefore, $-x_{\xi_1, \xi_2} = x_{\xi_2, \xi_1}$ (similar for $y_{\xi_1, \xi_2}$). We now claim that for all $(\xi_1, \xi_2) \in \Xi \times \Xi$ it holds that:

$$x_{\xi_1, \xi_2} - y_{\xi_1, \xi_2} + 2 \cdot \log \left( \frac{1 + \exp(y_{\xi_1, \xi_2})}{1 + \exp(x_{\xi_1, \xi_2})} \right) \leq |x_{\xi_1, \xi_2} - y_{\xi_1, \xi_2}| \tag{70}$$

We prove this claim via proof by cases:

$\underline{x_{\xi_1,\xi_2} > y_{\xi_1,\xi_2}}$: In this case we have $|x_{\xi_1,\xi_2} - y_{\xi_1,\xi_2}| = x_{\xi_1,\xi_2} - y_{\xi_1,\xi_2}$ and Equation (70) becomes:

$$2 \cdot \log\left(\frac{1 + \exp(y_{\xi_1,\xi_2})}{1 + \exp(x_{\xi_1,\xi_2})}\right) \leq 0.$$

And since $x_{\xi_1,\xi_2} > y_{\xi_1,\xi_2}$ the fraction inside the logarithm is smaller than 1, this equation must hold.

$\underline{x_{\xi_1,\xi_2} = y_{\xi_1,\xi_2}}$: In this case, Equation (70) reduces to $0 \geq 0$ which is trivially true.

$\underline{x_{\xi_1,\xi_2} < y_{\xi_1,\xi_2}}$: In this case, we have $|x_{\xi_1,\xi_2} - y_{\xi_1,\xi_2}| = y_{\xi_1,\xi_2} - x_{\xi_1,\xi_2}$ and we can reformulate Equation (70) as follows:

$$
\begin{aligned}
x_{\xi_1,\xi_2} - y_{\xi_1,\xi_2} + 2 \cdot \log\left(\frac{1 + \exp(y_{\xi_1,\xi_2})}{1 + \exp(x_{\xi_1,\xi_2})}\right) &\leq y_{\xi_1,\xi_2} - x_{\xi_1,\xi_2} \\
\iff \frac{1 + \exp(y_{\xi_1,\xi_2})}{1 + \exp(x_{\xi_1,\xi_2})} &\leq \frac{\exp(y_{\xi_1,\xi_2})}{\exp(x_{\xi_1,\xi_2})} \\
\iff \exp(x_{\xi_1,\xi_2}) &\leq \exp(y_{\xi_1,\xi_2}).
\end{aligned}
$$

Because we assume that $x_{\xi_1,\xi_2} < y_{\xi_1,\xi_2}$, the last equation, and therefore also the first, must be true.

Combining all the previous statements concludes the proof. $\qquad\square$

Finally, in some RLHF scenarios, one prefers to only compare trajectories with a common starting state. In the last lemma, we upper-bound the expected error in choice distributions with trajectories that share a common starting state by the expected error in choice distributions with arbitrary trajectories:

**Proposition C.26.** *Given an MDP $\langle \mathcal{S}, \mathcal{A}, \tau, \mu_0, R, \gamma \rangle$, a data sampling policy $\pi : \mathcal{S} \to \Delta(\mathcal{A})$ and a second reward function $\hat{R} : \mathcal{S} \times \mathcal{A} \to \mathbb{R}$, we can upper bound the expected KL divergence of preference distributions over trajectories with a common starting state as follows:*

$$\mathbb{E}_{\substack{s_0 \sim \mu_0, \\ \xi_1,\xi_2 \sim \pi(s_0)}} \left[ \mathbb{D}_{KL}\left(p_R(\cdot|\xi_1,\xi_2) \| p_{\hat{R}}(\cdot|\xi_1,\xi_2)\right) \right] \leq \frac{1}{\min_{\substack{s' \in \mathcal{S} \\ \mu_0(s')>0}} \mu_0(s')} \mathbb{E}_{\xi_1,\xi_2 \sim \pi \times \pi}\left[ \mathbb{D}_{KL}\left(p_R(\cdot|\xi_1,\xi_2) \| p_{\hat{R}}(\cdot|\xi_1,\xi_2)\right) \right]. \quad (71)$$

*Proof.* Let $s_0 : \Xi \to \mathcal{S}$ define the function which outputs the starting state $s \in \mathcal{S}$ of a trajectory $\xi \in \Xi$. We can then prove

the lemma by directly lower-bounding the right-hand side of Equation (71):

$$
\mathbb{E}_{\xi_1,\xi_2\sim\pi\times\pi}\left[\mathbb{D}_{\mathrm{KL}}\left(p_R(\cdot|\xi_1,\xi_2)||p_{\hat{R}}(\cdot|\xi_1,\xi_2)\right)\right]
$$

$$
= \sum_{s_1,s_2\in\mathcal{S}\times\mathcal{S}}\mu_0(s_1)\cdot\mu_0(s_2)\cdot\sum_{\substack{\xi_1,\xi_2\in\Xi\times\Xi\\ s_0(\xi_1)=s_1\\ s_0(\xi_2)=s_2}}p_{\pi,\tau}(\xi_1|s_1)\cdot p_{\pi,\tau}(\xi_2|s_2)\cdot\mathbb{D}_{\mathrm{KL}}\left(p_R(\cdot|\xi_1,\xi_2)||p_{\hat{R}}(\cdot|\xi_1,\xi_2)\right)
$$

$$
= \sum_{s_1=s_2}\mu_0(s_1)\cdot\mu_0(s_2)\cdot\sum_{\substack{\xi_1,\xi_2\in\Xi\times\Xi\\ s_0(\xi_1)=s_1\\ s_0(\xi_2)=s_2}}p_{\pi,\tau}(\xi_1|s_1)\cdot p_{\pi,\tau}(\xi_2|s_2)\cdot\mathbb{D}_{\mathrm{KL}}\left(p_R(\cdot|\xi_1,\xi_2)||p_{\hat{R}}(\cdot|\xi_1,\xi_2)\right)
$$

$$
+ \sum_{s_1\neq s_2}\mu_0(s_1)\cdot\mu_0(s_2)\cdot\sum_{\substack{\xi_1,\xi_2\in\Xi\times\Xi\\ s_0(\xi_1)=s_1\\ s_0(\xi_2)=s_2}}p_{\pi,\tau}(\xi_1|s_1)\cdot p_{\pi,\tau}(\xi_2|s_2)\cdot\mathbb{D}_{\mathrm{KL}}\left(p_R(\cdot|\xi_1,\xi_2)||p_{\hat{R}}(\cdot|\xi_1,\xi_2)\right)
$$

$$
\geq \sum_{s_1=s_2}\mu_0(s_1)\cdot\mu_0(s_2)\cdot\sum_{\substack{\xi_1,\xi_2\in\Xi\times\Xi\\ s_0(\xi_1)=s_1\\ s_0(\xi_2)=s_2}}p_{\pi,\tau}(\xi_1|s_1)\cdot p_{\pi,\tau}(\xi_2|s_2)\cdot\mathbb{D}_{\mathrm{KL}}\left(p_R(\cdot|\xi_1,\xi_2)||p_{\hat{R}}(\cdot|\xi_1,\xi_2)\right)
$$

$$
\geq \min_{\substack{s'\in\mathcal{S}\\ \mu_0(s')>0}}\mu_0(s')\cdot\sum_{s\in\mathcal{S}}\mu_0(s)\cdot\sum_{\substack{\xi_1,\xi_2\in\Xi\times\Xi\\ s_0(\xi_1)=s\\ s_0(\xi_2)=s}}p_{\pi,\tau}(\xi_1|s)\cdot p_{\pi,\tau}(\xi_2|s)\cdot\mathbb{D}_{\mathrm{KL}}\left(p_R(\cdot|\xi_1,\xi_2)||p_{\hat{R}}(\cdot|\xi_1,\xi_2)\right)
$$

$$
= \min_{\substack{s'\in\mathcal{S}\\ \mu_0(s')>0}}\mu_0(s')\cdot\mathbb{E}_{\substack{s_0\sim\mu_0,\\ \xi_1,\xi_2\sim\pi(s_0)}}\left[\mathbb{D}_{\mathrm{KL}}\left(p_R(\cdot|\xi_1,\xi_2)||p_{\hat{R}}(\cdot|\xi_1,\xi_2)\right)\right],
$$

where we used the fact that the KL divergence is always positive. $\qquad\square$

### C.4.2. RLHF BANDIT FORMULATION

RLHF, especially in the context of large language models, is usually modeled in a *contextual bandit* setting ( (Ziegler et al., 2019; Stiennon et al., 2020; Bai et al., 2022; Ouyang et al., 2022; Rafailov et al., 2023)). A *contextual bandit* $\langle\mathcal{S},\mathcal{A},\mu_0,R\rangle$ is defined by a set of states $\mathcal{S}$, a set of actions $\mathcal{A}$, a data distribution $\mu_0\in\Delta(\mathcal{S})$, and a reward function $R:\mathcal{S}\times\mathcal{A}\to\mathbb{R}$. The goal is to learn a policy $\pi:\mathcal{S}\to\Delta(\mathcal{A})$ which maximizes the expected return $J(\pi)=\mathbb{E}_{s\sim\mu_0,a\sim\pi(\cdot|s)}\left[R(s,a)\right]$. In the context of language models, $\mathcal{S}$ is usually called the set of prompts/contexts, and $\mathcal{A}$ the set of responses. Just as for the MDP case, we will assume for all our contextual bandits that $\max J-\min J>0$ since the reward function would otherwise be trivial. We model the human preference distribution over the set of answers $A$ using the Bradley-Terry model (Bradley & Terry, 1952). Given a prompt $s\in S$ and two answers $a_1,a_2\in A$, then the probability that a human supervisor prefers answer $a_1$ to answer $a_2$ is modelled as:

$$
p_R(a_1\succ a_2|\,s)\ =\ \frac{\exp(R(s,a_1))}{\exp(R(s,a_1))\ +\ \exp(R(s,a_2))}, \tag{72}
$$

where $R:\mathcal{S}\times\mathcal{A}\to\mathbb{R}$ is assumed to be the true, underlying reward function of the human.

RLHF is usually done with the following steps:

1. **Supervised finetuning:** Train/Fine-tune a language model $\pi_{\mathrm{ref}}$ using supervised training.

2. **Reward learning:** Given a data distribution over prompts $\mu\in\Delta(S)$, use $\mu$ and $\pi_{\mathrm{ref}}$ to sample a set of transitions $(s,a_0,a_1)\in\mathcal{S}\times\mathcal{A}\times\mathcal{A}$ where $s\sim\mu$ and $a_0,a_1\sim\pi_{\mathrm{ref}}(\cdot|s)$. Use this set of transitions to train a reward model $\hat{R}$ which minimizes the following loss:

$$
\mathcal{L}_R(\hat{R})\ =\ -\mathbb{E}_{(s,a_0,a_1,c)\sim\mu,\pi_{\mathrm{ref}},p_R}\left[\log\bigl(\sigma(\hat{R}(s,a_c)-\hat{R}(s,a_{1-c}))\bigr)\right], \tag{73}
$$

where $c\in\{0,1\}$ and $p(c=0|s,a_0,a_1)=p_R(a_0\succ a_1|s)$.

3. **RL finetuning:** Use the trained reward model $\hat{R}$ to further finetune the language model $\pi_{\text{ref}}$ using reinforcement learning. Make sure that the new model does not deviate too much from the original model by penalizing the KL divergence between the two models. This can be done by solving the following optimization problem for some $\lambda > 0$:

$$\pi = \arg \max_{\pi} \mathbb{E}_{s \sim \mu, a \sim \pi(\cdot|s)} \left[ \hat{R}(s, a) \right] - \lambda \cdot \mathbb{D}_{\text{KL}} \left( \pi(a|s) || \pi_{\text{ref}}(a|s) \right) \tag{74}$$

### C.4.3. SAFE AND UNSAFE DATA DISTRIBUTIONS FOR RLHF

**Definition C.27** (Safe- and unsafe data distributions for RLHF). For a given contextual bandit $\langle \mathcal{S}, \mathcal{A}, \mu_0, R \rangle$, let $\epsilon > 0$, $L \in [0, 1]$, $\lambda \in [0, \infty)$, and $\pi_{\text{ref}} : \mathcal{S} \to \Delta(\mathcal{A})$ an arbitrary reference policy. Similarly to Theorem 2.1, we define the set of *safe data distributions* $\mathbf{safe}^{\text{RLHF}} \left( R, \epsilon, L, \lambda, \mathbb{D}_{\text{KL}} \left( \cdot || \pi_{\text{ref}} \right) \right)$ for RLHF as all $D \in \Delta(\mathcal{S} \times \mathcal{A})$ such that for all reward functions $\hat{R} : \mathcal{S} \times \mathcal{A} \to \mathbb{R}$ and policies $\hat{\pi} : \mathcal{S} \to \Delta(\mathcal{A})$ that satisfy the following two properties:

1. **Low expected error:** $\hat{R}$ is similar to $R$ in expected choice probabilities under $D$, i.e.:

$$\mathbb{E}_{(s, a_1, a_2) \sim D} \left[ \mathbb{D}_{\text{KL}} \left( p_R(\cdot|s, a_1, a_2) || p_{\hat{R}}(\cdot|s, a_2, a_2) \right) \right] \leq \epsilon \cdot \text{range } R.$$

2. **Optimality:** $\hat{\pi}$ is optimal with respect to $\hat{R}$, i.e.:

$$\hat{\pi} \in \arg \max_{\pi} J_{\hat{R}}(\pi) - \lambda \cdot \mathbb{D}_{\text{KL}} \left( \pi(a|s) || \pi_{\text{ref}}(a|s) \right).$$

we can guarantee that $\hat{\pi}$ has regret smaller than $L$, i.e.:

3. **Low regret:** $\hat{\pi}$ has a regret smaller than $L$ with respect to $R$, i.e., $\text{Reg}^R(\hat{\pi}) < L$.

Similarly, we define the set of *unsafe data distributions* to be the complement of $\mathbf{safe}^{\text{RLHF}} \left( R, \epsilon, L, \lambda, \mathbb{D}_{\text{KL}} \left( \cdot || \pi_{\text{ref}} \right) \right)$:

$$\mathbf{unsafe}^{\text{RLHF}} \left( R, \epsilon, L, \lambda, \mathbb{D}_{\text{KL}} \left( \cdot || \pi_{\text{ref}} \right) \right) := \left\{ D \in \Delta(\mathcal{S} \times \mathcal{A}) \mid D \notin \mathbf{safe}^{\text{RLHF}} \left( R, \epsilon, L, \lambda, \mathbb{D}_{\text{KL}} \left( \cdot || \pi_{\text{ref}} \right) \right) \right\}.$$

*Note: Property 1 of Theorem C.27 is commonly phrased as minimizing (with respect to $\hat{R}$) the loss $-\mathbb{E}_{(s, a_1, a_2) \sim D, p_R} \left[ \log(\sigma(\hat{R}(s, a_1) - \hat{R}(s, a_2))) \right]$ (which includes $p_R$, the probability that $a_1$ is the preferred action over $a_2$, in the expectation). Our version of Property 1 is equivalent to this and can be derived from the former by adding the constant (w.r.t. $\hat{R}$) term $\mathbb{E}_{(s, a_1, a_2) \sim D, p_R} \left[ \log(\sigma(R(s, a_1) - R(s, a_2))) \right]$.*

### C.4.4. NEGATIVE RESULTS

In the following proofs, we will define $\pi_{R, \lambda}^{\text{rlhf}}$ to be the optimal policy after doing RLHF on $\pi_{\text{ref}}$ with some reward function $R$, i.e.,:

**Definition C.28** (RLHF-optimal policy). For any $\lambda \in \mathbb{R}_+$, reward function $R$ and reference policy $\pi_{\text{ref}}$, we define the policy maximizing the RLHF objective by:

$$\pi_{R, \lambda}^{\text{rlhf}} = \arg \max_{\pi} \mathbb{E}_{s \sim \mu, a \sim \pi(\cdot|s)} \left[ R(s, a) \right] - \lambda \cdot \mathbb{D}_{\text{KL}} \left( \pi(a|s) || \pi_{\text{ref}}(a|s) \right) \tag{75}$$

$\pi_{R, \lambda}^{\text{rlhf}}$ does have the following analytical definition (see Appendix A.1 of (Rafailov et al., 2023) for a derivation):

$$\pi_{R, \lambda}^{\text{rlhf}}(a|s) := \frac{\pi_{\text{ref}}(a|s) \cdot \exp \left( \frac{1}{\lambda} \cdot R(s, a) \right)}{\sum_{a' \in \mathcal{A}} \pi_{\text{ref}}(a'|s) \cdot \exp \left( \frac{1}{\lambda} \cdot R(s, a') \right)}. \tag{76}$$

Before stating the next negative result, we prove a small helper lemma which states that doing RLHF with some reward function $R$ on a policy $\pi_{\text{ref}}$ is guaranteed to improve the policy return concerning $R$:

**Lemma C.29.** *For any $\lambda \in \mathbb{R}_+$, reward function $R$ and reference policy $\pi_{\text{ref}}$, it holds that:*

$$J_R \left( \pi_{R, \lambda}^{\text{rlhf}} \right) \geq J_R \left( \pi_{\text{ref}} \right) \tag{77}$$

*Proof.* Define:

$$J_{\mathrm{KL}}^R(\pi, \pi_{\mathrm{ref}}) := J_R(\pi) - \lambda \mathbb{D}_{\mathrm{KL}}(\pi || \pi_{\mathrm{ref}})$$

Then we have

$$J_R\left(\pi_{R,\lambda}^{\mathrm{rlhf}}\right) \overset{(1)}{\geq} J_{\mathrm{KL}}^R(\pi_{R,\lambda}^{\mathrm{rlhf}}, \pi_{\mathrm{ref}}) \overset{(2)}{\geq} J_{\mathrm{KL}}^R(\pi_{\mathrm{ref}}, \pi_{\mathrm{ref}}) = J_R(\pi_{\mathrm{ref}})$$

where (1) follows from the non-negativity of the KL divergence and (2) follows from the fact that $\pi_{R,\lambda}^{\mathrm{rlhf}}$ maximizes $J_{\mathrm{KL}}^R(\pi, \pi_{\mathrm{ref}})$ (see Equation (75)). $\qquad\square$

We begin by proving a helper lemma that we are going to use in subsequent proofs.

**Lemma C.30.** *Let $\langle \mathcal{S}, \mathcal{A}, \mu_0, R \rangle$ be a contextual bandit.*

*Given a lower regret bound $L \in [0, 1)$, we define for every state $s \in \mathcal{S}$ the reward threshold:*

$$R_L(s) := (1 - L) \cdot \max_{a \in \mathcal{A}} R(s, a) + L \cdot \min_{a \in \mathcal{A}} R(s, a),$$

*and define $a_s \in \mathcal{A}$ to be an action such that $R(s, a_s) < R_L(s)$.*

*Let $\pi_{\mathrm{ref}} : \mathcal{S} \to \mathcal{A}$ be an arbitrary reference policy for which it holds that for every state $s \in \mathcal{S}$ we have $\pi_{\mathrm{ref}}(a|s) > 0$.*

*Then, performing KL-regularized policy optimization with some regularization constant $\lambda \in [0, \infty)$, starting from $\pi_{\mathrm{ref}}$ and using the reward function:*

$$\hat{R}(s, a) := \begin{cases} R(s, a) & \text{if } a \neq a_s \\ c_s \in \mathbb{R}_+ & \text{if } a = a_s \end{cases}, \tag{78}$$

*results in an optimal (w.r.t. the regularized optimization objective) policy $\hat{\pi}$ such that $\mathrm{Reg}^R(\hat{\pi}) \geq L$, whenever the constants $c_s$ are larger than the following lower bound:*

$$c_s \geq \lambda \cdot \log\left[\frac{\sum_{a \neq a_s}(R(s, a) - R_L(s)) \cdot \pi_{\mathrm{ref}}(a|s) \cdot \exp\left(\frac{1}{\lambda} \cdot R(s, a)\right)}{(R_L(s) - R(s, a_s)) \cdot \pi_{\mathrm{ref}}(a_s|s)}\right].$$

*Proof.* Denote by $\pi_{\hat{R},\lambda}^{\mathrm{rlhf}}$ the optimal policy for the following KL-regularized optimization problem:

$$\pi_{\hat{R},\lambda}^{\mathrm{rlhf}} \in \arg\max_{\pi} J_{\hat{R}}(\pi) - \lambda \cdot \mathbb{D}_{\mathrm{KL}}(\pi(a|s) || \pi_{\mathrm{ref}}(a|s)).$$

The closed-form solution for this optimization problem is known (see Theorem C.28). We prove the statement by assuming the specific definition of $\hat{R}$ (see Equation (78)), as well as that $\pi_{\hat{R},\lambda}^{\mathrm{rlhf}}$ has a regret at least $L$, and then working backward to derive a necessary lower bound for the individual constants $c_s$.

We start by defining a small helper policy. Let $\pi_\top$ be a deterministic optimal policy for $R$ and $\pi_\perp$ be a deterministic worst-case policy for $R$. We then define $\pi_L(a|s)$ as a convex combination of $\pi_\top$ and $\pi_\perp$:

$$\begin{aligned} \pi_L(a|s) &:= (1 - L) \cdot \pi_\top(a|s) + L \cdot \pi_\perp(a|s) \\ &= \begin{cases} 1 & \text{if } R(s, a) = \min_{a' \in \mathcal{A}} R(s, a') = \max_{a' \in \mathcal{A}} R(s, a') \\ 1 - L & \text{if } R(s, a) = \max_{a' \in \mathcal{A}} R(s, a') \\ L & \text{if } R(s, a) = \min_{a' \in \mathcal{A}} R(s, a') \\ 0 & \text{Otherwise} \end{cases} \end{aligned} \tag{79}$$

Next, we show that the regret of $\pi_L$ is $L$. Let $\eta_\top$ and $\eta_\perp$ be the corresponding occupancy measures of $\pi_\top$ and $\pi_\perp$. Then, we have:

$$J_R(\pi_L) = (1 - L) \cdot R^T \cdot \eta_\top + L \cdot R^T \cdot \eta_\perp,$$

from which it directly follows that:

$$\text{Reg}^R(\pi_L) = \frac{R^T \cdot \eta_\top - [(1-L) \cdot R^T \cdot \eta_\top + L \cdot R^T \cdot \eta_\bot]}{R^T \cdot \eta_\top - R^T \cdot \eta_\bot} = L.$$

Now, having defined $\pi_L$, we start the main proof. Assume that $\text{Reg}^R\left(\pi_{\hat{R},\lambda}^{\text{rlhf}}\right) \geq L$, which is equivalent to $J(\pi_{\hat{R},\lambda}^{\text{rlhf}}) \leq J(\pi_L)$. By using the definition of the policy evaluation function, we get:

$$J(\pi_{\hat{R},\lambda}^{\text{rlhf}}) \leq J(\pi_L)$$

$$\iff R^T \cdot (\eta^{\pi_{\hat{R},\lambda}^{\text{rlhf}}} - \eta^{\pi_L}) \leq 0$$

$$\iff \sum_{(s,a)\in\mathcal{S}\times\mathcal{A}} R(s,a) \cdot \mu_0(s) \cdot (\pi_{\hat{R},\lambda}^{\text{rlhf}}(a|s) - \pi_L(a|s)) \leq 0$$

We will prove the sufficient condition, that for every $s \in \mathcal{S}$, we have:

$$\sum_{a\in\mathcal{A}} R(s,a) \cdot \left(\pi_{\hat{R},\lambda}^{\text{rlhf}}(a|s) - \pi_L(a|s)\right) \leq 0 \tag{80}$$

Before continuing, note that with our definition of $\pi_L$ (see Equation (79)) we have:

$$\sum_{a\in\mathcal{A}} R(s,a) \cdot \pi_L(a|s) = (1-L) \cdot \max_{a\in\mathcal{A}} R(s,a) + L \cdot \min_{a\in\mathcal{A}} R(s,a) =: R_L(s).$$

Now, using this fact as well as the definitions of $\pi_L$ and $\pi_{\hat{R},\lambda}^{\text{rlhf}}$ (see Theorem C.28) we prove under which conditions Equation (80) holds:

$$\sum_{a\in\mathcal{A}} R(s,a) \cdot \left(\pi_{\hat{R},\lambda}^{\text{rlhf}}(a|s) - \pi_L(a|s)\right) \leq 0$$

$$\iff \sum_{a\in\mathcal{A}} R(s,a) \cdot \left[\frac{\pi_{\text{ref}}(a|s) \cdot \exp\left(\frac{1}{\lambda} \cdot \hat{R}(s,a)\right)}{\sum_{a'\in\mathcal{A}} \pi_{\text{ref}}(a'|s) \cdot \exp\left(\frac{1}{\lambda} \cdot \hat{R}(s,a')\right)} - \pi_L(a|s)\right] \leq 0$$

$$\iff \sum_{a\in\mathcal{A}} R(s,a) \cdot \pi_{\text{ref}}(a|s) \cdot \exp\left(\frac{1}{\lambda} \cdot \hat{R}(s,a)\right)$$

$$\leq \left[\sum_{a\in\mathcal{A}} R(s,a) \cdot \pi_L(a|s)\right] \cdot \sum_{a'\in\mathcal{A}} \pi_{\text{ref}}(a'|s) \cdot \exp\left(\frac{1}{\lambda} \cdot \hat{R}(s,a')\right)$$

$$\iff \sum_{a\in\mathcal{A}} (R(s,a) - R_L(s)) \cdot \pi_{\text{ref}}(a|s) \cdot \exp\left(\frac{1}{\lambda} \cdot \hat{R}(s,a)\right) \leq 0$$

$$\iff \sum_{\substack{a\in\mathcal{A} \\ R(s,a)>R_L(s)}} (R(s,a) - R_L(s)) \cdot \pi_{\text{ref}}(a|s) \cdot \exp\left(\frac{1}{\lambda} \cdot \hat{R}(s,a)\right)$$

$$\leq \sum_{\substack{a\in\mathcal{A} \\ R(s,a)<R_L(s)}} (R_L(s) - R(s,a)) \cdot \pi_{\text{ref}}(a|s) \cdot \exp\left(\frac{1}{\lambda} \cdot \hat{R}(s,a)\right)$$

Now, according to the assumptions of the lemma, we know that there exists some action $a_s$ for which $R(s,a_s) < R_L(s)$ and $\pi_{\text{ref}}(a_s|s) > 0$. According to our definition of $\hat{R}$ (see Equation (78)), we have $\hat{R}(s,a_s) = c_s$ and $\hat{R}(s,a) = R(s,a)$ for

all other actions. We can use this definition to get a lower bound for $c_s$:

$$\sum_{\substack{a \in \mathcal{A} \\ R(s,a) > R_L(s)}} (R(s,a) - R_L(s)) \cdot \pi_{\text{ref}}(a|s) \cdot \exp\left(\frac{1}{\lambda} \cdot \hat{R}(s,a)\right)$$

$$\leq \sum_{\substack{a \in \mathcal{A} \\ R(s,a) < R_L(s)}} (R_L(s) - R(s,a)) \cdot \pi_{\text{ref}}(a|s) \cdot \exp\left(\frac{1}{\lambda} \cdot \hat{R}(s,a)\right) \tag{81}$$

$$\iff \sum_{a \neq a_s} (R(s,a) - R_L(s)) \cdot \pi_{\text{ref}}(a|s) \cdot \exp\left(\frac{1}{\lambda} \cdot R(s,a)\right)$$

$$\leq (R_L(s) - R(s,a_s)) \cdot \pi_{\text{ref}}(a_s|s) \cdot \exp\left(\frac{1}{\lambda} \cdot \hat{R}(s,a_s)\right) \tag{82}$$

$$\iff \lambda \cdot \log\left[\frac{\sum_{a \neq a_s} (R(s,a) - R_L(s)) \cdot \pi_{\text{ref}}(a|s) \cdot \exp\left(\frac{1}{\lambda} \cdot R(s,a)\right)}{(R_L(s) - R(s,a_s)) \cdot \pi_{\text{ref}}(a_s|s)}\right] \leq \hat{R}(s,a_s). \tag{83}$$

$$\square$$

We can now use this lemma to prove a more general result:

**Proposition C.31.** *Let $\langle \mathcal{S}, \mathcal{A}, \mu_0, R \rangle$ be a contextual bandit.*

*Given a lower regret bound $L \in [0,1)$, we define for every state $s \in \mathcal{S}$ the reward threshold:*

$$R_L(s) := (1-L) \cdot \max_{a \in \mathcal{A}} R(s,a) + L \cdot \min_{a \in \mathcal{A}} R(s,a),$$

*Lastly, $\pi_{\text{ref}} : \mathcal{S} \to \mathcal{A}$ be an arbitrary reference policy for which it holds that for every state $s \in \mathcal{S}$, $\pi_{\text{ref}}(a|s) > 0$ and there exists at least one action $a_s \in \mathcal{A}$ such that:*

a) *$\pi_{\text{ref}}(a_s|s)$ is small enough, that the following inequality holds:*

$$\log\left[\sum_{a \neq a_s} \pi_{\text{ref}}(a|s) \cdot \exp\left(\frac{1}{\lambda} \cdot (R(s,a) - R(s,a_s))\right) \cdot \frac{R(s,a) - R_L(s)}{R_L(s) - R(s,a_s)}\right] \leq \frac{\epsilon \cdot \text{range } R}{2 \cdot \lambda \cdot \pi_{\text{ref}}(a_s|s)} + \log(\pi_{\text{ref}}(a_s|s)) \tag{84}$$

b) *$R(s,a_s) < R_L(s)$*

*for some $\epsilon > 0$ and $\lambda \in [0, \infty)$. Let $D_\mu^{\text{ref}}(s,a) := \mu \cdot \pi_{\text{ref}}(a|s)$ be a data distribution based on the reference policy and some $\mu \in \Delta(\mathcal{S})$. Then $D_\mu^{\text{ref}} \in \textbf{unsafe}^{\text{RLHF}}\left(R, \epsilon, L, \lambda, \mathbb{D}_{KL}(\cdot||\pi_{\text{ref}})\right)$*

*Proof.* According to the definition of a safe data distribution for RLHF (see Theorem C.27), $D_\mu^{\text{ref}} \in \textbf{unsafe}^{\text{RLHF}}\left(R, \epsilon, L, \lambda, \mathbb{D}_{KL}(\cdot||\pi_{\text{ref}})\right)$ if there exists a reward function $\hat{R} : \mathcal{S} \times \mathcal{A} \to \mathbb{R}$, and a policy $\hat{\pi} : S \to \Delta(\mathcal{A})$ such that:

1. $\mathbb{E}_{s,a_1,a_2 \sim \mu, \pi_{\text{ref}}} \left[\mathbb{D}_{\text{KL}}\left(p_R(\cdot|s,a_1,a_2)||p_{\hat{R}}(\cdot|s,a_1,a_2)\right)\right] \leq \epsilon \cdot \text{range } R$

2. $\hat{\pi} \in \arg\max_\pi J_{\hat{R}}(\pi) - \lambda \cdot \mathbb{D}_{\text{KL}}(\pi(a|s)||\pi_{\text{ref}}(a|s))$

3. $\text{Reg}^R(\hat{\pi}) \geq L$,

We will prove the lemma by construction. Namely, given the assumptions $a)$ and $b)$ of Theorem C.31, we choose:

$$\hat{R}(s,a) := \begin{cases} R(s,a) & \text{if } a \neq a_s \\ c_s \in \mathbb{R}_+ & \text{if } a = a_s \end{cases} \tag{85}$$

where the different $c_s$ are some positive constants defined as follows:

$$\hat{R}(s, a_s) = c_s \; \geq \; l_s := \max \left( R(s, a_s), \; \lambda \cdot \log \left[ \frac{\sum_{a \neq a_s} (R(s, a) - R_L(s)) \cdot \pi_{\text{ref}}(a|s) \cdot \exp \left( \frac{1}{\lambda} \cdot R(s, a) \right)}{(R_L(s) - R(s, a_s)) \cdot \pi_{\text{ref}}(a_s|s)} \right] \right). \quad (86)$$

Furthermore, the closed-form of the optimal policy $\hat{\pi}$ of the KL-regularized optimization problem is known to be $\pi_{\hat{R}, \lambda}^{\text{rlhf}}$ (see Theorem C.28). We now claim that this choice of $\hat{R}$ and $\hat{\pi}$ fulfills properties (1) and (3) of the above list (property (2) is true by assumption).

Property (3) is true because every reference policy $\pi_{\text{ref}}$ and corresponding reward function $R$ that fulfills the conditions of this proposition also fulfills the conditions of Theorem C.30. Hence, we can directly apply Theorem C.30 and get the guarantee that $\text{Reg}^R(\hat{\pi}) \geq L$.

All that remains to be shown, is that condition (1) can be satisfied by using the definition of $\hat{R}$ and the lower bounds in Equation Equation (86). First, note that we can reformulate the expected error definition in condition (1) as follows:

$$\mathbb{E}_{s, a_1, a_2 \sim \mu, \pi_{\text{ref}}} \left[ \mathbb{D}_{\text{KL}} \left( p_R(\cdot|s, a_1, a_2) || p_{\hat{R}}(\cdot|s, a_1, a_2) \right) \right]$$

$$= \sum_{s \in \mathcal{S}} \mu(s) \cdot \sum_{a_1, a_2 \in \mathcal{A} \times \mathcal{A}} \pi_{\text{ref}}(a_1|s) \cdot \pi_{\text{ref}}(a_2|s) \cdot \sum_{i, j \in \{1, 2\}} \sigma(R(s, a_i) - R(s, a_j)) \cdot \log \left( \frac{\sigma(R(s, a_i) - R(s, a_j))}{\sigma(\hat{R}(s, a_i) - \hat{R}(s, a_j))} \right)$$

$$= 2 \cdot \sum_{s \in \mathcal{S}} \mu(s) \cdot \sum_{a_1, a_2 \in \mathcal{A} \times \mathcal{A}} \pi_{\text{ref}}(a_1|s) \cdot \pi_{\text{ref}}(a_2|s) \cdot \underbrace{\sigma(R(s, a_1) - R(s, a_2)) \cdot \log \left( \frac{\sigma(R(s, a_1) - R(s, a_2))}{\sigma(\hat{R}(s, a_1) - \hat{R}(s, a_2))} \right)}_{=:\mathcal{IS}(a_1, a_2)}$$

$$= 2 \cdot \sum_{s \in \mathcal{S}} \mu(s) \cdot \sum_{a_1, a_2 \in \mathcal{A} \times \mathcal{A}} \pi_{\text{ref}}(a_1|s) \cdot \pi_{\text{ref}}(a_2|s) \cdot \mathcal{IS}(a_1, a_2).$$

Next, note that for every tuple $(a_1, a_2) \in \mathcal{A}$, the sum $\mathcal{IS}(a_1, a_2) + \mathcal{IS}(a_2, a_1)$ can be reformulated as follows:

$$\mathcal{IS}(a_1, a_2) \; + \; \mathcal{IS}(a_2, a_1)$$

$$= \sigma(R(s, a_1) - R(s, a_2)) \cdot \log \left( \frac{\sigma(R(s, a_1) - R(s, a_2))}{\sigma(\hat{R}(s, a_1) - \hat{R}(s, a_2))} \right)$$

$$+ \sigma(R(s, a_2) - R(s, a_1)) \cdot \log \left( \frac{\sigma(R(s, a_2) - R(s, a_1))}{\sigma(\hat{R}(s, a_2) - \hat{R}(s, a_1))} \right)$$

$$= \sigma(R(s, a_1) - R(s, a_2)) \cdot \log \left( \frac{\sigma(R(s, a_1) - R(s, a_2))}{\sigma(\hat{R}(s, a_1) - \hat{R}(s, a_2))} \right)$$

$$+ \left( 1 - \sigma(R(s, a_1) - R(s, a_2)) \right) \cdot \log \left( \frac{\sigma(R(s, a_2) - R(s, a_1))}{\sigma(\hat{R}(s, a_2) - \hat{R}(s, a_1))} \right)$$

$$= \sigma(R(s, a_1) - R(s, a_2)) \cdot \underbrace{\left[ \log \left( \frac{\sigma(R(s, a_1) - R(s, a_2))}{\sigma(\hat{R}(s, a_1) - \hat{R}(s, a_2))} \right) - \log \left( \frac{\sigma(R(s, a_2) - R(s, a_1))}{\sigma(\hat{R}(s, a_2) - \hat{R}(s, a_1))} \right) \right]}_{(A)}$$

$$+ \underbrace{\log \left( \frac{\sigma(R(s, a_2) - R(s, a_1))}{\sigma(\hat{R}(s, a_2) - \hat{R}(s, a_1))} \right)}_{(B)}.$$

The term (A) can now be simplified as follows:

$$\log\left(\frac{\sigma(R(s,a_1) - R(s,a_2))}{\sigma(\hat{R}(s,a_1) - \hat{R}(s,a_2))}\right) - \log\left(\frac{\sigma(R(s,a_2) - R(s,a_1))}{\sigma(\hat{R}(s,a_2) - \hat{R}(s,a_1))}\right)$$

$$= \log\left(\frac{\sigma(R(s,a_1) - R(s,a_2))}{1 - \sigma(R(s,a_1) - R(s,a_2))}\right) + \log\left(\frac{1 - \sigma(\hat{R}(s,a_1) - \hat{R}(s,a_2))}{\sigma(\hat{R}(s,a_1) - \hat{R}(s,a_2))}\right)$$

$$= [R(s,a_1) - R(s,a_2)] - [\hat{R}(s,a_1) - \hat{R}(s,a_2)],$$

where we used the definition of the inverse of the logistic function. Similarly, the term (B) can be simplified as follows:

$$\log\left(\frac{\sigma(R(s,a_2) - R(s,a_1))}{\sigma(\hat{R}(s,a_2) - \hat{R}(s,a_1))}\right)$$

$$= \log\left(\frac{\exp(R(s,a_2) - R(s,a_1))}{1 + \exp(R(s,a_2) - R(s,a_1))} \cdot \frac{1 + \exp(\hat{R}(s,a_2) - \hat{R}(s,a_1))}{\exp(\hat{R}(s,a_2) - \hat{R}(s,a_1))}\right)$$

$$= [R(s,a_2) - R(s,a_1)] - [\hat{R}(s,a_2) - \hat{R}(s,a_1)] + \log\left(\frac{1 + \exp(\hat{R}(s,a_2) - \hat{R}(s,a_1))}{1 + \exp(R(s,a_2) - R(s,a_1))}\right)$$

These expressions, together with the fact that $\mathcal{IS}(a, a) = 0$ for all $a \in \mathcal{A}$, allow us to choose an arbitrary ordering $\prec$ on the set of actions $\mathcal{A}$, and then re-express the sum:

$$\sum_{a_1, a_2 \in \mathcal{A} \times \mathcal{A}} \pi_{\text{ref}}(a_1|s) \cdot \pi_{\text{ref}}(a_2|s) \cdot \mathcal{IS}(a_1, a_2) = \sum_{\substack{a_1, a_2 \in \mathcal{A} \times \mathcal{A} \\ a_1 \prec a_2}} \pi_{\text{ref}}(a_1|s) \cdot \pi_{\text{ref}}(a_2|s) \cdot \big(\mathcal{IS}(a_1, a_2) + \mathcal{IS}(a_2, a_1)\big). \quad (87)$$

Summarizing all the equations above, we get:

$$\mathbb{E}_{s, a_1, a_2 \sim \mu, \pi_{\text{ref}}} \big[\mathbb{D}_{\text{KL}}\big(p_R(\cdot|s, a_1, a_2) || p_{\hat{R}}(\cdot|s, a_1, a_2)\big)\big]$$

$$= 2 \cdot \sum_{s \in \mathcal{S}} \mu(s) \cdot \sum_{a_1, a_2 \in \mathcal{A} \times \mathcal{A}} \pi_{\text{ref}}(a_1|s) \cdot \pi_{\text{ref}}(a_2|s) \cdot \mathcal{IS}(a_1, a_2)$$

$$= 2 \cdot \sum_{s \in \mathcal{S}} \mu(s) \cdot \sum_{\substack{a_1, a_2 \in \mathcal{A} \times \mathcal{A} \\ a_1 \prec a_2}} \pi_{\text{ref}}(a_1|s) \cdot \pi_{\text{ref}}(a_2|s) \cdot \left[\Big([R(s,a_1) - R(s,a_2)] - [\hat{R}(s,a_1) - \hat{R}(s,a_2)]\Big) \right. \quad (88)$$

$$\left. \cdot \Big(\sigma(R(s,a_1) - R(s,a_2)) - 1\Big) + \log\left(\frac{1 + \exp(\hat{R}(s,a_2) - \hat{R}(s,a_1))}{1 + \exp(R(s,a_2) - R(s,a_1))}\right)\right].$$

Now, by using our particular definition of $\hat{R}$ (see Equation (85)), we notice that whenever both $a_1 \neq a_s$, and $a_2 \neq a_s$, the inner summand of Equation (88) is zero. What remains of Equation (88) can be restated as follows:

$$= 2 \cdot \sum_{s \in \mathcal{S}} \mu(s) \cdot \pi_{\text{ref}}(a_s|s) \cdot \sum_{a \in \mathcal{A}} \pi_{\text{ref}}(a|s) \cdot \left[\big(R(s,a_s) - c_s\big) \cdot \Big(\sigma(R(s,a_s) - R(s,a)) - 1\Big)\right. \quad (89)$$

$$\left. + \log\left(\frac{1 + \exp(R(s,a) - c_s)}{1 + \exp(R(s,a) - R(s,a_s))}\right)\right]$$

To prove property (1), we must show that Equation (89) is smaller or equal to $\epsilon \cdot \text{range } R$. We do this in two steps. First, note that for all states $s$ it holds that $c_s \geq R(s, a_s)$ (this is obvious from the definition of $c_s$, see Equation (86)). This allows us to simplify Equation (89) by dropping the logarithm term.

$$\mathbb{E}_{s,a_1,a_2 \sim \mu,\pi_{\text{ref}}} \left[ \mathbb{D}_{\text{KL}} \left( p_R(\cdot|s,a_1,a_2) || p_{\hat{R}}(\cdot|s,a_1,a_2) \right) \right]$$

$$= 2 \cdot \sum_{s \in \mathcal{S}} \mu(s) \cdot \pi_{\text{ref}}(a_s|s) \cdot \sum_{a \in \mathcal{A}} \pi_{\text{ref}}(a|s) \cdot \left[ (R(s,a_s) - c_s) \cdot \left( \sigma(R(s,a_s) - R(s,a)) - 1 \right) \right.$$
$$\left. + \log \left( \frac{1 + \exp(R(s,a) - c_s)}{1 + \exp(R(s,a) - R(s,a_s))} \right) \right]$$

$$= 2 \cdot \sum_{s \in \mathcal{S}} \mu(s) \cdot \pi_{\text{ref}}(a_s|s) \cdot \left( c_s - R(s,a_s) \right) \cdot \sum_{a \in \mathcal{A}} \pi_{\text{ref}}(a|s) \cdot \left( 1 - \sigma(R(s,a_s) - R(s,a)) \right) \tag{90}$$
$$+ 2 \cdot \sum_{s \in \mathcal{S}} \mu(s) \cdot \pi_{\text{ref}}(a_s|s) \cdot \sum_{a \in \mathcal{A}} \pi_{\text{ref}}(a|s) \cdot \log \left( \frac{1 + \exp(R(s,a) - c_s)}{1 + \exp(R(s,a) - R(s,a_s))} \right).$$

Now, we choose to define $c_s := l_s + \delta_s$, where $l_s$ is defined in Equation (86) and $\delta_s \geq 0$ such that:

$$2 \cdot \sum_{s \in \mathcal{S}} \mu(s) \cdot \pi_{\text{ref}}(a_s|s) \cdot \left( l_s + \delta_s - R(s,a_s) \right) \cdot \sum_{a \in \mathcal{A}} \pi_{\text{ref}}(a|s) \cdot \underbrace{\left( 1 - \sigma(R(s,a_s) - R(s,a)) \right)}_{<1}$$
$$+ 2 \cdot \sum_{s \in \mathcal{S}} \mu(s) \cdot \pi_{\text{ref}}(a_s|s) \cdot \sum_{a \in \mathcal{A}} \pi_{\text{ref}}(a|s) \cdot \underbrace{\log \left( \frac{1 + \exp(R(s,a) - l_s - \delta_s)}{1 + \exp(R(s,a) - R(s,a_s))} \right)}_{\leq 0 \text{ (because } c_s := l_s + \delta_s \geq R(s,a_s))}$$
$$\leq 2 \cdot \sum_{s \in \mathcal{S}} \mu(s) \cdot \pi_{\text{ref}}(a_s|s) \cdot \left( l_s - R(s,a_s) \right) \overset{!}{\leq} \epsilon \cdot \text{range } R. \tag{91}$$

Note that the first inequality is always feasible, as we could just choose $\delta_s = 0$ for all $s \in \mathcal{S}$ in which case the inequality must hold due to the last term in the first line being smaller than one and the last term in the second line being negative. Now, to prove Equation (91), we prove the sufficient condition that for every state $s \in \mathcal{S}$:

$$\pi_{\text{ref}}(a_s|s) \cdot (l_s - R(s,a_s)) \overset{!}{\leq} \frac{\epsilon \cdot \text{range } R}{2}. \tag{92}$$

In case that $l_s = R(s,a_s)$, the left-hand side of Equation (92) cancels and the inequality holds trivially. We can therefore focus on the case where $l_s > R(s,a_s)$. In this case, we get:

$$\pi_{\text{ref}}(a_s|s) \cdot \lambda \cdot \log \left[ \frac{\sum_{a \neq a_s} (R(s,a) - R_L(s)) \cdot \pi_{\text{ref}}(a|s) \cdot \exp \left( \frac{1}{\lambda} \cdot R(s,a) \right)}{(R_L(s) - R(s,a_s)) \cdot \pi_{\text{ref}}(a_s|s) \cdot \exp \left( \frac{1}{\lambda} \cdot R(s,a_s) \right)} \right] \overset{!}{\leq} \frac{\epsilon \cdot \text{range } R}{2}$$

$$\iff \log \left[ \sum_{a \neq a_s} \pi_{\text{ref}}(a|s) \cdot \exp \left( \frac{1}{\lambda} \cdot (R(s,a) - R(s,a_s)) \right) \cdot \frac{R(s,a) - R_L(s)}{R_L(s) - R(s,a_s)} \right]$$
$$\overset{!}{\leq} \frac{\epsilon \cdot \text{range } R}{2 \cdot \lambda \cdot \pi_{\text{ref}}(a_s|s)} + \log(\pi_{\text{ref}}(a_s|s))$$

which holds by assumption (a) of the lemma. Therefore, property (1) of the lemma must hold as well which concludes the proof. $\qquad \square$

**Theorem C.32.** *Let $\langle \mathcal{S}, \mathcal{A}, \mu_0, R \rangle$ be a contextual bandit. Given $L \in [0,1)$, we define for every state $s \in \mathcal{S}$ the reward threshold:$R_L(s) := (1 - L) \cdot \max_{a \in \mathcal{A}} R(s,a) + L \cdot \min_{a \in \mathcal{A}} R(s,a)$. Lastly, let $\pi_{\text{ref}} : \mathcal{S} \to \mathcal{A}$ be an arbitrary reference policy for which it holds that for every $(s,a) \in \mathcal{S} \times \mathcal{A}$, $\pi_{\text{ref}}(a|s) > 0$, and there exists at least one action $a_s \in \mathcal{A}$ such that $R(s,a_s) < R_L(s)$ and $\pi_{\text{ref}}(a_s|s)$ satisfies the following inequality:*

$$\pi_{\text{ref}}(a_s|s) \leq \frac{(R_L(s) - R(s,a_s)) \cdot \text{range } R}{L \cdot \exp \left( \frac{1}{\lambda} \cdot \text{range } R \right)} \cdot \frac{\epsilon^2}{4 \cdot \lambda^2}.$$

*Let $D_\mu^{\text{ref}}(s,a) := \mu(s) \cdot \pi_{\text{ref}}(a|s)$ for some $\mu \in \Delta(S)$. Then*
*$D_\mu^{\text{ref}} \in \textbf{unsafe}^{\text{RLHF}} \left( R, \epsilon, L, \lambda, \mathbb{D}_{KL}(\cdot||\pi_{\text{ref}}) \right)$*

*Proof.* We begin by showing that every $\pi_{\text{ref}}$ that fulfills the conditions of the theorem also satisfies properties $a)$ and $b)$ of Theorem C.31. Let $s$ be an arbitrary state and $a_s$ the corresponding action that fulfills the conditions stated in the theorem. We show that condition $a)$ of Theorem C.31 holds via direct derivation:

$$
\pi_{\text{ref}}(a_s|s) \quad \leq \quad \frac{R_L(s) - R(s, a_s)}{L} \cdot \frac{\text{range } R}{\exp\left(\frac{1}{\lambda} \cdot \text{range } R\right)} \cdot \frac{\epsilon^2}{4 \cdot \lambda^2}
$$

$$
\implies \quad \frac{1}{\sqrt{\text{range } R}} \cdot \lambda \cdot \sqrt{\frac{\pi_{\text{ref}}(a_s|s) \cdot L \cdot \exp\left(\frac{1}{\lambda} \cdot \text{range } R\right)}{R_L(s) - R(s, a_s)}} \quad \leq \quad \frac{\epsilon}{2}
$$

$$
\implies \quad \pi_{\text{ref}}(a_s|s) \cdot \lambda \cdot \sqrt{\frac{L \cdot \text{range } R \cdot \exp\left(\frac{1}{\lambda} \cdot \text{range } R\right)}{(R_L(s) - R(s, a_s)) \cdot \pi_{\text{ref}}(a_s|s)}} \quad \leq \quad \frac{\epsilon \cdot \text{range } R}{2}
$$

We continue by lower-bounding the square-root term as follows:

$$
\lambda \cdot \sqrt{\frac{L \cdot \text{range } R \cdot \exp\left(\frac{1}{\lambda} \cdot \text{range } R\right)}{(R_L(s) - R(s, a_s)) \cdot \pi_{\text{ref}}(a_s|s)}}
$$

$$
\geq \quad \lambda \cdot \log\left[\frac{L \cdot \text{range } R \cdot \exp\left(\frac{1}{\lambda} \cdot \text{range } R\right)}{(R_L(s) - R(s, a_s)) \cdot \pi_{\text{ref}}(a_s|s)}\right]
$$

$$
\geq \quad \lambda \cdot \log\left[\frac{L \cdot \text{range } R \cdot \exp\left(\frac{1}{\lambda} \cdot \left[\max_{a \in \mathcal{A}} R(s, a) - R(s, a_s)\right]\right)}{(R_L(s) - R(s, a_s)) \cdot \pi_{\text{ref}}(a_s|s)}\right]
$$

$$
\geq \quad \lambda \cdot \log\left[\frac{(\max_{a \in \mathcal{A}} R(s, a) - R_L(s)) \cdot \exp\left(\frac{1}{\lambda} \cdot \max_{a \in \mathcal{A}} R(s, a)\right)}{(R_L(s) - R(s, a_s)) \cdot \pi_{\text{ref}}(a_s|s) \cdot \exp\left(\frac{1}{\lambda} \cdot R(s, a_s)\right)}\right]
$$

$$
\geq \quad \lambda \cdot \log\left[\frac{\sum_{a \neq a_s}(R(s, a) - R_L(s)) \cdot \pi_{\text{ref}}(a|s) \cdot \exp\left(\frac{1}{\lambda} \cdot R(s, a)\right)}{(R_L(s) - R(s, a_s)) \cdot \pi_{\text{ref}}(a_s|s) \cdot \exp\left(\frac{1}{\lambda} \cdot R(s, a_s)\right)}\right]
$$

By applying this lower bound, we can finish the derivation of condition $a)$ of Theorem C.31:

$$
\pi_{\text{ref}}(a_s|s) \quad \leq \quad \frac{R_L(s) - R(s, a_s)}{L} \cdot \frac{\text{range } R}{\exp\left(\frac{1}{\lambda} \cdot \text{range } R\right)} \cdot \frac{\epsilon^2}{4 \cdot \lambda^2}
$$

$$
\implies \quad \pi_{\text{ref}}(a_s|s) \cdot \lambda \cdot \sqrt{\frac{L \cdot \text{range } R \cdot \exp\left(\frac{1}{\lambda} \cdot \text{range } R\right)}{(R_L(s) - R(s, a_s)) \cdot \pi_{\text{ref}}(a_s|s)\cdot}} \quad \leq \quad \frac{\epsilon \cdot \text{range } R}{2}
$$

$$
\implies \quad \pi_{\text{ref}}(a_s|s) \cdot \lambda \cdot \log\left[\frac{\sum_{a \neq a_s}(R(s, a) - R_L(s)) \cdot \pi_{\text{ref}}(a|s) \cdot \exp\left(\frac{1}{\lambda} \cdot R(s, a)\right)}{(R_L(s) - R(s, a_s)) \cdot \pi_{\text{ref}}(a_s|s) \cdot \exp\left(\frac{1}{\lambda} \cdot R(s, a_s)\right)}\right] \quad \leq \quad \frac{\epsilon \cdot \text{range } R}{2}
$$

$$
\implies \quad \log\left[\sum_{a \neq a_s} \pi_{\text{ref}}(a|s) \cdot \exp\left(\frac{1}{\lambda} \cdot (R(s, a) - R(s, a_s))\right) \cdot \frac{R(s, a) - R_L(s)}{R_L(s) - R(s, a_s)}\right]
$$

$$
\leq \quad \frac{\epsilon \cdot \text{range } R}{2 \cdot \lambda \cdot \pi_{\text{ref}}(a_s|s)} + \log(\pi_{\text{ref}}(a_s|s))
$$

Before moving on, note that condition $b)$ of Theorem C.31 holds directly by assumption of this theorem. Therefore, we have shown that every $\pi_{\text{ref}}$ that fulfills the conditions of this theorem also fulfills the two conditions of Theorem C.31. This must mean that $D_\mu^{\text{ref}} \in \textbf{unsafe}^{\text{RLHF}}\left(R, \epsilon, L, \lambda, \mathbb{D}_{\text{KL}}\left(\cdot || \pi_{\text{ref}}\right)\right)$ for arbitrary $\mu \in \Delta(S)$ thereby concluding the proof. $\qquad \square$

### C.4.5. ANOTHER NEGATIVE RESULT WITH REGULARIZATION

**Proposition C.33.** *Let $\langle \mathcal{S}, \mathcal{A}, \mu_0, R \rangle$ be a contextual bandit.*

*Given a lower regret bound $L \in [0, 1)$, we define for every state $s \in \mathcal{S}$ the reward threshold:*

$$
R_L(s) := (1 - L) \cdot \max_{a \in \mathcal{A}} R(s, a) + L \cdot \min_{a \in \mathcal{A}} R(s, a),
$$

*Lastly, let $\pi_{\text{ref}} : \mathcal{S} \to \mathcal{A}$ be an arbitrary reference policy for which it holds that for every state $s \in \mathcal{S}$, $\pi_{\text{ref}}(a|s) > 0$ and there exists at least one action $a_s \in \mathcal{A}$ such that:*

a) *$\pi_{\text{ref}}(a_s|s)$ is small enough, that the following inequality holds:*

$$\pi_{\text{ref}}(a_s|s) \;\leq\; \frac{(R_L(s) - R(s, a_s))}{L} \cdot \frac{\text{range } R}{\exp\left(\frac{1}{\lambda} \cdot \text{range } R\right)} \cdot \frac{\epsilon^2}{\lambda^2} \tag{93}$$

b) *$R(s, a_s) < R_L(s)$*

*Let $D^{\text{ref}}(s, a) := \mu_0(s) \cdot \pi_{\text{ref}}(a \mid s)$. Then $D^{\text{ref}} \in \mathbf{unsafe}(R, \epsilon, L, \lambda, \omega)$.*

*Proof.* To prove the proposition we show that there exists some reward function $\hat{R}$, as well as a policy $\hat{\pi}$ such that the following properties hold:

1. $\mathbb{E}_{(s,a) \sim D^{\text{ref}}} \left[ \frac{|R(s,a) - \hat{R}(s,a)|}{\text{range } R} \right] \leq \epsilon$.

2. $\hat{\pi} \in \arg\max_\pi J_{\hat{R}}(\pi) - \lambda \omega(\pi)$

3. $\text{Reg}^R(\hat{\pi}) \geq L$.

In particular, we choose:

$$\hat{R}(s, a) \;:=\; \begin{cases} R(s, a) & \text{if } a \neq a_s \\ c_s \in \mathbb{R}_+ & \text{if } a = a_s \end{cases}, \tag{94}$$

where the different $c_s$ are some positive constants defined as follows:

$$\hat{R}(s, a_s) = c_s := \max\left( R(s, a_s), \; \lambda \cdot \log\left[ \frac{\sum_{a \neq a_s}(R(s, a) - R_L(s)) \cdot \pi_{\text{ref}}(a|s) \cdot \exp\left(\frac{1}{\lambda} \cdot R(s, a)\right)}{(R_L(s) - R(s, a_s)) \cdot \pi_{\text{ref}}(a_s|s)} \right] \right). \tag{95}$$

Furthermore, the closed-form of the optimal policy $\hat{\pi}$ of the KL-regularized optimization problem is known to be $\pi_{\hat{R}, \lambda}^{\text{rlhf}}$ (see Theorem C.28). We now claim that this choice of $\hat{R}$ and $\hat{\pi}$ fulfills properties (1) and (3) of the lemma (property (2) is true by assumption).

Property (3) is true because every reference policy $\pi_{\text{ref}}$ and corresponding reward function $R$ that fulfills the conditions of this proposition also fulfills the conditions of Theorem C.30. Hence, we can directly apply Theorem C.30 and get the guarantee that $\text{Reg}^R(\hat{\pi}) \geq L$.

All that remains to be shown, is that condition (1) can be satisfied by using the definition of $\hat{R}$ and in particular, the definition of the individual $c_s$ (see Equation (95)). The expected error expression in condition (1) can be expanded as follows:

$$\mathbb{E}_{(s,a) \sim D^{\text{ref}}} \left[ \frac{|R(s, a) - \hat{R}(s, a)|}{\text{range } R} \right] = \sum_{(s,a) \in \mathcal{S} \times \mathcal{A}} \mu_0(s) \cdot \pi_{\text{ref}}(a|s) \cdot \frac{|R(s, a) - \hat{R}(s, a)|}{\text{range } R} \overset{!}{\leq} \epsilon.$$

We show the sufficient condition that for each state $s \in \mathcal{S}$ it holds:

$$\sum_{a \in \mathcal{A}} \pi_{\text{ref}}(a|s) \cdot \frac{|R(s, a) - \hat{R}(s, a)|}{\text{range } R} \overset{!}{\leq} \epsilon.$$

By using our definition of $\hat{R}$ (see Equation (94)), this further simplifies as follows:

$$\sum_{a \in \mathcal{A}} \pi_{\text{ref}}(a|s) \cdot \frac{|R(s, a) - \hat{R}(s, a)|}{\text{range } R} = \pi_{\text{ref}}(a_s|s) \cdot \frac{\hat{R}(s, a_s) - R(s, a_s)}{\text{range } R} \overset{!}{\leq} \epsilon. \tag{96}$$

In the last equation, we were able to drop the absolute value sign because our definition of the constants $c_s$ (see Equation (95)) guarantees that $\hat{R}(s, a_s) \geq R(s, a_s)$.

Next, note that whenever $\hat{R}(s, a_s) = R(s, a_s)$ the left-hand side of Equation (96) cancels out and so the inequality holds trivially. In the following, we will therefore only focus on states where $\hat{R}(s, a_s) > R(s, a_s)$. Note that this allows us to drop the $\max$ statement in the definition of the $c_s$ constants (see Equation (95)).

We continue by upper-bounding the difference $\hat{R}(s, a_s) - R(s, a_s)$. By making use of the following identity:

$$R(s, a_s) \quad = \quad \lambda \cdot \log \left[ \exp \left( \frac{1}{\lambda} \cdot R(s, a_s) \right) \right],$$

we can move the $R(s, a_s)$ term into the logarithm term of the $c_s$ constants, and thereby upper-bounding the difference $\hat{R}(s, a_s) - R(s, a_s)$ as follows:

$$\hat{R}(s, a_s) - R(s, a_s)$$
$$= \quad \lambda \cdot \log \left[ \frac{\sum_{a \neq a_s} (R(s, a) - R_L(s)) \cdot \pi_{\mathrm{ref}}(a|s) \cdot \exp \left( \frac{1}{\lambda} \cdot R(s, a) \right)}{(R_L(s) - R(s, a_s)) \cdot \pi_{\mathrm{ref}}(a_s|s) \cdot \exp \left( \frac{1}{\lambda} \cdot R(s, a_s) \right)} \right]$$
$$\leq \quad \lambda \cdot \log \left[ \frac{(\max_{a \in \mathcal{A}} R(s, a) - R_L(s)) \cdot \exp \left( \frac{1}{\lambda} \cdot \max_{a \in \mathcal{A}} R(s, a) \right)}{(R_L(s) - R(s, a_s)) \cdot \pi_{\mathrm{ref}}(a_s|s) \cdot \exp \left( \frac{1}{\lambda} \cdot R(s, a_s) \right)} \right]$$
$$\leq \quad \lambda \cdot \log \left[ \frac{L \cdot \mathrm{range}\, R \cdot \exp \left( \frac{1}{\lambda} \cdot \left[ \max_{a \in \mathcal{A}} R(s, a) - R(s, a_s) \right] \right)}{(R_L(s) - R(s, a_s)) \cdot \pi_{\mathrm{ref}}(a_s|s)} \right]$$
$$\leq \quad \lambda \cdot \log \left[ \frac{L \cdot \mathrm{range}\, R \cdot \exp \left( \frac{1}{\lambda} \cdot \mathrm{range}\, R \right)}{(R_L(s) - R(s, a_s)) \cdot \pi_{\mathrm{ref}}(a_s|s)} \right]$$
$$\leq \quad \lambda \cdot \sqrt{\frac{L \cdot \mathrm{range}\, R \cdot \exp \left( \frac{1}{\lambda} \cdot \mathrm{range}\, R \right)}{(R_L(s) - R(s, a_s)) \cdot \pi_{\mathrm{ref}}(a_s|s)}}$$

We can now put this upper bound back into Equation (96) and convert the inequality into an upper bound for $\pi_{\mathrm{ref}}(a_s|s)$ as follows:

$$\pi_{\mathrm{ref}}(a_s|s) \cdot \frac{\hat{R}(s, a_s) - R(s, a_s)}{\mathrm{range}\, R}$$
$$\leq \quad \frac{\pi_{\mathrm{ref}}(a_s|s)}{\mathrm{range}\, R} \cdot \lambda \cdot \sqrt{\frac{L \cdot \mathrm{range}\, R \cdot \exp \left( \frac{1}{\lambda} \cdot \mathrm{range}\, R \right)}{(R_L(s) - R(s, a_s)) \cdot \pi_{\mathrm{ref}}(a_s|s)}}$$
$$= \quad \frac{1}{\sqrt{\mathrm{range}\, R}} \cdot \lambda \cdot \sqrt{\frac{\pi_{\mathrm{ref}}(a_s|s) \cdot L \cdot \exp \left( \frac{1}{\lambda} \cdot \mathrm{range}\, R \right)}{R_L(s) - R(s, a_s)}} \quad \overset{!}{\leq} \quad \epsilon$$
$$\implies \quad \pi_{\mathrm{ref}}(a_s|s) \quad \leq \quad \frac{R_L(s) - R(s, a_s)}{L} \cdot \frac{\mathrm{range}\, R}{\exp \left( \frac{1}{\lambda} \cdot \mathrm{range}\, R \right)} \cdot \frac{\epsilon^2}{\lambda^2}.$$

The last line in the previous derivation holds by assumption of the proposal. That was to show. $\qquad\square$

## C.5. A regularized negative result for general MDPs

Throughout, let $\langle \mathcal{S}, \mathcal{A}, \tau, \mu_0, R, \gamma \rangle$ be an MDP. Additionally, assume there to be a data distribution $D \in \Delta(\mathcal{S} \times \mathcal{A})$ used for learning the reward function. We do a priori *not assume* that $D$ is induced by a reference policy, but we will specialize to that case later on.

We also throughout fix $\epsilon > 0, \lambda > 0, L \in (0, 1)$, which will represent, respectively, an approximation-error for the reward function, the regularization strength, and a lower regret bound. Furthermore, let $\omega : \Pi \to \mathbb{R}$ be any continuous regularization function of policies with $\omega(\pi) \geq 0$ for all $\pi \in \Pi$. For example, if there is a nowhere-zero reference policy $\pi_{\mathrm{ref}}$, then $\omega$ could be given by $\omega(\pi) = \mathbb{D}_{\mathrm{KL}} (\pi || \pi_{\mathrm{ref}})$. For any reward function $\hat{R}$, a policy $\hat{\pi}$ exists that is optimal with respect to regularized

maximization of reward:

$$\hat{\pi} \in \arg\max_{\pi} J_{\hat{R}}(\pi) - \lambda\omega(\pi).$$

We will try to answer the following question: Do there exist realistic conditions on $\omega$ and $D$ for which there exists $\hat{R}$ together with $\hat{\pi}$ such that the following properties hold?

- $\mathbb{E}_{(s,a) \sim D}\left[\frac{|\hat{R}(s,a) - R(s,a)|}{\text{range } R}\right] \leq \epsilon.$

- $\text{Reg}^R(\hat{\pi}) \geq L.$

Furthermore, we now fix $\pi_*$, a worst-case policy for $R$, meaning that $\text{Reg}^R(\pi_*) = 1$. We assume $\pi_*$ to be deterministic.

**Lemma C.34.** *Define $C(L, R) \coloneqq \frac{(1-L)\cdot\text{range } J_R}{\|R\|}$. Then the following implication holds:*

$$\|D^{\pi} - D^{\pi_*}\| \leq C(L, R) \quad \implies \quad \text{Reg}^R(\pi) \geq L.$$

*Proof.* Using the Cauchy-Schwarz inequality, the left side of the implication implies:

$$\begin{aligned}
J_R(\pi) - \min J_R &= J_R(\pi) - J_R(\pi_*) \\
&= \left(D^{\pi} - D^{\pi_*}\right) \cdot R \\
&\leq \|D^{\pi} - D^{\pi_*}\| \cdot \|R\| \\
&\leq (1 - L) \cdot \text{range } J_R.
\end{aligned}$$

By subtracting $\text{range } J_R = \max J_R - \min J_R$ from both sides, then multiplying by $-1$, and then dividing by $\text{range } R$, we obtain the result. $\square$

**Lemma C.35.** *For any $(s, a)$, we have*

$$\frac{D^{\pi}(s, a)}{1 - \gamma} = \sum_{t=0}^{\infty} \gamma^t \sum_{s_0, a_0, \ldots, s_{t-1}, a_{t-1}} \tau(s_0, a_0, \ldots, s_{t-1}, a_{t-1}, s) \cdot \pi(s_0, a_0, \ldots, s_{t-1}, a_{t-1}, s, a),$$

*where*

$$\tau(s_0, a_0, \ldots, s) \coloneqq \mu_0(s_0) \cdot \left[\prod_{i=1}^{t-1} \tau(s_i \mid s_{i-1}, a_{i-1})\right] \cdot \tau(s \mid s_{t-1}, a_{t-1}),$$

*which is the part in the probability of a trajectory that does not depend on the policy, and*

$$\pi(s_0, a_0, \ldots, s, a) \coloneqq \pi(a \mid s) \cdot \prod_{i=0}^{t-1} \pi(a_i \mid s_i).$$

*Proof.* We have

$$\begin{aligned}
\frac{D^{\pi}(s, a)}{1 - \gamma} &= \sum_{t=0}^{\infty} \gamma^t P(s_t = s, a_t = a \mid \xi \sim \pi) \\
&= \sum_{t=0}^{\infty} \gamma^t \sum_{s_0, a_0, \ldots, s_{t-1}, a_{t-1}} P(s_0, a_0, \ldots, s_{t-1}, a_{t-1}, s, a \mid \pi) \\
&= \sum_{t=0}^{\infty} \gamma^t \sum_{s_0, a_0, \ldots, s_{t-1}, a_{t-1}} \mu_0(s_0)\pi(a_0 \mid s_0) \left[\prod_{i=1}^{t-1} \tau(s_i \mid s_{i-1}, a_{i-1})\pi(a_i \mid s_i)\right] \tau(s \mid s_{t-1}, a_{t-1})\pi(a \mid s) \\
&= \sum_{t=0}^{\infty} \gamma^t \sum_{s_0, a_0, \ldots, s_{t-1}, a_{t-1}} \tau(s_0, a_0, \ldots, s_{t-1}, a_{t-1}, s) \cdot \pi(s_0, a_0, \ldots, s_{t-1}, a_{t-1}, s, a).
\end{aligned}$$

$\square$

**Lemma C.36.** *Let $1 \geq \delta > 0$. Assume that $\pi(a \mid s) \geq 1 - \delta$ for all $(s, a) \in \mathrm{supp}\, D^{\pi_*}$ and that $\pi_*$ is a deterministic policy.[4] Then for all $(s, a) \in \mathcal{S} \times \mathcal{A}$, one has*

$$D^{\pi_*}(s, a) - \delta \cdot (1 - \gamma) \cdot \frac{\partial}{\partial \gamma} \left( \frac{\gamma}{1 - \gamma} D^{\pi_*}(s, a) \right) \leq D^{\pi}(s, a) \leq D^{\pi_*}(s, a) + \frac{\delta}{1 - \gamma}. \tag{97}$$

*This also results in the following two inequalities:*

$$D^{\pi}(\mathrm{supp}\, D^{\pi_*}) \geq 1 - \frac{\delta}{1 - \gamma}, \quad \|D^{\pi} - D^{\pi_*}\| \leq \sqrt{|\mathcal{S} \times \mathcal{A}|} \cdot \frac{\delta}{1 - \gamma}. \tag{98}$$

*Proof.* Let $(s, a) \in \mathrm{supp}\, D^{\pi_*}$. We want to apply the summation formula in Lemma C.35, which we recommend to recall. For simplicity, in the following we will write $s_0, a_0, \dots$ when we implicitly mean trajectories up until $s_{t-1}, a_{t-1}$. Now, we will write "$\pi_*$-comp" into a sum to indicate that we only sum over states and actions that make the whole trajectory-segment *compatible* with policy $\pi_*$, meaning all transitions have positive probability and the actions are deterministically selected by $\pi_*$. Note that if we restrict to such summands, then each consecutive pair $(s_i, a_i) \in \mathrm{supp}\, D^{\pi_*}$ is in the support of $D^{\pi_*}$, and thus we can use our assumption $\pi(a_i \mid s_i) \geq 1 - \delta$ on those. We can use this strategy for a lower-bound:

$$\begin{aligned}
\frac{D^{\pi}(s, a)}{1 - \gamma} &\geq \sum_{t=0}^{\infty} \gamma^t \sum_{\substack{s_0, a_0, \dots \\ \pi_* - \mathrm{comp}}} \tau(s_0, a_0, \dots, s) \cdot \pi(s_0, a_0, \dots, s, a) \\
&\geq \sum_{t=0}^{\infty} \gamma^t \sum_{\substack{s_0, a_0, \dots \\ \pi_* - \mathrm{comp}}} \tau(s_0, a_0, \dots, s) \cdot (1 - \delta)^{t+1} \\
&\geq \sum_{t=0}^{\infty} \gamma^t \sum_{\substack{s_0, a_0, \dots \\ \pi_* - \mathrm{comp}}} \tau(s_0, a_0, \dots, s) \cdot \left( 1 - \delta \cdot (t + 1) \right).
\end{aligned} \tag{99}$$

In the last step, we used the classical formula $(1 - \delta)^t \geq 1 - \delta \cdot t$, which can easily be proved by induction over $t$. Now, we split the sum up into two parts. For the first part, we note:

$$\begin{aligned}
\sum_{t=0}^{\infty} \gamma^t \sum_{\substack{s_0, a_0, \dots \\ \pi_* - \mathrm{comp}}} \tau(s_0, a_0, \dots, s) \cdot 1 &= \sum_{t=0}^{\infty} \gamma^t \sum_{\substack{s_0, a_0, \dots \\ \pi_* - \mathrm{comp}}} \tau(s_0, a_0, \dots, s) \cdot \pi_*(s_0, a_0, \dots, s, a) \\
&= \sum_{t=0}^{\infty} \gamma^t \sum_{s_0, a_0, \dots} \tau(s_0, a_0, \dots, s) \cdot \pi_*(s_0, a_0, \dots, s, a) \\
&= \frac{D^{\pi_*}(s, a)}{1 - \gamma}.
\end{aligned} \tag{100}$$

For the second part, we similarly compute:

$$\begin{aligned}
\sum_{t=0}^{\infty} (t + 1) \gamma^t \sum_{\substack{s_0, a_0, \dots \\ \pi_* - \mathrm{comp}}} \tau(s_0, a_0, \dots, s) &= \sum_{t=0}^{\infty} \frac{\partial}{\partial \gamma} \gamma^{t+1} P(s_t = s, a_t = a \mid \pi_*) \\
&= \frac{\partial}{\partial \gamma} \left( \frac{\gamma}{1 - \gamma} \cdot D^{\pi_*}(s, a) \right).
\end{aligned} \tag{101}$$

Putting Equations (100) and (101) into Equation (99) gives the first equation of Equation (97) for the case that $(s, a) \in \mathrm{supp}\, D^{\pi_*}$. For the case that $(s, a) \notin \mathrm{supp}\, D^{\pi_*}(s, a)$, the inequality is trivial since then $D^{\pi_*}(s, a) = 0$ and since the stated derivative is easily shown to be non-negative by writing out the occupancy explicitly (i.e., by reversing the previous computation).

---

[4] In this lemma, one does not need the assumption that $\pi_*$ is a worst-case policy, but this case will be the only application later on.

This then implies

$$
\begin{aligned}
D^\pi(\text{supp } D^{\pi*}) &= \sum_{(s,a)\in\text{supp } D^{\pi*}} D^\pi(s,a) \\
&\geq \sum_{(s,a)\in\text{supp } D^{\pi*}} \left( D^{\pi*}(s,a) - \delta\cdot(1-\gamma)\cdot\frac{\partial}{\partial\gamma}\left(\frac{\gamma}{1-\gamma}D^{\pi*}(s,a)\right)\right) \\
&= 1 - \delta\cdot(1-\gamma)\cdot\frac{\partial}{\partial\gamma}\left(\frac{\gamma}{1-\gamma}\sum_{(s,a)\in\text{supp } D^{\pi*}} D^{\pi*}(s,a)\right) \\
&= 1 - \delta\cdot(1-\gamma)\cdot\frac{1}{(1-\gamma)^2} \\
&= 1 - \frac{\delta}{1-\gamma}.
\end{aligned}
$$

This shows the first inequality in Equation (98). To show the second inequality in Equation (97), we use the first one and compute:

$$
\begin{aligned}
D^\pi(s,a) &= 1 - \sum_{(s',a')\neq(s,a)} D^\pi(s',a') \\
&\leq 1 - \sum_{(s',a')\in\text{supp } D^{\pi*}\setminus\{(s,a)\}} D^\pi(s',a') \\
&\leq 1 - \sum_{(s',a')\in\text{supp } D^{\pi*}\setminus\{(s,a)\}} D^{\pi*}(s',a') \\
&\quad + \sum_{(s',a')\in\text{supp } D^{\pi*}\setminus\{(s,a)\}} \delta\cdot(1-\gamma)\cdot\frac{\partial}{\partial\gamma}\left(\frac{\gamma}{1-\gamma}D^{\pi*}(s',a')\right) \\
&\leq D^{\pi*}(s,a) + \frac{\delta}{1-\gamma},
\end{aligned}
$$

where in the last step we again used the trick of the previous computation of pulling the sum through the derivative. Finally, we prove the second inequality in Equation (98), using what we know so far. First, note that

$$
\delta\cdot(1-\gamma)\cdot\frac{\partial}{\partial\gamma}\left(\frac{\gamma}{1-\gamma}D^{\pi*}(s,a)\right) \leq \frac{\delta}{1-\gamma}
$$

since we showed that the left-hand-side is non-negative and sums to the right-hand-side over all $(s,a)$. Consequently, we obtain:

$$
\begin{aligned}
\|D^\pi - D^{\pi*}\| &= \sqrt{\sum_{(s,a)}\left(D^\pi(s,a)-D^{\pi*}(s,a)\right)^2} \\
&\leq \sqrt{\sum_{(s,a)}\left|\frac{\delta}{1-\gamma}\right|^2} \\
&= \sqrt{|\mathcal{S}\times\mathcal{A}|}\cdot\frac{\delta}{1-\gamma}.
\end{aligned}
$$

This finishes the proof. $\qquad\square$

We now fix more constants and notation. Define $\mathcal{S}_0 := \text{supp } \mu_0$ as the support of $\mu_0$, and more generally $\mathcal{S}_t$ as the states reachable within $t$ timesteps using the fixed worst-case policy $\pi_*$:

$$
\mathcal{S}_t := \left\{ s \mid \exists \pi_* - \text{compatible sequence } s_0, a_0, \ldots, s_{k-1}, a_{k-1}, s \text{ for } k \leq t \right\}.
$$

Since there are only finitely many states and $\mathcal{S}_t \subseteq \mathcal{S}_{t+1}$, there is a $t_0$ such that $\mathcal{S}_{t_0}$ is maximal. Set $D^{\pi_*}(s) := \sum_a D^{\pi_*}(s, a)$. Recall the notation $\tau$ from Lemma C.35. Define the following constant which, given the MDP, only depends on $\delta > 0$ and $\pi_*$:

$$C(\delta, \pi_*, \mu_0, \tau, \gamma) := \min_{\substack{t \in [0:t_0] \\ s_0, a_0, \ldots, s_{t-1}, a_{t-1}, s: \, \pi_* - \text{comp}}} \gamma^t \tau(s_0, a_0 \ldots, s) \cdot (1 - \delta)^t \cdot \delta > 0. \tag{102}$$

We get the following result:

**Lemma C.37.** *Define the reward function $\hat{R} : \mathcal{S} \times \mathcal{A} \to \mathbb{R}$ as follows:*

$$\hat{R}(s, a) := \begin{cases} R(s, a), & (s, a) \notin \text{supp } D^{\pi_*}, \\ \max R + \frac{\lambda}{C(\delta, \pi_*, \mu_0, \tau, \gamma)} \cdot \omega(\pi_*), & \text{else.} \end{cases} \tag{103}$$

*Assume that $\hat{\pi}$ is $(\lambda, \omega)$-RLHF optimal with respect to $\hat{R}$. Then for all $(s, a) \in \text{supp } D^{\pi_*}$, we have $\hat{\pi}(a \mid s) \geq 1 - \delta$.*

*Proof.* We show this statement by induction over the number of timesteps that $\pi_*$ needs to reach a given state. Thus, first assume $s \in \mathcal{S}_0$ and $a = \pi_*(s)$. We do a proof by contradiction. Thus, assume that $\hat{\pi}(a \mid s) < 1 - \delta$. This means that $\sum_{a' \neq a} \hat{\pi}(a' \mid s) \geq \delta$, and consequently

$$\sum_{a' \neq a} D^{\hat{\pi}}(s, a') \geq \mu_0(s) \cdot \delta \geq C(\delta, \pi_*, \mu_0, \tau, \gamma). \tag{104}$$

We now claim that from this it follows that $\pi_*$ is more optimal than $\hat{\pi}$ with respect to RLHF, a contradiction to the optimality of $\hat{\pi}$. Indeed:

$$
\begin{aligned}
J_{\hat{R}}(\hat{\pi}) - \lambda \omega(\hat{\pi}) &\overset{(1)}{\leq} J_{\hat{R}}(\hat{\pi}) \\
&\overset{(2)}{=} \sum_{a' \neq a} D^{\hat{\pi}}(s, a') \cdot R(s, a') + \sum_{(s', a') \notin \{s\} \times \mathcal{A} \setminus \{a\}} D^{\hat{\pi}}(s', a') \cdot \hat{R}(s', a') \\
&\overset{(3)}{\leq} \sum_{a' \neq a} D^{\hat{\pi}}(s, a') \cdot \max R + \hat{R}(s, a) \cdot \sum_{(s', a') \notin \{s\} \times \mathcal{A} \setminus \{a\}} D^{\hat{\pi}}(s', a'') \\
&= \sum_{a' \neq a} D^{\hat{\pi}}(s, a') \cdot \max R + \left(1 - \sum_{a' \neq a} D^{\hat{\pi}}(s, a')\right) \cdot \hat{R}(s, a) \\
&\overset{(4)}{\leq} C(\delta, \pi_*, \mu_0, \tau, \gamma) \cdot \max R + \left(1 - C(\delta, \pi_*, \mu_0, \tau, \gamma)\right) \cdot \hat{R}(s, a) \\
&\overset{(5)}{=} J_{\hat{R}}(\pi_*) + C(\delta, \pi_*, \mu_0, \tau, \gamma) \cdot \left(\max R - \hat{R}(s, a)\right) \\
&\overset{(6)}{=} J_{\hat{R}}(\pi_*) - C(\delta, \pi_*, \mu_0, \tau, \gamma) \cdot \frac{\lambda}{C(\delta, \pi_*, \mu_0, \tau, \gamma)} \cdot \omega(\pi_*) \\
&= J_{\hat{R}}(\pi_*) - \lambda \omega(\pi_*).
\end{aligned}
\tag{105}
$$

In step (1), we use the non-negativity of $\omega$. In step (2), we use that $(s, a') \notin \text{supp } D^{\pi_*}$, and so $\hat{R}(s, a') = R(s, a')$. In the right term in step (3), we use that $(s, a) \in \text{supp } D^{\pi_*}$, and thus $\hat{R}(s, a) \geq \hat{R}(s', a')$, by definition of $\hat{R}$. In step (4), we use that $\hat{R}(s, a) \geq \max R$ and Equation (104). Step (5) uses that $J_{\hat{R}}(\pi_*) = \hat{R}(s, a)$, following from the fact that $\hat{R}$ is constant for policy $\pi_*$. Step (6) uses the concrete definition of $\hat{R}$. Thus, we have showed a contradiction to the RLHF-optimality of $\hat{\pi}$, from which it follows that $\hat{\pi}(a \mid s) \geq 1 - \delta$.

Now assume the statement is already proven for $t - 1$ and let $s \in \mathcal{S}_t \setminus \mathcal{S}_{t-1}$. Then there exists a $\pi_*$-compatible sequence $s_0, a_0, \ldots, s_{t-1}, a_{t-1}$ leading to $s$. We necessarily have $s_i \in \mathcal{S}_i$ for all $i = 0, \ldots, t-1$, and so we obtain $\hat{\pi}(a_i \mid s_i) \geq 1 - \delta$ by the induction hypothesis. Now, let $a := \pi_*(s)$ and assume we had $\hat{\pi}(a \mid s) < 1 - \delta$. As before, we then have

$\sum_{a' \neq a} \hat{\pi}(a' \mid s) \geq \delta$. Consequently, we get

$$\sum_{a' \neq a} D^{\hat{\pi}}(s, a') \geq \gamma^t \cdot \sum_{a' \neq a} \tau(s_0, a_0, \ldots, s) \cdot \hat{\pi}(s_0, a_0, \ldots, s, a')$$
$$\geq \gamma^t \cdot \tau(s_0, a_0, \ldots, s) \cdot (1 - \delta)^t \cdot \delta$$
$$\geq C(\delta, \pi_*, \mu_0, \tau, \gamma)$$

Then the same computation as in Equation (105) leads to the same contradiction again, and we are done. $\qquad\square$

**Theorem C.38.** *Define*

$$\delta := \frac{(1 - \gamma) \cdot (1 - L) \cdot \operatorname{range} J_R}{\sqrt{|\mathcal{S} \times \mathcal{A}| \cdot \|R\|}} > 0.$$

*Let* $\mathcal{M} = \langle \mathcal{S}, \mathcal{A}, \tau, \mu_0, R, \gamma \rangle$ *be our MDP. Set*

$$C := C(\mathcal{M}, \pi_*, L, \lambda, \omega) := \frac{\lambda \cdot \omega(\pi_*)}{\operatorname{range} R \cdot C(\delta, \pi_*, \mu_0, \tau, \gamma)} < \infty, \tag{106}$$

*with the "inner"* $C(\delta, \pi_*, \mu_0, \tau, \gamma)$ *defined in Equation* (102). *Assume that*

$$D(\operatorname{supp} D^{\pi_*}) \leq \frac{\epsilon}{1 + C}. \tag{107}$$

*Then* $D \in \mathbf{unsafe}(R, \epsilon, L, \lambda, \omega)$.

*Proof.* We prove the theorem by showing that for every data distribution $D \in \Delta(\mathcal{S} \times \mathcal{A})$ that fulfills the conditions of Theorem C.38, there exists a reward function $\hat{R}$ together with a $(\lambda, \omega)$-RLHF optimal policy $\hat{\pi}$ with respect to $\hat{R}$ such that

- $\mathbb{E}_{(s,a) \sim D} \left[ \frac{|\hat{R}(s,a) - R(s,a)|}{\operatorname{range} R} \right] \leq \epsilon$,

- $\operatorname{Reg}^R(\hat{\pi}) \geq L$.

Towards that goal, define $\hat{R}$ as in Equation (103) and $\hat{\pi}$ as a $(\lambda, \omega)$-RLHF optimal policy for $\hat{R}$. Then Theorem C.37 shows that $\hat{\pi}(s \mid a) \geq 1 - \delta$ for all $(s, a) \in \operatorname{supp} D^{\pi_*}$. Consequently, Theorem C.36 implies that

$$\|D^{\hat{\pi}} - D^{\pi_*}\| \leq \sqrt{|\mathcal{S} \times \mathcal{A}|} \cdot \frac{\delta}{1 - \gamma} = \frac{(1 - L) \cdot \operatorname{range} J_R}{\|R\|}.$$

Consequently, Theorem C.34 shows that $\operatorname{Reg}^R(\hat{\pi}) \geq L$, and thus the second claim. For the first claim, note that

$$\mathbb{E}_{(s,a) \sim D} \left[ |\hat{R}(s, a) - R(s, a)| \right] = \sum_{(s,a) \in \operatorname{supp} D^{\pi_*}} D(s, a) \cdot \left( \max R + \frac{\lambda}{C(\delta, \pi_*, \mu_0, \tau, \gamma)} \omega(\pi_*) - R(s, a) \right)$$
$$\leq D(\operatorname{supp} D^{\pi_*}) \cdot \left( \operatorname{range} R + \frac{\lambda}{C(\delta, \pi_*, \mu_0, \tau, \gamma)} \omega(\pi_*) \right)$$
$$\leq \epsilon \cdot \operatorname{range} R,$$

where the last claim follows from the assumed inequality in $D(\operatorname{supp} D^{\pi_*})$. $\qquad\square$

We obtain the following corollary, which is very similar to Theorem C.5. The main difference is that the earlier result only assumed a poliy of regret $L$ and not regret 1:

**Corollary C.39.** *Theorem C.38 specializes as follows for the case* $\lambda = 0$: *Assume* $D(\operatorname{supp} D^{\pi_*}) \leq \epsilon$. *Then there exists a reward function* $\hat{R}$ *together with an optimal policy* $\hat{\pi}$ *that satisfies the two inequalities from the previous result.*

*Proof.* This directly follows from $\lambda = 0$. For completeness, we note that the definition of $\hat{R}$ also simplifies, namely to

$$\hat{R}(s,a) = \begin{cases} R(s,a), & (s,a) \notin \text{supp } D^{\pi_*} \\ \max R, & \text{else.} \end{cases}$$

$\square$

We now present another specialization of Theorem C.38. Namely, from now on, assume that $D = D^{\pi_{\text{ref}}}$ and $\omega(\pi) = \mathbb{D}_{\text{KL}}(\pi||\pi_{\text{ref}})$. In other words, the dataset used to evaluate the reward function is sampled from the same (safe) policy used in KL-regularization. This leads to the following condition specializing the one from Equation (107):

$$D^{\pi_{\text{ref}}}(\text{supp } D^{\pi_*}) \leq \frac{\epsilon}{1 + \frac{\lambda \cdot \mathbb{D}_{\text{KL}}(\pi_*||\pi_{\text{ref}})}{\text{range } R \cdot C(\delta, \pi_*, \mu_0, \tau, \gamma)}}. \tag{108}$$

$\pi_{\text{ref}}$ now appears on both the left and right side of the equation, and so one can wonder whether it is ever possible that the inequality holds. After all, if $D^{\pi_{\text{ref}}}(\text{supp } D^{\pi_*})$ "gets smaller", then $\mathbb{D}_{\text{KL}}(\pi_*||\pi_{\text{ref}})$ should usually get "larger". However, halfing each of the probabilities $D^{\pi_{\text{ref}}}(s,a)$ for $(s,a) \in \text{supp } D^{\pi_*}$ leads to only an increase by the addition of $\log 2$ of $\mathbb{D}_{\text{KL}}(\pi_*||\pi_{\text{ref}})$. Thus, intuitively, we expect the inequality to hold when the left-hand-side is very small. An issue is that the KL divergence can disproportionally blow up in size if some *individual* probabilities $D^{\pi_{\text{ref}}}(s,a)$ for $(s,a) \in \text{supp } D^{\pi_*}$ are very small compared to other such probabilities. This can be avoided by a bound in the proportional difference of these probabilities. We thus obtain the following sufficient condition for a "negative result":[5]

**Corollary C.40.** *Let the notation be as in Theorem C.38 and assume $D = D^{\pi_{\text{ref}}}$ and $\omega(\pi) = \mathbb{D}_{KL}(\pi||\pi_{\text{ref}})$. Let $K \geq 0$ be a constant such that*

$$\max_{(s,a) \in \text{supp } D^{\pi_*}} D^{\pi_{\text{ref}}}(s,a) \leq K \cdot \min_{(s,a) \in \text{supp } D^{\pi_*}} D^{\pi_{\text{ref}}}(s,a).$$

*Assume that*

$$\min_{(s,a) \in \text{supp } D^{\pi_*}} D^{\pi_{\text{ref}}}(s,a) \leq \left( \frac{\epsilon}{K \cdot |\mathcal{S}| \cdot \left( 1 + \frac{\lambda}{\text{range } R \cdot C(\delta, \pi_*, \mu_0, \tau, \gamma)} \right)} \right)^2. \tag{109}$$

*Then Equation (107) holds, and the conclusion of the theorem thus follows.*

*Proof.* As argued before, the equation to show can be written as Equation (108). We can upper-bound the left-hand-side as follows:

$$\begin{aligned} D^{\pi_{\text{ref}}}(\text{supp } D^{\pi_*}) &= \sum_{(s,a) \in \text{supp } D^{\pi_*}} D^{\pi_{\text{ref}}}(s,a) \\ &\leq |\text{supp } D^{\pi_*}| \cdot \max_{(s,a) \in \text{supp } D^{\pi_*}} D^{\pi_{\text{ref}}}(s,a) \\ &\leq |\mathcal{S}| \cdot K \cdot \min_{(s,a) \in \text{supp } D^{\pi_*}} D^{\pi_{\text{ref}}}(s,a). \end{aligned} \tag{110}$$

In one step, we used that $\pi_*$ is assumed to be deterministic, which leads to a bound in the size of the support. Now, we lower-bound the other side by noting that

$$\begin{aligned} \mathbb{D}_{\text{KL}}(\pi_*||\pi_{\text{ref}}) &= \sum_{(s,a) \in \text{supp } D^{\pi_*}} D^{\pi_*}(s,a) \cdot \log \frac{D^{\pi_*}(s,a)}{D^{\pi_{\text{ref}}}(s,a)} \\ &\leq \sum_{(s,a) \in \text{supp } D^{\pi_*}} D^{\pi_*}(s,a) \cdot \log \frac{1}{\min_{(s',a') \in \text{supp } D^{\pi_*}} D^{\pi_{\text{ref}}}(s',a')} \\ &= \log \frac{1}{\min_{(s,a) \in \text{supp } D^{\pi_*}} D^{\pi_{\text{ref}}}(s,a)}. \end{aligned}$$

---

[5]The condition is quite strong and we would welcome attempts to weaken it.

Thus, for the right-hand-side, we obtain

$$\frac{\epsilon}{1 + \frac{\lambda \cdot \mathbb{D}_{\mathrm{KL}}(\pi_* || \pi_{\mathrm{ref}})}{\mathrm{range}\ R \cdot C(\delta, \pi_*, \mu_0, \tau, \gamma)}} \geq \frac{\epsilon}{1 + \frac{\lambda}{\mathrm{range}\ R \cdot C(\delta, \pi_*, \mu_0, \tau, \gamma)} \cdot \log \frac{1}{\min_{(s,a) \in \mathrm{supp}\ D^{\pi_*}} D^{\pi_{\mathrm{ref}}}(s,a)}} \tag{111}$$

Now, set $A := |\mathcal{S}| \cdot K$, $B := \frac{\lambda}{\mathrm{range}\ R \cdot C(\delta, \pi_*, \mu_0, \tau, \gamma)}$ and $x := \min_{(s,a) \in \mathrm{supp}\ D^{\pi_*}} D^{\pi_{\mathrm{ref}}}(s,a)$. Then comparing with Equations (110) and (111), we are left with showing the following, which we also equivalently rewrite:

$$A \cdot x \leq \frac{\epsilon}{1 + B \cdot \log \frac{1}{x}}$$
$$\Longleftrightarrow A \cdot \left( x + Bx \log \frac{1}{x} \right) \leq \epsilon.$$

Now, together with the assumed condition on $x$ from Equation (109), and upper-bounding the logarithm with a square-root, and $x$ by $\sqrt{x}$ since $x \leq 1$, we obtain:

$$\begin{aligned}
A \cdot \left( x + Bx \log \frac{1}{x} \right) &\leq A \cdot \left( x + B\sqrt{x} \right) \\
&\leq A \cdot \left( (1 + B) \cdot \sqrt{x} \right) \\
&\leq A \cdot (1 + B) \cdot \frac{\epsilon}{A \cdot (1 + B)} \\
&= \epsilon.
\end{aligned}$$

That was to show. $\qquad\square$

## D. Requirements for safe optimization

In this section, we answer the question under which circumstances we can guarantee a safe optimization of a given reward function. Wherever applicable, we make the same assumptions as stated in Appendix C.1.

### D.1. Applying Berge's maximum theorem

**Definition D.1** (Correspondence). Let $X, Y$ be two sets. A *correspondence* $C : X \rightrightarrows Y$ is a function $X \to \mathcal{P}(Y)$ from $X$ to the power set of $Y$.

**Definition D.2** (Upper Hemicontinuous, Lower Hemicontinuous, Continuous, Compact-Valued). Let $C : X \rightrightarrows Y$ be a correspondence where $X$ and $Y$ are topological spaces. Then:

- $C$ is called *upper hemicontinuous* if for every $x \in X$ and every open set $V \subseteq Y$ with $C(x) \subseteq V$, there exists an open set $U \subseteq X$ with $x \in U$ and such that for all $x' \in U$ one has $C(x') \subseteq V$.

- $C$ is called *lower hemicontinuous* if for every $x \in X$ and every open set $V \subseteq Y$ with $C(x) \cap V \neq \emptyset$, there exists an open set $U \subseteq X$ with $x \in U$ and such that for all $x' \in U$ one has $C(x') \cap V \neq \emptyset$.

- $C$ is called *continuous* if it is both upper and lower hemicontinuous.

- $C$ is called *compact-valued* if $C(x)$ is a compact subset of $Y$ for all $x \in X$.

**Theorem D.3** (Maximum Theorem, (Berge, 1963)). *Let $\Theta$ and $X$ be topological spaces, $f : \Theta \times X \to \mathbb{R}$ a continuous function, and $C : \Theta \rightrightarrows X$ be a continuous, compact-valued correspondence such that $C(\theta) \neq \emptyset$ for all $\theta \in \Theta$. Define the optimal value function $f^* : \Theta \to \mathbb{R}$ by*

$$f^*(\theta) := \max_{x \in C(\theta)} f(\theta, x)$$

*and the* maximizer function $C^* : \Theta \rightrightarrows X$ *by*

$$C^*(\theta) := \underset{x \in C(\theta)}{\arg\max}\ f(\theta, x) = \left\{ x \in C(\theta) \mid f(\theta, x) = f^*(\theta) \right\}.$$

*Then $f^*$ is continuous and $C^*$ is a compact-valued, upper hemicontinuous correspondence with nonempty values, i.e. $C^*(\theta) \neq \emptyset$ for all $\theta \in \Theta$.*

We now show that this theorem corresponds to our setting. Namely, replace $X$ be by $\Pi$, the set of all policies. Every policy $\pi \in \Pi$ can be viewed as a vector $\vec{\pi} = \big(\pi(a \mid s)\big)_{s \in \mathcal{S}, a \in \mathcal{A}} \in \mathbb{R}^{\mathcal{S} \times \mathcal{A}}$, and so we view $\Pi$ as a subset of $\mathbb{R}^{\mathcal{S} \times \mathcal{A}}$. $\Pi$ inherits the standard Euclidean metric and thus topology from $\mathbb{R}^{\mathcal{S} \times \mathcal{A}}$. Replace $\Theta$ by $\mathcal{R}$, the set of all reward functions. We can view each reward function $R \in \mathcal{R}$ as a vector $\vec{R} = \big(R(s, a)\big)_{(s,a) \in \mathcal{S} \times \mathcal{A}} \in \mathbb{R}^{\mathcal{S} \times \mathcal{A}}$. So we view $\mathcal{R}$ as a subset of $\mathbb{R}^{\mathcal{S} \times \mathcal{A}}$ and thus a topological space. Replace $f$ by the function $J : \mathcal{R} \times \Pi \to \mathbb{R}$ given by

$$J(R, \pi) \coloneqq J^R(\pi) = \eta^\pi \cdot \vec{R}.$$

Take as the correspondence $C : \mathcal{R} \rightrightarrows \Pi$ the trivial function $C(R) \coloneqq \Pi$ that maps every reward function to the full set of policies.

**Proposition D.4.** *These definitions satisfy the conditions of Theorem D.3, that is:*

1. *$J : \mathcal{R} \times \Pi \to \mathbb{R}$ is continuous.*

2. *$C : \mathcal{R} \rightrightarrows \Pi$ is continuous and compact-valued with non-empty values.*

*Proof.* Let us prove 1. Since the scalar product is continuous, it is enough to show that $\eta : \Pi \to \mathbb{R}^{\mathcal{S} \times \mathcal{A}}$ is continuous. Let $(s, a) \in \mathcal{S} \times \mathcal{A}$ be arbitrary. Then it is enough to show that each componentfunction $\eta(s, a) : \Pi \to \mathbb{R}$ given by

$$\big[\eta(s, a)\big](\pi) \coloneqq \eta^\pi(s, a)$$

is continuous.

Now, for any $t \geq 0$, define the function $P_t(s, a) : \Pi \to \mathbb{R}$ by

$$\big[P_t(s, a)\big](\pi) \coloneqq P(s_t = s, a_t = a \mid \xi \sim \pi).$$

We obtain

$$\eta(s, a) = \sum_{t=0}^{\infty} \gamma^t P_t(s, a).$$

Furthermore, this convergence is uniform since $\big[P_t(s, a)\big](\pi) \leq 1$ for all $\pi$ and since $\sum_{t=0}^{\infty} \gamma^t$ is a convergent series. Thus, by the uniform limit theorem, it is enough to show that each $P_t(s, a)$ is a continuous function.

Concretely, we have

$$\big[P_t(s, a)\big](\pi) = \sum_{s_0, a_0, \ldots, s_{t-1}, a_{t-1}} P\big(s_0, a_0, \ldots, s_{t-1}, a_{t-1}, s, a \mid \xi \sim \pi\big)$$

$$= \sum_{s_0, a_0, \ldots, s_{t-1}, a_{t-1}} \mu_0(s_0) \cdot \pi(a_0 \mid s_0) \cdot \left[ \prod_{l=1}^{t-1} \tau(s_l \mid s_{l-1}, a_{l-1}) \cdot \pi(a_l \mid s_l) \right] \cdot \tau(s \mid s_{t-1}, a_{t-1}) \cdot \pi(a \mid s).$$

Since $\mathcal{S}$ and $\mathcal{A}$ are finite, this whole expression can be considered as a polynomial with variables given by all $\pi(a \mid s)$ for all $(s, a) \in \mathcal{S} \times \mathcal{A}$ and coefficients specified by $\mu_0$ and $\tau$. Since polynomials are continuous, this shows the result.

Let us prove 2. Since $\Pi \neq \emptyset$, $C$ has non-empty values. Furthermore, $\Pi$ is compact because it is a finite cartesian product of compact simplices. And finally, since $C$ is constant, it is easily seen to be continuous. That was to show. □

Define the optimal value function $J^* : \mathcal{R} \to \mathbb{R}$ by

$$J^*(R) \coloneqq \max_{\pi \in \Pi} J^R(\pi)$$

and the maximizer function $\Pi^* : \mathcal{R} \rightrightarrows \Pi$ by

$$\Pi^*(R) \coloneqq \arg\max_{\pi \in \Pi} J^R(\pi) = \big\{ \pi \in \Pi \mid J^R(\pi) = J^*(R) \big\}.$$

**Corollary D.5.** $J^*$ *is continuous and* $\Pi^*$ *is upper hemicontinuous and compact-valued with non-empty values.*

*Proof.* This follows from Theorem D.3 and Proposition D.4. □

In particular, every reward function has a compact and non-empty set of optimal policies, and their value changes continuously with the reward function. The most important part of the corollary is the upper hemicontinuity, which has the following consequence:

**Corollary D.6.** *Let* $R$ *be a fixed, non-trivial reward function, meaning that* $\max J^R \neq \min J^R$. *Let* $U \in (0,1]$ *be arbitrary. Then there exists* $\epsilon > 0$ *such that for all* $\hat{R} \in \mathcal{B}_\epsilon(R)$ *and all* $\hat{\pi} \in \Pi^*(\hat{R})$, *we have* $\mathrm{Reg}^R(\hat{\pi}) < U$.

*Proof.* The condition $\max J^R \neq \min J^R$ ensures that the regret function $\mathrm{Reg}^R : \Pi \to [0,1]$ is well-defined. Recall its definition:

$$\mathrm{Reg}^R(\pi) = \frac{\max J^R - J^R(\pi)}{\max J^R - \min J^R}.$$

Since $J^R$ is continuous by Proposition D.4, the regret function $\mathrm{Reg}^R$ is continuous as well. Consequently, the set $V := \left(\mathrm{Reg}^R\right)^{-1}\left([0,U)\right)$ is open in $\Pi$.

Notice that $\Pi^*(R) \subseteq V$ (optimal policies have no regret). Thus, by Corollary D.5, there exists an open set $W \subseteq \mathcal{R}$ with $R \in W$ such that for all $\hat{R} \in W$ we have $\Pi^*(\hat{R}) \subseteq V$. Consequently, for all $\hat{\pi} \in \Pi^*(\hat{R})$, we get $\mathrm{Reg}^R(\hat{\pi}) < U$. Since $W$ is open, it contains a whole epsilon ball around $R$, showing the result. □

Now we translate the results to the distance defined by $D$, a data distribution. Namely, let $D \in \Delta(\mathcal{S} \times \mathcal{A})$ a distribution that assigns a positive probability to each transition. Then define the $D$-norm by

$$d^D(R) := \mathbb{E}_{(s,a) \sim D}\left[\left|R(s,a)\right|\right]. \tag{112}$$

This is indeed a norm, i.e.: for all $\alpha \in \mathbb{R}$ and all $R, R' \in \mathcal{R}$, we have

- $d^D(R + R') \leq d^D(R) + d^D(R')$;

- $d^D(\alpha \cdot R) = |\alpha| \cdot d^D(R)$;

- $d^D(R) = 0$ if and only if $R = 0$.

For the third property, one needs the assumption that $D(s,a) > 0$ for all $(s,a) \in \mathcal{S} \times \mathcal{A}$.

This norm then induces a metric that we denote the same way:

$$d^D(R, R') := d^D(R - R'). \tag{113}$$

We obtain:

**Corollary D.7.** *Let* $\langle \mathcal{S}, \mathcal{A}, \tau, \mu_0, R, \gamma \rangle$ *be an arbitrary non-trivial MDP, meaning that* $\max J^R \neq \min J^R$. *Furthermore, let* $L \in (0,1]$ *be arbitrary, and* $D \in \Delta(\mathcal{S} \times \mathcal{A})$ *a positive data distribution, i.e., a distribution* $D$ *such that* $\forall(s,a) \in \mathcal{S} \times \mathcal{A}$, $D(s,a) > 0$. *Then there exists* $\epsilon > 0$ *such that* $D \in \mathbf{safe}(R, \epsilon, L)$

*Proof.* To prove the corollary, we will show that there exists $\epsilon > 0$ such that for all $\hat{R} \in \mathcal{R}$ with

$$\frac{d^D(R, \hat{R})}{\mathrm{range}\, R} < \epsilon$$

and all $\hat{\pi} \in \Pi^*(\hat{R})$ we have $\mathrm{Reg}^R(\hat{\pi}) < L$. We know from Corollary D.6 that there is $\epsilon' > 0$ such that for all $\hat{R} \in \mathcal{B}_{\epsilon'}(R)$ and all $\hat{\pi} \in \Pi^*(\hat{R})$, we have $\mathrm{Reg}^R(\hat{\pi}) < L$. Now, let $c > 0$ be a constant such that

$$c \cdot \|R' - R''\| \leq d^D(R', R'')$$

for all $R'$, $R'' \in \mathcal{R}$, where $\| \cdot \|$ is the standard Euclidean norm. This exists since all norms in $\mathbb{R}^{\mathcal{S} \times \mathcal{A}}$ are equivalent, but one can also directly argue that

$$c := \min_{(s,a) \in \mathcal{S} \times \mathcal{A}} D(s, a)$$

is a valid choice. Then, set

$$\epsilon := \epsilon' \cdot \frac{c}{\operatorname{range} R}.$$

Then for all $\hat{R} \in \mathcal{R}$ with

$$\frac{d^D(R, \hat{R})}{\operatorname{range} R} < \epsilon$$

we obtain

$$
\begin{aligned}
\|R - \hat{R}\| &\leq \frac{d^D(R, \hat{R})}{c} \\
&= \frac{d^D(R, R')}{\operatorname{range} R} \cdot \frac{\operatorname{range} R}{c} \\
&\leq \epsilon \cdot \frac{\operatorname{range} R}{c} \\
&= \epsilon'.
\end{aligned}
$$

Thus, for all $\hat{\pi} \in \Pi^*(\hat{R})$, we obtain $\operatorname{Reg}^R(\hat{\pi}) < L$, showing the result. $\qquad \square$

*Remark* D.8. If $c := \min_{(s,a) \in \mathcal{S} \times \mathcal{A}} D(s, a)$ is very small, then the proof of the preceding corollary shows that $d^D(R, \hat{R})$ must be correspondingly smaller to guarantee a low regret of $\hat{\pi} \in \Pi^*(\hat{R})$. This makes sense since a large effective distance between $R$ and $\hat{R}$ can "hide" in the regions where $D$ is small when distance is measured via $d^D$.

### D.2. Elementary proof of a regret bound

In this section, we provide another elementary proof of a regret bound, but without reference to Berge's theorem. This will also lead to a better quantification of the bound. In an example, we will show that the bound we obtain is tight.

Define the cosine of an angle between two vectors ad hoc as usual:

$$\cos\Big(\operatorname{ang}(v, w)\Big) := \frac{v \cdot w}{\|v\| \cdot \|w\|},$$

where $v \cdot w$ is the dot product.

**Lemma D.9.** *Let $R$, $\hat{R}$ be two reward functions. Then for any policy $\pi$, we have*

$$J^R(\pi) - J^{\hat{R}}(\pi) = \frac{1}{1 - \gamma} \cdot \|D^\pi\| \cdot \|R - \hat{R}\| \cdot \cos\Big(\operatorname{ang}\big(\eta^\pi, \vec{R} - \vec{\hat{R}}\big)\Big).$$

*Proof.* We have

$$J^R(\pi) - J^{\hat{R}}(\pi) = \eta^\pi \cdot \big(\vec{R} - \vec{\hat{R}}\big) = \|\eta^\pi\| \cdot \|\vec{R} - \vec{\hat{R}}\| \cdot \cos\Big(\operatorname{ang}\big(\eta^\pi, \vec{R} - \vec{\hat{R}}\big)\Big).$$

The result follows from $\eta^\pi = \frac{1}{1-\gamma} \cdot D^\pi$. $\qquad \square$

we will make use of another lemma:

**Lemma D.10.** *Let $a$, $\hat{a}$, and $r$ be three vectors. Assume $a \cdot \hat{a} \geq 0$, where $\cdot$ is the dot product. Then*

$$\cos\Big(\operatorname{ang}(a, r)\Big) - \cos\Big(\operatorname{ang}(\hat{a}, r)\Big) \leq \sqrt{2}.$$

*Proof.* None of the angles change by replacing any of the vectors with a normed version. We can thus assume $\|a\| = \|\hat{a}\| = \|r\| = 1$. We obtain

$$
\begin{aligned}
\left| \cos\big( \text{ang}(a, r) \big) - \cos\big( \text{ang}(\hat{a}, r) \big) \right|^2 &= \left| a \cdot r - \hat{a} \cdot r \right|^2 \\
&= \left| (a - \hat{a}) \cdot r \right|^2 \\
&\leq \|a - \hat{a}\|^2 \cdot \|r\|^2 \\
&= \|a - \hat{a}\|^2 \\
&= \|a\|^2 + \|\hat{a}\|^2 - 2a \cdot \hat{a} \\
&\leq 2.
\end{aligned}
$$

In the first, fourth, and sixth step, we used that all vectors are normed. In the third step, we used the Cauchy-Schwarz inequality. Finally, we used that $a \cdot \hat{a} \geq 0$. The result follows. $\qquad\square$

Recall that for two vectors $v, w$, the projection of $v$ onto $w$ is defined by

$$
\text{proj}_w v := \frac{v \cdot w}{\|w\|^2} w.
$$

This projection is a multiple of $w$, and it minimizes the distance to $v$:

$$
\left\| v - \text{proj}_w v \right\| = \min_{\alpha \in \mathbb{R}} \left\| v - \alpha w \right\|.
$$

We can now formulate and prove our main regret bound:

**Theorem D.11.** *Let $R$ be a fixed, non-trivial reward function, meaning that $\max J^R \neq \min J^R$. Then for all $\hat{R} \in \mathcal{R}$ and all $\hat{\pi} \in \Pi^*(\hat{R})$, we have*

$$
\text{Reg}^R(\hat{\pi}) \leq \frac{\sqrt{2}}{(1 - \gamma) \cdot (\max J^R - \min J^R)} \cdot \left\| \vec{R} - \vec{\hat{R}} \right\|.
$$

*Furthermore, if $\vec{R} \cdot \vec{\hat{R}} \geq 0$, then we also obtain the following stronger bound:*

$$
\text{Reg}^R(\hat{\pi}) \leq \frac{\sqrt{2}}{(1 - \gamma) \cdot (\max J^R - \min J^R)} \cdot \left\| \vec{R} - \text{proj}_{\vec{\hat{R}}} \vec{R} \right\|.
$$

*Now, let $D \in \Delta(\mathcal{S} \times \mathcal{A})$ be a positive data distribution (positive meaning $D(s, a) > 0$ for all $(s, a) \in \mathcal{S} \times \mathcal{A}$). Then we obtain the following consequence:*

$$
\text{Reg}^R(\hat{\pi}) \leq \frac{\sqrt{2}}{(1 - \gamma) \cdot \big( \max J^R - \min J^R \big) \cdot \min_{(s,a) \in \mathcal{S} \times \mathcal{A}} D(s, a)} \cdot d^D(R, \hat{R}).
$$

*Proof.* We start with the first claim. First, notice that the inequality we want to show is equivalent to the following:

$$
J^R(\hat{\pi}) \geq \max J^R - \frac{\sqrt{2}}{1 - \gamma} \cdot \left\| \vec{R} - \vec{\hat{R}} \right\|. \tag{114}
$$

From Lemma D.9, we obtain

$$
J^R(\hat{\pi}) = J^{\hat{R}}(\hat{\pi}) + \frac{1}{1 - \gamma} \cdot \|D^{\hat{\pi}}\| \cdot \left\| \vec{R} - \vec{\hat{R}} \right\| \cdot \cos\big( \text{ang}\big( \eta^{\hat{\pi}}, R - \hat{R} \big) \big).
$$

Now, let $\pi \in \Pi^*(R)$ be an optimal policy for $R$. Then also from Lemma D.9, we obtain

$$
\begin{aligned}
\max J^R = J^R(\pi) &= J^{\hat{R}}(\pi) + \frac{1}{1 - \gamma} \cdot \|D^\pi\| \cdot \left\| \vec{R} - \vec{\hat{R}} \right\| \cdot \cos\big( \text{ang}\big( \eta^\pi, R - \hat{R} \big) \big) \\
&\leq J^{\hat{R}}(\hat{\pi}) + \frac{1}{1 - \gamma} \cdot \|D^\pi\| \cdot \left\| \vec{R} - \vec{\hat{R}} \right\| \cdot \cos\big( \text{ang}\big( \eta^\pi, R - \hat{R} \big) \big).
\end{aligned}
$$

In the last step, we used that $\hat{\pi} \in \Pi^*(\vec{R})$ and so $J^{\hat{R}}(\pi) \leq J^{\hat{R}}(\hat{\pi})$. Combining both computations, we obtain:

$$J^R(\hat{\pi}) \geq \max J^R - \frac{1}{1-\gamma} \cdot \|\vec{R} - \vec{\hat{R}}\| \cdot \left[\|D^\pi\| \cdot \cos\Big(\mathrm{ang}\,\big(\eta^\pi, R - \hat{R}\big)\Big) - \|D^{\hat{\pi}}\| \cdot \cos\Big(\mathrm{ang}\,\big(\eta^{\hat{\pi}}, R - \hat{R}\big)\Big)\right]$$

Since we want to show Equation (114), we are done if we can bound the big bracket by $\sqrt{2}$. By the Cauchy-Schwarz inequality, $\cos\Big(\mathrm{ang}\,(v, w)\Big) \in [-1, 1]$ for all vectors $v, w$. Thus, if the first cosine term is negative or the second cosine term is positive, then since $\|D^\pi\| \leq \|D^\pi\|_1 = 1$, the bound by $\sqrt{2}$ is trivial. Thus, assume that the first cosine term is positive and the second is negative. We obtain

$$\|D^\pi\| \cdot \cos\Big(\mathrm{ang}\,\big(\eta^\pi, R - \hat{R}\big)\Big) - \|D^{\hat{\pi}}\| \cdot \cos\Big(\mathrm{ang}\,\big(\eta^{\hat{\pi}}, R - \hat{R}\big)\Big)$$
$$\leq \cos\Big(\mathrm{ang}\,\big(\eta^\pi, R - \hat{R}\big)\Big) - \cos\Big(\mathrm{ang}\,\big(\eta^{\hat{\pi}}, R - \hat{R}\big)\Big)$$
$$\leq \sqrt{2}$$

by Lemma D.10. Here, we used that $\eta^\pi$ and $\eta^{\hat{\pi}}$ have only non-negative entries and thus also nonnegative dot product $\eta^\pi \cdot \eta^{\hat{\pi}} \geq 0$.

For the second claim, notice the following: if $\vec{R} \cdot \vec{\hat{R}} \geq 0$, then $\mathrm{proj}_{\vec{\hat{R}}} \vec{R} = \alpha \cdot \vec{\hat{R}}$ for some constant $\alpha \geq 0$. Consequently, we have $\hat{\pi} \in \Pi^*\Big(\mathrm{proj}_{\vec{\hat{R}}} \vec{R}\Big)$. The claim thus follows from the first result.

For the third claim, notice that

$$\min_{(s,a)\in\mathcal{S}\times\mathcal{A}} D(s,a) \cdot \|\vec{R} - \vec{\hat{R}}\| \leq \min_{(s,a)\in\mathcal{S}\times\mathcal{A}} D(s,a) \cdot \|\vec{R} - \vec{\hat{R}}\|_1$$
$$= \min_{(s,a)\in\mathcal{S}\times\mathcal{A}} D(s,a) \cdot \sum_{(s,a)\in\mathcal{S}\times\mathcal{A}} \big|R(s,a) - \hat{R}(s,a)\big|$$
$$\leq \sum_{(s,a)\in\mathcal{S}\times\mathcal{A}} D(s,a) \cdot \big|R(s,a) - \hat{R}(s,a)\big|$$
$$= d^D(R, \hat{R}).$$

So the first result implies the third. $\qquad\square$

**Corollary D.12.** *The theorem implies Theorem 3.2.*

*Proof.* Let $L \in (0, 1]$ and assume $\epsilon > 0$ satisfies

$$\epsilon < \frac{1-\gamma}{\sqrt{2}} \cdot \frac{\mathrm{range}\, J^R}{\mathrm{range}\, R} \cdot \min_{(s,a)\in\mathcal{S}\times\mathcal{A}} D(s,a) \cdot L.$$

We want to show $D \in \mathrm{safe}(R, \epsilon, L)$. For this aim, assume that $\hat{R}$ and $\hat{\pi}$ are given with

$$\mathbb{E}_{(s,a)\sim D}\left[\frac{|\hat{R}(s,a) - R(s,a)|}{\mathrm{range}\, R}\right] \leq \epsilon$$

and such that $\hat{\pi}$ is optimal for $\hat{R}$, i.e. (in the notation of the appendix): $\hat{\pi} \in \Pi^*(\hat{R})$. Then the last claim in Theorem D.11

implies

$$\operatorname{Reg}^R(\hat{\pi}) \leq \frac{\sqrt{2}}{(1-\gamma) \cdot \left(\max J^R - \min J^R\right) \cdot \min_{(s,a) \in \mathcal{S} \times \mathcal{A}} D(s,a)} \cdot d^D(R, \hat{R})$$

$$= \frac{\sqrt{2}}{(1-\gamma) \cdot \operatorname{range} J^R \cdot \min_{(s,a) \in \mathcal{S} \times \mathcal{A}} D(s,a)} \cdot \mathbb{E}_{(s,a) \sim D}\left[|R(s,a) - \hat{R}(s,a)|\right]$$

$$= \frac{\sqrt{2}}{1-\gamma} \cdot \frac{\operatorname{range} R}{\operatorname{range} J^R} \cdot \frac{1}{\min_{(s,a) \in \mathcal{S} \times \mathcal{A}} D(s,a)} \cdot \mathbb{E}_{(s,a) \sim D}\left[\frac{|R(s,a) - \hat{R}(s,a)|}{\operatorname{range} R}\right]$$

$$\leq \frac{\sqrt{2}}{1-\gamma} \cdot \frac{\operatorname{range} R}{\operatorname{range} J^R} \cdot \frac{1}{\min_{(s,a) \in \mathcal{S} \times \mathcal{A}} D(s,a)} \cdot \epsilon$$

$$< L.$$

In the first step, we used Theorem D.11. Then, we substituted the definition of $d^D$ from Equations (112) and (113). Then we expanded the term by multiplying with $\operatorname{range} R$ in both the numerator and denominator. Then, we used the assumption that $\hat{R}$ is $\epsilon$-close to $R$ in the data distribution $D$. Finally, we used the assumed bound on $\epsilon$ from Theorem 3.2. Overall, this shows $\operatorname{Reg}^R(\hat{\pi}) < L$, and thus, since $\hat{R}$ and $\hat{\pi}$ were arbitrary, we obtain $D \in \operatorname{safe}(R, \epsilon, L)$. This is precisely the claim from Theorem 3.2. $\qquad\square$

We now include more discussion of Theorem D.11:

*Remark* D.13. As one can easily see geometrically, but also prove directly, there is the following equality of sets for a reward function $R$

$$\left\{ \operatorname{proj}_{\vec{R}} \vec{R} \mid \hat{R} \in \mathcal{R} \right\} = \left\{ \frac{1}{2}\vec{R} + \frac{1}{2}\|\vec{R}\|v \mid v \in \mathbb{R}^{\mathcal{S} \times \mathcal{A}}, \|v\| = 1 \right\}.$$

In other words, the projections form a sphere of radius $\frac{1}{2}\|\vec{R}\|$ around the midpoint $\frac{1}{2}\vec{R}$.

We now show that the regret bound in Theorem D.11 is tight:

**Example D.14.** *Let $U \in [0,1]$ and $\gamma \in [0,1)$ be arbitrary. Then there exists an MDP $\langle \mathcal{S}, \mathcal{A}, \tau, \mu_0, R, \gamma \rangle$ together with a reward function $\hat{R}$ with $\vec{R} \cdot \vec{\hat{R}} \geq 0$ and a policy $\hat{\pi} \in \Pi^*(\hat{R})$ such that*

$$U = \operatorname{Reg}^R(\hat{\pi}) = \frac{\sqrt{2}}{(1-\gamma) \cdot \left(\max J^R - \min J^R\right)} \cdot \left\| \vec{R} - \operatorname{proj}_{\vec{R}} \vec{R} \right\|.$$

*Furthermore, there exists a data distribution $D \in \Delta(\mathcal{S} \times \mathcal{A})$ such that*

$$\operatorname{Reg}^R(\hat{\pi}) = \frac{1}{(1-\gamma) \cdot \left(\max J^R - \min J^R\right) \cdot \min_{(s,a) \in \mathcal{S} \times \mathcal{A}} D(s,a)} \cdot d^D(R, \hat{R}).$$

*Proof.* If $U = 0$ then $\hat{R} = R$ always works. If $U > 0$, then set $\mathcal{S} = \{\star\}$ and $\mathcal{A} = \{a, b, c\}$. This determines $\tau$ and $\mu_0$. Define $R(x) \coloneqq R(\star, x, \star)$ for any action $x \in \mathcal{A}$. Let $R(a) > R(b)$ be arbitrary and set

$$R(c) \coloneqq R(a) - \frac{R(a) - R(b)}{U} \leq R(b).$$

Define

$$\hat{R}(a) \coloneqq \hat{R}(b) \coloneqq \frac{R(a) + R(b)}{2}, \quad \hat{R}(c) \coloneqq R(c).$$

For a policy $\pi$, define $\pi(x) \coloneqq \pi(x \mid \star)$ for any action $x \in \mathcal{A}$ and set the policy $\hat{\pi}$ by $\hat{\pi}(b) = 1$.

We obtain:

$$\|\vec{R} - \hat{\vec{R}}\| = \sqrt{\left(R(a) - \hat{R}(a)\right)^2 + \left(R(b) - \hat{R}(b)\right)^2 + \left(R(c) - \hat{R}(c)\right)^2}$$

$$= \frac{1}{2} \cdot \sqrt{\left(R(a) - R(b)\right)^2 + \left(R(b) - R(a)\right)^2}$$

$$= \frac{1}{\sqrt{2}} \cdot \left(R(a) - R(b)\right)$$

$$= U \cdot \frac{R(a) - R(c)}{\sqrt{2}}$$

$$= U \cdot \frac{\max R - \min R}{\sqrt{2}}$$

$$= U \cdot \frac{(1 - \gamma) \cdot \left(\max J^R - \min J^R\right)}{\sqrt{2}}.$$

Furthermore, we have

$$\mathrm{Reg}^R(\hat{\pi}) = \frac{\frac{1}{1-\gamma} \cdot R(a) - \frac{1}{1-\gamma} \cdot R(b)}{\frac{1}{1-\gamma} \cdot R(a) - \frac{1}{1-\gamma} \cdot R(c)}$$

$$= U.$$

This shows

$$U = \mathrm{Reg}^R(\hat{\pi}) = \frac{\sqrt{2}}{(1 - \gamma) \cdot \left(\max J^R - \min J^R\right)} \cdot \|\vec{R} - \hat{\vec{R}}\|.$$

We are done if we can show that $\mathrm{proj}_{\vec{R}} \vec{R} = \hat{\vec{R}}$. This is equivalent to

$$\hat{\vec{R}} \cdot \vec{R} = \|\hat{\vec{R}}\|^2,$$

which is in turn equivalent to

$$\hat{\vec{R}} \cdot \left[\vec{R} - \hat{\vec{R}}\right] = 0.$$

This can easily be verified.

Finally, for the claim about the data distribution, simply set $D(a) = D(b) = D(c) = \frac{1}{3}$. Then one can easily show that

$$\sqrt{2} \cdot \|\vec{R} - \hat{\vec{R}}\| = R(a) - R(b) = \frac{d^D(R, \hat{R})}{\min_{(s,a) \in \mathcal{S} \times \mathcal{A}} D(s, a)}.$$

That shows the result. □

### D.3. Safe optimization via approximated choice probabilities

In this section, we will show that for any chosen upper regret bound $U$, there is an $\epsilon > 0$ s.t. if the choice probabilities of $\hat{R}$ are $\epsilon$-close to those of $R$, the regret of an optimal policy for $\hat{R}$ is bounded by $U$.

Assume a finite time horizon $T$. Trajectories are then given by $\xi = s_0, a_0, s_1, \ldots, a_{T-1}, s_T$. Let $\Xi$ be the set of all trajectories of length $T$. Let $D \in \Delta(\Xi)$ be a distribution. Assume that the human has a true reward function $R$ and makes choices in trajectory comparisons given by

$$P_R\left(1 \mid \xi_1, \xi_2\right) = \frac{\exp\left(G(\xi_1)\right)}{\exp\left(G(\xi_1)\right) + \exp\left(G(\xi_2)\right)}. \tag{115}$$

Here, the return function $G$ is given by

$$G(\xi) = \sum_{t=0}^{T-1} \gamma^t R(s_t, a_t, s_{t+1}).$$

We can then define the choice distance of proxy reward $\hat{R}$ to true reward $R$ as

$$d_{\mathrm{KL}}^D(R, \hat{R}) := \mathbb{E}_{\xi_1, \xi_2 \sim D \times D}\left[D_{\mathrm{KL}}\left(P_R(\cdot \mid \xi_1, \xi_2) \,\|\, P_{\hat{R}}(\cdot \mid \xi_1, \xi_2)\right)\right]$$

Here, $D_{\mathrm{KL}}\left(P_R(\cdot \mid \xi_1, \xi_2) \,\|\, P_{\hat{R}}(\cdot \mid \xi_1, \xi_2)\right)$ is the Kullback-Leibler divergence of two binary distributions over values $1, 2$. Explicitly, for $P := P_R(\cdot \mid \xi_1, \xi_2)$ and similarly $\hat{P}$, we have

$$\begin{aligned}
D_{\mathrm{KL}}\left(P \,\|\, \hat{P}\right) &= P(1) \log \frac{P(1)}{\hat{P}(1)} + \left(1 - P(1)\right) \log \frac{1 - P(1)}{1 - \hat{P}(1)} \\
&= -\left[P(1) \log \hat{P}(1) + \left(1 - P(1)\right) \log \left(1 - \hat{P}(1)\right)\right] - H\left(P(1)\right).
\end{aligned} \tag{116}$$

Here, $H(p) := -\left[p \log p + (1 - p) \log(1 - p)\right]$ is the binary entropy function.

Fix in this whole section the true reward function $R$ with $\max J^R \neq \min J^R$ in a fixed MDP.

The goal of this section is to prove the following proposition:

**Proposition D.15.** *Let $U \in (0, 1]$. Then there exists an $\epsilon > 0$ such that for all $\hat{R}$ with*

$$d_{\mathrm{KL}}^D(R, \hat{R}) < \epsilon$$

*and all $\hat{\pi} \in \Pi^*(\hat{R})$ we have $\mathrm{Reg}^R(\hat{\pi}) < U$.*

We prove this by chaining together four lemmas. The first of the four lemmas needs its own lemma, so we end up with five lemmas overall:

**Lemma D.16.** *Assume $R, \hat{R}$ are two reward functions and $\pi$ a policy. Then*

$$\left|J^R(\pi) - J^{\hat{R}}(\pi)\right| \leq \max_{\xi \in \Xi}\left|G(\xi) - \hat{G}(\xi)\right|.$$

*Proof.* We have

$$\begin{aligned}
\left|J^R(\pi) - J^{\hat{R}}(\pi)\right| &= \left|\widetilde{D}^\pi \cdot \left(G - \hat{G}\right)\right| \\
&= \left|\sum_{\xi \in \Xi} \widetilde{D}^\pi(\xi) \cdot \left(G(\xi) - \hat{G}(\xi)\right)\right| \\
&\leq \sum_{\xi \in \Xi} \widetilde{D}^\pi(\xi) \cdot \left|G(\xi) - \hat{G}(\xi)\right| \\
&\leq \max_{\xi \in \Xi}\left|G(\xi) - \hat{G}(\xi)\right| \cdot \sum_{\xi \in \Xi} \widetilde{D}^\pi(\xi) \\
&= \max_{\xi \in \Xi}\left|G(\xi) - \hat{G}(\xi)\right|
\end{aligned}$$

In the last step, we used that distributions sum to one. $\qquad\square$

**Lemma D.17.** *Let $U \in (0, 1]$. Then there exists $\sigma(U) > 0$ such that for all $\hat{R}$ and $\hat{\pi} \in \Pi^*(\hat{R})$ for which there exists $c \in \mathbb{R}$ such that $\max_{\xi \in \Xi}\left|\hat{G}(\xi) - G(\xi) - c\right| < \sigma(U)$, we have $\mathrm{Reg}^R(\hat{\pi}) < U$.*

*Concretely, we can set $\sigma(U) := \frac{\max J^R - \min J^R}{2} \cdot U$.*

*Proof.* Set $\sigma(U)$ as stated and let $\hat{R}$, $\hat{\pi}$ and $c$ have the stated properties. The regret bound we want to show is equivalent to the following statement:

$$J^R(\hat{\pi}) > \max J^R - \left(\max J^R - \min J^R\right) \cdot U = \max J^R - 2\sigma(U). \tag{117}$$

Let $\tilde{c}$ be the constant such that $\hat{G} - c$ is the return function of $\hat{R} - \tilde{c}$. Concretely, one can set $\tilde{c} = \frac{1-\gamma}{1-\gamma^{T+1}} \cdot c$. Lemma D.16 ensures that

$$J^R(\hat{\pi}) > J^{\hat{R}-\tilde{c}}(\hat{\pi}) - \sigma(U). \tag{118}$$

Now, let $\pi$ be an optimal policy for $R$. Again, Lemma D.16 ensures

$$\max J^R = J^R(\pi) < J^{\hat{R}-\tilde{c}}(\pi) + \sigma(U) \leq J^{\hat{R}-\tilde{c}}(\hat{\pi}) + \sigma(U). \tag{119}$$

In the last step, we used that $\hat{\pi}$ is optimal for $\hat{R}$ and thus also $\hat{R} - \tilde{c}$. Combining Equations (118) and (119), we obtain the result, Equation (117). $\qquad\square$

**Lemma D.18.** *For $q \in (0,1)$, define $g_q : (-q, 1-q) \to \mathbb{R}$ by*

$$g_q(x) := \log \frac{q+x}{1-(q+x)}.$$

*Then for all $\sigma > 0$ there exists $\delta(q, \sigma) > 0$ such that for all $x \in (-q, 1-q)$ with $|x| < \delta(q, \sigma)$, we have $|g_q(x) - g_q(0)| < \sigma$.*

*Concretely, one can choose*

$$\delta(q, \sigma) := \big( \exp(\sigma) - 1 \big) \cdot \min \left\{ \frac{1}{\frac{1}{q} + \frac{\exp(\sigma)}{1-q}}, \ \frac{1}{\frac{1}{1-q} + \frac{\exp(\sigma)}{q}} \right\}$$

*Proof.* If one does not care about the precise quantification, then the result is simply a reformulation of the continuity of $g_q$ at the point $x_0 = 0$.

Now we show more specifically that $\delta(q, \sigma)$, as defined above, has the desired property. Namely, notice the following sequence of equivalences (followed by a one-sided implication) that holds whenever $x \geq 0$:

$$
\begin{aligned}
\big| g_q(x) - g_q(0) \big| < \sigma \quad &\Longleftrightarrow \quad \log \frac{(q+x) \cdot (1-q)}{\big(1-(q+x)\big) \cdot q} < \sigma \\
&\Longleftrightarrow \quad \frac{(q+x) \cdot (1-q)}{\big(1-(q+x)\big) \cdot q} < \exp(\sigma) \\
&\Longleftrightarrow \quad (q+x) < (1-q-x) \cdot \frac{q}{1-q} \cdot \exp(\sigma) \\
&\Longleftrightarrow \quad \left( 1 + \frac{q}{1-q} \cdot \exp(\sigma) \right) \cdot x < q \cdot \big( \exp(\sigma) - 1 \big) \\
&\Longleftrightarrow \quad x < \frac{\exp(\sigma) - 1}{\frac{1}{q} + \frac{\exp(\sigma)}{1-q}} \\
&\Longleftarrow \quad |x| < \delta(q, \sigma).
\end{aligned}
$$

In the first step, we used the monotonicity of $g_q$ to get rid of the absolute value. Similarly, whenever $x \leq 0$, we have

$$
\begin{aligned}
\big| g_q(x) - g_q(0) \big| < \sigma \quad &\Longleftrightarrow \quad x > \frac{1 - \exp(\sigma)}{\frac{1}{1-q} + \frac{\exp(\sigma)}{q}} \\
&\Longleftarrow \quad |x| < \delta(q, \sigma).
\end{aligned}
$$

This shows the result. $\qquad\square$

**Lemma D.19.** *For $q \in (0,1)$, define $f_q : (0,1) \to \mathbb{R}$ by*

$$f_q(p) := -\big[ q \log p + (1-q) \log(1-p) \big].$$

*Then for all $\delta > 0$ there exists $\mu(\delta) > 0$ such that for all $p \in (0,1)$ with $f_q(p) < H(q) + \mu(\delta)$, we have $|p - q| < \delta$. Concretely, one can choose $\mu(\delta) := 2\delta^2$.*

*Proof.* Let $\delta > 0$ and define $\mu(\delta) := 2\delta^2$. Assume that $f_q(p) < H(q) + \mu(\delta)$. By Pinker's inequality, we have

$$
\begin{aligned}
2(p-q)^2 &\leq q \log \frac{q}{p} + (1-q) \cdot \log \frac{1-q}{1-p} \\
&= -H(q) + f_q(p) \\
&< \mu(\delta) \\
&= 2\delta^2.
\end{aligned}
$$

Consequently, we have $|p - q| < \delta$. $\qquad\square$

**Lemma D.20.** *Define $f_q(p)$ as in Lemma D.19. Then for all $\mu > 0$ there exists $\epsilon(\mu) > 0$ such that for all $\hat{R}$ with $d_{\mathrm{KL}}^D(R, \hat{R}) < \epsilon(\mu)$, we have the following for all $\xi_1, \xi_2 \in \Xi$:*

$$
f_{P_R(1|\xi_1,\xi_2)}\big(P_{\hat{R}}(1 \mid \xi_1, \xi_2)\big) < H\big(P_R(1 \mid \xi_1, \xi_2)\big) + \mu.
$$

*Concretely, we can set $\epsilon(\mu) := \mu \cdot \min_{\xi_1,\xi_2 \in \Xi} D(\xi_1) \cdot D(\xi_2)$*

*Proof.* We have the following for all $\xi_1, \xi_2 \in \Xi$:

$$
\begin{aligned}
\mu \cdot \min_{\xi,\xi'} D(\xi) \cdot D(\xi) = \epsilon(\mu) \\
> d_{\mathrm{KL}}^D(R, \hat{R}) \\
= \mathbb{E}_{\xi,\xi' \sim D \times D}\Big[D_{\mathrm{KL}}\Big(P_R(\cdot \mid \xi, \xi') \,\|\, P_{\hat{R}}(\cdot \mid \xi, \xi')\Big)\Big] \\
\geq \Big(\min_{\xi,\xi'} D(\xi) \cdot D(\xi')\Big) \cdot D_{\mathrm{KL}}\Big(P_R(\cdot \mid \xi_1, \xi_2) \,\|\, P_{\hat{R}}(\cdot \mid \xi_1, \xi_2)\Big)
\end{aligned}
$$

Now, Equation (116) shows that

$$
D_{\mathrm{KL}}\Big(P_R(\cdot \mid \xi_1, \xi_2) \,\|\, P_{\hat{R}}(\cdot \mid \xi_1, \xi_2)\Big) = f_{P_R(1|\xi_1,\xi_2)}\big(P_{\hat{R}}(1 \mid \xi_1, \xi_2)\big) - H\big(P_R(1 \mid \xi_1, \xi_2)\big).
$$

The result follows. $\qquad\square$

**Corollary D.21.** *Let $\sigma > 0$. Then there exists $\epsilon := \epsilon(\sigma) > 0$ such that $d_{\mathrm{KL}}^D(R, \hat{R}) < \epsilon$ implies that there exists $c \in \mathbb{R}$ such that $\big\|G - (\hat{G} - c)\big\|_\infty < \sigma$.*

*Proof.* Set

$$
\delta := \min_{\xi_1,\xi_2 \in \Xi \times \Xi} \delta\Big(P_R(1 \mid \xi_1, \xi_2), \sigma\Big), \ \mu := \mu(\delta), \ \epsilon := \epsilon(\mu),
$$

with the constants satisfying the properties from Lemmas D.18, D.19, and D.20. Now, let $\hat{R}$ be such that $d_{\mathrm{KL}}^D(R, \hat{R}) < \epsilon$.

First of all, Lemma D.20 ensures that

$$
f_{P_R(1|\xi_1,\xi_2)}\big(P_{\hat{R}}(1 \mid \xi_1, \xi_2)\big) < H\big(P_R(1 \mid \xi_1, \xi_2)\big) + \mu
$$

for all $\xi_1, \xi_2 \in \Xi$. Then Lemma D.19 shows that

$$
\big|P_{\hat{R}}(1 \mid \xi_1, \xi_2) - P_R(1 \mid \xi_1, \xi_2)\big| < \delta
$$

for all $\xi_1, \xi_2 \in \Xi$. From Lemma D.18, we obtain that

$$
\left| g_{P_R(1|\xi_1,\xi_2)}\Big(P_{\hat{R}}\big(1 \mid \xi_1, \xi_2\big) - P_R\big(1 \mid \xi_1, \xi_2\big)\Big) - g_{P_R(1|\xi_1,\xi_2)}(0)\right| < \sigma \tag{120}
$$

for all $\xi_1, \xi_2 \in \Xi$. Now, note that

$$
g_{P_R(1|\xi_1,\xi_2)}\Big(P_{\hat{R}}\big(1 \mid \xi_1, \xi_2\big) - P_R\big(1 \mid \xi_1, \xi_2\big)\Big) = g_{P_{\hat{R}}(1|\xi_1,\xi_2)}(0).
$$

Furthermore, for $R' \in \{R, \hat{R}\}$, Equation (115) leads to the following computation:

$$
\begin{aligned}
g_{P_{R'}(1|\xi_1,\xi_2)}(0) &= \log \frac{P_{R'}(1 \mid \xi_1, \xi_2)}{P_{R'}(2 \mid \xi_1, \xi_2)} \\
&= \log \frac{\exp\left(G'(\xi_1)\right)}{\exp\left(G'(\xi_2)\right)} \\
&= G'(\xi_1) - G'(\xi_2).
\end{aligned}
$$

Therefore, Equation (120) results in

$$
\left| \left(\hat{G}(\xi_1) - G(\xi_1)\right) - \left(\hat{G}(\xi_2) - G(\xi_2)\right) \right| = \left| \left(\hat{G}(\xi_1) - \hat{G}(\xi_2)\right) - \left(G(\xi_1) - G(\xi_2)\right) \right| < \sigma
$$

for all $\xi_1, \xi_2 \in \Xi$. Now, let $\xi^* \in \Xi$ be any reference trajectory. Define $c \coloneqq \hat{G}(\xi^*) - G(\xi^*)$. Then the preceding equation shows that

$$
\left| \hat{G}(\xi) - G(\xi) - c \right| < \sigma
$$

for all $\xi \in \Xi$. That shows the claim. □

*Proof of Proposition D.15.* We prove Proposition D.15 by chaining together the constants from the preceding results. We have $U \in (0, 1]$ given. Then, set $\sigma \coloneqq \sigma(U)$ and $\epsilon \coloneqq \epsilon(\sigma)$ as in Lemma D.17 and Corollary D.21. Now, let $\hat{R}$ be such that $d_{\mathrm{KL}}^D(R, \hat{R}) < \epsilon$ and let $\hat{\pi} \in \Pi^*(\hat{R})$. Our goal is to show that $\mathrm{Reg}^R(\hat{\pi}) < U$.

By Corollary D.21, there is $c > 0$ such that $\max_{\xi \in \Xi} \left| \hat{G}(\xi) - G(\xi) - c \right| < \sigma$. Consequently, Lemma D.17 ensures that $\mathrm{Reg}^R(\hat{\pi}) < U$. This was to show. □

## D.4. Positive result for regularized RLHF

Here, we present simple positive results for regularized RLHF, both in a version with the expected reward distance, and in a version using the distance in choice probabilities. Some of it will directly draw from the positive results proved before.

**Theorem D.22.** *Let $\lambda \in (0, \infty)$ be given and fixed. Assume we are given an MDP $\langle \mathcal{S}, \mathcal{A}, \tau, \mu_0, R, \gamma \rangle$, and a data distribution $D \in \mathcal{S} \times \mathcal{A}$ which assigns positive probability to all transitions, i.e., $\forall (s, a) \in \mathcal{S} \times \mathcal{A}, \, D(s, a) > 0$. Let $\omega : \Pi \to \mathbb{R}$ be a continuous regularization function that has a reference policy $\pi_{\mathrm{ref}}$ as one of its minima.[6] Assume that $\pi_{\mathrm{ref}}$ is not $(\lambda, \omega)$-optimal for $R$ and let $L = \mathrm{Reg}^R(\pi_{\mathrm{ref}})$. Then there exists $\epsilon > 0$ such that $D \in \mathbf{safe}(R, \epsilon, L, \lambda, \omega)$.*

*Proof.* We prove the theorem by showing that for every $D \in \Delta(\mathcal{S} \times \mathcal{A})$ such that $D(s, a) > 0$ for all $(s, a) \in \mathcal{S} \times \mathcal{A}$, there exists $\epsilon > 0$ such that for all $\hat{R}$ with $\mathbb{E}_{(s,a) \sim D}\left[ \frac{|\hat{R}(s,a) - R(s,a)|}{\mathrm{range}\, R} \right] < \epsilon$ and all policies $\hat{\pi}$ that are $(\lambda, \omega)$-RLHF optimal wrt. $\hat{R}$, we have $\mathrm{Reg}^R(\hat{\pi}) < \mathrm{Reg}^R(\pi_{\mathrm{ref}})$. Because $L = \mathrm{Reg}^R(\hat{\pi}) < \mathrm{Reg}^R(\pi_{\mathrm{ref}})$ this proves that then $D \in \mathbf{safe}(R, \epsilon, L, \lambda, \omega)$.

The proof is an application of Berge's maximum Theorem, Theorem D.3. Namely, define the function

$$
f : \mathcal{R} \times \Pi \to \mathbb{R}, \quad f(R, \pi) \coloneqq J_R(\pi) - \lambda \omega(\pi).
$$

Furthermore, define the correspondence $C : \mathcal{R} \rightrightarrows \Pi$ as the trivial map $C(R) = \Pi$. Let $f^* : \mathcal{R} \to \mathbb{R}$ map a reward function to the value of a $(\lambda, \omega)$-RLHF optimal policy, i.e., $f^*(R) \coloneqq \max_{\pi \in \Pi} f(R, \pi)$. Define $C^*$ as the corresponding argmax, i.e., $C^*(R) \coloneqq \{ \pi \mid f(R, \pi) = f^*(R) \}$. Assume on $\mathcal{R}$ we have the standard Euclidean topology. Since $\omega$ is assumed continuous and by Proposition D.4 also $J$ is continuous, it follows that $f$ is continuous. Thus, Theorem D.3 implies that $C^*$ is upper hemicontinuous, see Definition D.2. The rest of the proof is simply an elaboration of why upper hemicontinuity of $C^*$ gives the result.

Now, define the set

$$
\mathcal{V} \coloneqq \left\{ \pi' \in \Pi \mid \mathrm{Reg}^R(\pi') < \mathrm{Reg}^R(\pi_{\mathrm{ref}}) \right\}.
$$

---

[6]E.g., if $\pi_{\mathrm{ref}}(a \mid s) > 0$ for all $(s, a) \in \mathcal{S} \times \mathcal{A}$ and $\omega(\pi) \coloneqq \mathbb{D}_{\mathrm{KL}}(\pi \| \pi_{\mathrm{ref}})$, then the minimum is given by $\pi_{\mathrm{ref}}$.

Since the regret is a continuous function, this set is open. Now, let $\pi \in C^*(R)$ be $(\lambda, \omega)$-RLHF optimal with respect to $R$. It follows

$$\begin{aligned}
J_R(\pi) &= f(R, \pi) + \lambda\omega(\pi) \\
&> f(R, \pi_{\text{ref}}) + \lambda\omega(\pi_{\text{ref}}) \\
&= J_R(\pi_{\text{ref}}),
\end{aligned}$$

where we used the optimality of $\pi$ for $f$, that $\pi_{\text{ref}}$ is not optimal for it, and that $\pi_{\text{ref}}$ is the minimum of $\omega$. So overall, this shows $C^*(R) \subseteq \mathcal{V}$.

Since $C^*$ is upper hemicontinuous, this means there exists an open set $\mathcal{U} \subseteq \mathcal{R}$ with $R \in \mathcal{U}$ and such that for all $\hat{R} \in \mathcal{U}$, we have $C^*(\hat{R}) \subseteq \mathcal{V}$. Let $\epsilon > 0$ be so small that all reward functions $\hat{R}$ with $\mathbb{E}_{(s,a)\sim D}\left[\frac{|\hat{R}(s,a) - R(s,a)|}{\text{range } R}\right] < \epsilon$ satisfy $\hat{R} \in \mathcal{U}$ — which exists since $\mathcal{U}$ is open in the Euclidean topology. Then for all such $\hat{R}$ and any policy $\hat{\pi}$ that is $(\lambda, \omega)$-RLHF optimal wrt. $\hat{R}$, we by definition have

$$\hat{\pi} \in C^*(\hat{R}) \subseteq \mathcal{V},$$

and thus, by definition of $\mathcal{V}$, the desired regret property. This was to show. $\qquad\square$

Now, we show the same result, but with the choice distance instead of expected reward distance:

**Theorem D.23.** *Let $\lambda \in (0, \infty)$ be given and fixed. Assume we are given an MDP $\langle \mathcal{S}, \mathcal{A}, \tau, \mu_0, R, \gamma \rangle$, and a data distribution $D \in \mathcal{S}\times\mathcal{A}$ which assigns positive probability to all transitions, i.e., $\forall(s,a) \in \mathcal{S}\times\mathcal{A}, \; D(s,a) > 0$. Let $\omega : \Pi \to \mathbb{R}$ be a continuous regularization function that has a reference policy $\pi_{\text{ref}}$ as one of its minima. Assume that $\pi_{\text{ref}}$ is not $(\lambda, \omega)$-optimal for $R$ and let $L = \text{Reg}^R(\pi_{\text{ref}})$. Then there exists $\epsilon > 0$ such that $D \in \textbf{safe}^{\mathbb{D}_{KL}}\left(R, \epsilon, L, \lambda, \omega\right)$.*

*Proof.* Let $\mathcal{G} := \mathbb{R}^\Xi$ be the vector space of return functions, which becomes a topological space when equipped with the infinity norm. Define the function

$$f : \mathcal{G} \times \Pi \to \mathbb{R}, \quad f(G, \pi) := J^G(\pi) - \lambda\omega(\pi),$$

where $J^G(\pi) := \mathbb{E}_{\xi\sim\pi}[G(\xi)]$ is the policy evaluation function of the return function $G$. $f$ is continuous. Define the correspondence $C : \mathcal{G} \rightrightarrows \Pi$ as the trivial map $C(G) = \Pi$. Let $f^* : \mathcal{G} \to \mathbb{R}$ map a return function to the value of a $(\lambda, \omega)$-optimal policy, i.e., $f^*(G) := \max_{\pi\in\Pi} f(G, \pi)$. Define $C^*$ as the corresponding argmax. Then Theorem D.3 implies that $C^*$ is upper hemicontinuous, see Definition D.2. As in the previous proof, the rest is an elaboration of why this gives the desired result.

Set $G$ as the return function corresponding to $R$. Define

$$\mathcal{V} := \left\{\pi' \in \Pi \mid \text{Reg}^R(\pi') < L\right\}.$$

We now claim that $C^*(G) \subseteq \mathcal{V}$. Indeed, let $\pi \in C^*(G)$. Then

$$\begin{aligned}
J^R(\pi) &= f(G, \pi) + \lambda\omega(\pi) \\
&> f(G, \pi_{\text{ref}}) + \lambda\omega(\pi_{\text{ref}}) \\
&= J^R(\pi_{\text{ref}}).
\end{aligned}$$

Note that we used the optimality of $\pi$ for $f$, that $\pi_{\text{ref}}$ is not optimal for it, and also that $\pi_{\text{ref}}$ minimizes $\omega$ by assumption. This shows $\text{Reg}^R(\pi) < \text{Reg}^R(\pi_{\text{ref}}) = L$, and thus the claim.

Since $C^*$ is upper hemicontinuous and $\mathcal{V}$ an open set, this implies that there exists $\sigma > 0$ such that for all $\hat{G} \in \mathcal{G}$ with $\left\|G - \hat{G}\right\|_\infty < \sigma$, we have $C^*(\hat{G}) \subseteq \mathcal{V}$.

Now, define $\epsilon := \epsilon(\sigma)$ as in Corollary D.21 and let $\hat{R}$ be any reward function with $d_{\text{KL}}^D(R, \hat{R}) < \epsilon$. Then by that corollary, there exists $c \in \mathbb{R}$ such that $\left\|G - (\hat{G} - c)\right\|_\infty < \sigma$. Consequently, we have $C^*(\hat{G}) = C^*(\hat{G} - c) \subseteq \mathcal{V}$ by what we showed before, which shows the result. $\qquad\square$

