# OpenReview forum: "The Perils of Optimizing Learned Reward Functions: Low Training Error Does Not Guarantee Low Regret"
_ICML.cc/2025/Conference — ICML 2025 poster_

### Official Review · Reviewer_oQAd · 2025-03-15

**Overall Recommendation:** 5

**Summary:**

This paper seeks to characterize the relationship between data distributions over the state-action space of a prescribed Markov decision process (MDP), reward learning from such data distributions, and the resulting regret (which the authors define as normalized suboptimality w.r.t. the true reward function) of optimal policies for the resulting learned reward functions. The core goal of the work can be summed up as characterizing conditions on the data distribution and learned reward error under which large regret is possible, a situation the authors call "error-regret mismatch". The paper develops theoretical machinery to achieve this, first defining the notion of "unsafe" and "safe" data distributions capturing when error-regret mismatch is and is not possible, respectively, then establishing several results characterizing specific conditions on MDPs, data distributions, and learned rewards under which error-regret mismatch occurs. A key result, Theorem 3.5, provides necessary and sufficient conditions under which a data distribution is safe for a given MDP, learned reward error, and regret; the conditions are provided in the form of a matrix inequality (a system of linear inequalities), and tools are provided for explicitly computing the associated matrix in Appendix C when the MDP is known. Several of the error-regret mismatch results are extended to the regularized MDP setting and connections to reinforcement learning from human feedback (RLHF) are discussed in detail.

**Claims And Evidence:**

*Are the claims made in the submission supported by clear and convincing evidence? If not, which claims are problematic and why?*

Yes. See Strength 1 from **Strengths and Weaknesses** section below.

**Essential References Not Discussed:**

None, to my knowledge.

**Experimental Designs Or Analyses:**

n/a

**Methods And Evaluation Criteria:**

*Do proposed methods and/or evaluation criteria (e.g., benchmark datasets) make sense for the problem or application at hand?*

Yes, the theoretical methods and criteria proposed make sense.

**Other Comments Or Suggestions:**

n/a

**Other Strengths And Weaknesses:**

**Strengths**

The paper enjoys major strengths:
1. The paper establishes a rigorous and clearly useful theoretical framework for studying the relationship between data distributions and suboptimality when performing reward learning. Such a framework has been lacking in the reward learning / RLHF literature, to my knowledge. Definition 2.1 provides a well-motivated and concrete characterization of "safe"/"unsafe" data distributions, as described in the **Summary** above. The results in Sections 3-6 provide concrete (though somewhat partial, as detailed in **Weaknesses** below) insight into the conditions under which error-regret mismatch occurs. The proofs of the key results provided in the appendix appear to be correct (I read those of Appendix C and Appendix D through D.7 fairly closely, and skimmed the rest) and provide significant additional insight. I especially highlight Theorem 3.5 discussed in my **Summary**: I believe that this result, along with the proof, explicit characterization of the system of inequalities, and algorithm for computing $M$ provided in Appendix C, all provide fundamental and important tools that can likely be used to study the error-regret mismatch problem for specific classes of MDPs in future work; I feel that these results may constitute the most significant contribution of the paper.
2. The quality of the presentation and writing is very high. All contributions, assumptions, and results are clearly stated, thoroughly motivated, and satisfactorily discussed. The limitations of the setup (e.g., the discussion starting line 214, left column, regarding the difficulty of guaranteeing the learned reward error in eq. (1)) and the computational challenges of using some of the results (e.g., potential intractability of computing $M$ in Theorem 3.5, discussed at the end of Section 3) are clearly described. The proofs in the appendix are well-written, well-organized, and sufficient intuition and discussion are provided.
3. The motivation and relevance of this work to the community is very high, particularly due to the practical importance of RLHF in tuning large language models (LLMs), as discussed in Section 6. Beyond the LLM and RLHF communities, addressing questions of data coverage and reward learning are important to the offline RL and inverse RL communities, respectively, and the theoretical machinery established in this paper may prove useful to them as well.

**Weaknesses**

My primary complaint is that two of the key results on "unsafety" of data distributions are not constructive, in the sense outlined below. Specifically, Proposition 3.3 and Corollary 3.4 directly assume the existence of certain classes of policies that can then be shown to force a given data distribution or all data distributions to be unsafe. While this provides a (potentially) useful sufficient condition for "unsafety", it leaves open the question of whether such policies actually exist and is therefore not constructive. It would be more satisfying and clearly useful to have existence results or at least worked examples showing that the sufficient conditions of Prop. 3.3 and Cor. 3.4 do hold under reasonable conditions. I suspect that other results from the paper (e.g., Thm. 3.5 or the $D$ construction from the pf. of Thm. 4.2) could be used to construct such results/examples.

**Questions For Authors:**

1. When might we expect the $\hat{\pi}$ from Prop. 3.3 to exist?
2. When might we expect the policy class $\Pi_L$ from Cor. 3.4 to exist?

**Relation To Broader Scientific Literature:**

The key results from this paper provide a theoretical framework for studying the relationship between data distributions and suboptimality when performing reward learning. Such a framework has been lacking in the reward learning / RLHF literature, to my knowledge. This is of potentially high interest to the community due to the practical importance of RLHF in tuning large language models (LLMs). Beyond the LLM and RLHF communities, addressing questions of data coverage and reward learning are important to the offline RL and inverse RL communities, respectively, and the theoretical machinery established in this paper may prove useful to them as well.

**Theoretical Claims:**

*Did you check the correctness of any proofs for theoretical claims? Please specify which ones, and discuss any issues.*

Yes: Sections 3 & 4 in detail, skimmed those of Sections 5 & 6.

---

> ### Author Rebuttal · Authors · 2025-03-31
>
> Thank you for your thorough review! We address your main concern below.
>
> > It would be more satisfying and clearly useful to have existence results or at least worked examples showing that the sufficient conditions of Prop. 3.3 and Cor. 3.4 do hold under reasonable conditions. [...] 1. When might we expect the \hat{\pi} from Prop. 3.3 to exist? 2. When might we expect the policy class \Pi_L from Cor. 3.4 to exist?
>
> This is a reasonable question. We do not yet have concrete theoretical results “constructing” bad policies in general settings, but we think the example in Appendix B.4 provides useful intuitions that one could expand to a general result with further work. In particular, in the example, there are many *equivalent styles* for an LLM to phrase an answer that are all equally bad according to the true reward function (e.g., imagine instructions to build a nuclear weapon in many different languages). If this is the case, then no data distribution can cover *all* of these different styles; intuitively, the distribution has only a total weight of “one” to distribute, and so some styles necessarily get low weight.
>
> We think this could be turned into a general result by assuming an *equivalence relation* on the set of state-action pairs, such that equivalence classes of state-action pairs
>
> - are “large”; in particular, for every fixed state s, and every action a, there exist many equivalent pairs (s,a′) sharing the same state;
> - have constant true reward;
> - have well-defined (total) transition probabilities to equivalence classes of states.
>
> We believe an example of such equivalence relations is given by *MDPs with symmetry* [1] for sufficiently large symmetry groups. If the three conditions hold, then for each “bad” policy and for each state-action pair in its support, one can find an equivalent action that is relatively unsupported by the data distribution D. If one then replaces the policy with a new policy that always chooses an equivalent relatively unsupported action in each state, then the new policy should turn out to be equally bad, but not very supported by D, leading to the condition in Proposition 3.3. Similarly, for each bad policy, one can construct many equivalent ones whose supports are mutually disjoint, leading to the sufficient condition in Corollary 3.4.
>
> In the revised version of the paper, we will discuss Example B.4 and its general properties in more detail in the main paper. Beyond that, we are unsure whether to include a general result along the lines we just sketched, mainly since this is a quick idea that we haven’t yet checked in detail. What do you think?
>
> [1] Elise van der Pol et al., *MDP Homomorphic Networks: Group Symmetries in Reinforcement Learning*, NeurIPS 2020

---

> > ### Comment · Reviewer_oQAd · 2025-04-04
> >
> > Thanks for your response. The potential approach you've outlined for constructing examples supporting Cor. 3.4 for more general classes of problems (MDPs with symmetry, at least) sounds reasonable, and I'd be interested to see the details once it's fully worked out. For the current paper, however, a high-level discussion of the types of problems for which \hat{\pi} from Prop. 3.3 and \Pi_L from Cor. 3.4 can be expected to exist should suffice.

---

> > > ### Author Response · Authors · 2025-04-08
> > >
> > > Thank you again for your thoughtful feedback.
> > >
> > > As you suggest, we will expand the discussion in the revised paper to clarify when we expect the relevant policies to exist. We believe the intuitive explanation we previously outlined --- particularly around MDPs with symmetries --- will be helpful. Given the current stage of the review process, we prefer not to add entirely new theoretical results, but we will ensure the final paper clearly addresses this aspect.
> > >
> > > Thank you once more for your constructive comments.

---

### Official Review · Reviewer_QgBs · 2025-03-17

**Overall Recommendation:** 4

**Summary:**

This paper defines a notion called "error-regret" mismatch in the context of optimizing a learned reward function. Error-regret mismatch refers to when the learned reward is close to the true reward on a fixed distribution (low error), but when optimized the learned reward leads to a policy which performs poorly under the true reward function (high regret). The authors present a range of theoretical results showing that error-regret mismatch is difficult to avoid, even when using regularized optimization.

**Claims And Evidence:**

The claims in the paper seem to be well-supported; error-regret mismatch is clearly motivated, defined, and explored through theoretical results and explanations.

**Essential References Not Discussed:**

These two papers study quite similar settings and are not referenced:
 * Kwa et al. Catastrophic Goodhart: regularizing RLHF with KL divergence does not mitigate heavy-tailed reward misspecification. NeurIPS 2024.
 * Laidlaw et al. Correlated Proxies: A New Definition and Improved Mitigation for Reward Hacking. ICLR 2025

It would be helpful to have a comparison to the results of these papers, which both show cases in which regularization can succeed at preventing an error-regret mismatch.

**Experimental Designs Or Analyses:**

No experiments.

**Methods And Evaluation Criteria:**

No empirical results.

**Other Comments Or Suggestions:**

No other comments.

**Other Strengths And Weaknesses:**

I found the paper clear and easy to read. The paper defines an important phenomenon that has significant implications for real-world RL training and AI safety.

**Questions For Authors:**

No questions.

**Relation To Broader Scientific Literature:**

In general the relation to the literature seems good, although I think there are a couple of missing references (see below).

**Theoretical Claims:**

I did not carefully check the proofs but the theorems seem intuitively correct to me.

---

> ### Author Rebuttal · Authors · 2025-04-01
>
> Thank you very much for your review! We are happy you found the paper clear, and that you highlighted the significance of this work.
>
> We would also like to thank you for pointing out two further related works. We are integrating them into our revised related work section. In the following, we provide a comparison between these works and ours.
>
> The results of our paper demonstrate that one gets very few mathematical safety guarantees in a wide variety of different reward-learning and (regularized or unregularized) policy optimization settings. An interesting question then is figuring out what can be done to make reward learning safer. The two works provide attempts at an answer:
>
> **1. Develop well-motivated algorithms that don’t provide mathematical safety guarantees but work well empirically.**
>
> The paper *Laidlaw et al. Correlated Proxies: A New Definition and Improved Mitigation for Reward Hacking* pursues this approach. In particular, they use the fact that if you have a data sampling policy $\pi_{ref}$ over which the true reward function and the learned reward functions correlate, then you can constrain your policy training procedure to avoid states that are unlikely under your data sampling policy. They develop a regularization method that penalizes going off training distribution (by penalizing the Chi-squared divergence of occupancy measures, see their Theorem 5.1 and Equation 4) and show in Section 6 that this method works well empirically for the environments they test. Note that while the restriction to remain “close” to the reference policy/training distribution in occupancy measure space can prevent reward hacking behavior, it also makes you dependent on the quality of said reference policy.
>
>  On theoretical guarantees: In Appendix A.1.3 (in particular Lemma A.3) they show that for their setup there always exists an MDP for which their method allows reward hacking, i.e., that their algorithm can’t always guarantee safety. In Appendix A.2 (in particular Theorem A.5) they show that this result is not specific to their algorithm, but generalizes to every algorithm that uses some form of penalty on the f-divergence between action distributions. While their results show the existence of a *single* MDP for which regularized policy optimization algorithms might not be safe (this might not be too bad in practice), we show (see Theorem 4.2) that, in fact, large classes of MDPs have many different unsafe data distributions for many different policy regularization methods. We show this by considering the more fine-grained setting of analyzing what subset of data distributions are safe/unsafe for *arbitrary* MDPs.
>
> **2. Add a sufficient amount of structural constraints until your reward learning method becomes provably safe**
>
> In our work, we make almost no structural assumptions on our setup. This allows for our results to generalize over a wide range of MDPs, reward-learning, and policy-optimization techniques. Therefore, one strategy to develop provably safe reward learning methods is to assume additional constraints on these structures. The second paper you mentioned (*Kwa et al. Catastrophic Goodhart: regularizing RLHF with KL divergence does not mitigate heavy-tailed reward misspecification*) as well as all works in the “Upper bound results” paragraph of our related work section pursue this approach. In particular, Kwa et al. show in their Theorems 4 and 6 that doing RLHF or Conditioning can be provably safe under the following structural assumptions:
> - MDP: environmental transitions are deterministic and the policy return only depends on the final state reached.
> - The true reward and the error of the proxy reward are independently distributed according to a data distribution generated by an arbitrary reference policy and their distributions are light-tailed (more assumptions are required for Theorem 6).
> - The true reward is unbounded (i.e., the true reward function can attain arbitrarily large values)
>
> While the paper provides some empirical evidence (see Section 4) that the error of the proxy reward is indeed light-tailed in some settings, they also observe (Section 5.2) that some of their assumptions are rather strong and don’t hold in practice, such as the assumption that the true reward and the error of the proxy reward are independently distributed.
>
> ---
>
> In general, our results suggest that it is highly unlikely for a reward learning algorithm to be both fully general and provably safe, so we welcome works such as the ones described above which explore the trade-off between these two requirements.
>
> We would once again like to thank you for your review and we are happy to answer any further comments or questions that you might have in the discussion phase!

---

> > ### Comment · Reviewer_QgBs · 2025-04-01
> >
> > Thank you for the detailed rebuttal. I appreciate the comparison to prior works.
> >
> > I looked into the results you mentioned in the comparison to Laidlaw et al., and based on my reading I believe you may be misinterpreting their results. Lemma A.3 seems to show that their are some cases in which their regularization scheme cannot improve the true reward. I don't think it means that the regularization method actually allows reward hacking. I think that the main result (Theorem 5.1) shows that optimizing the objective with chi-squared divergence penalty can never allow reward hacking according to their definition.
> >
> > Furthermore, their Theorem A.5 appears to only apply to regularization based on action distributions, not occupancy measures. Theorem 5.1 seems to show that regularization using occupancy measure provably avoids reward hacking.
> >
> > It would be good to clarify if my interpretations are correct, and if so make sure to update your comparison to the related work.

---

> > > ### Author Response · Authors · 2025-04-08
> > >
> > > Thank you very much for your comment! We apologize for the misrepresentation, we missed the fact that the optimum of the RHS of the inequality in Theorem 5.1 is always at least zero, so when optimizing this expression one ends up with a policy that is at least as good as the reference policy. After carefully re-reading the paper we agree with your comments. We therefore plan to put the following comparison in our related work section:
> > >
> > > Laidlaw et al. (2025) consider a setting where the learned and true reward functions are positively correlated under a reference policy. They prove that maximizing the proxy reward with a chi-squared divergence penalty yields regret no worse than that of the reference policy. In experiments, they approximate this regularized objective and report favorable results.

---

### Official Review · Reviewer_Eqzu · 2025-03-22

**Overall Recommendation:** 2

**Summary:**

The paper considers the problem of reward learning where the environment is modeled as an MDP and an unknown reward is estimated with a learning algorithm whose solution is used as a proxy objective in a downstream policy optimization setting. This paper formalizes conditions under which learned reward functions experience "error-regret mismatch." In such settings the proxy is a poor substitute for achieving low regret on the true objective. The authors demonstrate, through rigorous theoretical analysis, that achieving low error in a learned reward model on the training data does not guarantee low regret in the resulting policy. They introduce the concepts of "safe" and "unsafe" data distributions, providing a framework for understanding when this mismatch occurs. The paper also considers regularized policy optimization and contextual bandit settings, highlighting the persistence of the problem in these cases.

**Claims And Evidence:**

See the section on theoretical claims and evidence.

**Essential References Not Discussed:**

There are no additional references that are essential to include.

**Experimental Designs Or Analyses:**

This is a purely theoretical paper with no empirical support.

**Methods And Evaluation Criteria:**

This paper was purely theoretical. See the section on theoretical claims and evidence for more information about how it gave support.

**Other Comments Or Suggestions:**

- The paper describes RLHF as a reward learning algorithm. RLHF is more accurately described as problem setting in which a class of algorithms can be brought to bear.
- The example starting on 81 did not clearly illustrate the concern for me.
- "A policy maximizing J is an optimal policy." This is missing a condition on policies; many policies may maximize J, though it is only reasonable to call those which evaluate higher than the rest optimal.
-  Currently the Appendix is 56 pages long. Much of the content felt extraneous to the points the paper needed to support. Culling unnecessary content would improve the paper's accessibility.

**Other Strengths And Weaknesses:**

*Strengths*

- Positioned for generality: Condition (1) defines an epsilon-accurate reward model. This provides the analysis with enough generality to remain agnostic to the details of any particular reward learning method.
- Clear problem definition: The paper articulates the error-regret mismatch problem in a clear and accessible manner.
- Rigorous theoretical treatment: The paper offers an extensive theoretical approach to analyze the error-regret mismatch setting.
- Topically relevant: The issue addressed is highly relevant to modern RLHF systems.

*Weaknesses*
- Mixed amounts of support: It is unclear whether all the theoretical claims are supported with correct proofs. See my comments about theoretical claims.
- Complex proofs: While the paper presents a rigorous analysis, the proofs felt quite dense, sometimes terse, and in some places redundantly developed. Revising these for easy consumption could improve accessibility and help build trust in the results.
- Potentially unrealistic assumptions: The analysis assumes the existence of an epsilon-accurate reward model over the full data distribution (Condition 1). While this allows the results to remain agnostic to training, the condition is quite strong. The paper points this out. Still, weakening this requirement could strengthen the results.

   Another example is Definition 2.1, which includes an particularly loose condition on regret. Regret is defined to be in the interval [0,1]. The definition considers a regret to be "low" if it falls with $[0,1)$. Thus "safe" distributions are those that don't lead to maximal regret of one. Similarly, the definition allows for the set to be empty by requiring $L=0$.

- Lacks empirical validation: Although the primary contributions are theoretical, there were several points which could benefit from empirical support. For instance, the defining conditions of a "safe" policy are quite technical, and it is not obvious how prevalent such data distributions are in practice. Providing an empirical demonstration of safe and unsafe data distributions would add clarity to this section and provide some validation to the definition.
- Motivation: Several choices could be better motivated. For example, the paper presents regularization as the de facto method to address objective mismatch. Though it is not explained how this is the case.

**Questions For Authors:**

- Regret is one choice among many to define performance. Why is regret the right quantity to analyze RLHF systems?

See other sections for more questions.

**Relation To Broader Scientific Literature:**

The paper generally did a good job positioning itself within the larger body of related work. The appendix contained an extended section of related work too.

Below are few other papers for understanding rewards apart from the approach used in Skalse et al. 2023.
1. [Settling the Reward Hypothesis](https://arxiv.org/pdf/2212.10420)
2. [Rethinking the discount factor in reinforcement learning: A decision theoretic approach.](https://arxiv.org/pdf/1902.02893)
3. [On the Expressivity of Markov Reward](https://arxiv.org/pdf/2111.00876)
4. [Utility Theory for Sequential Decision Making](https://arxiv.org/pdf/2206.13637)

**Theoretical Claims:**

*Claims*
1. As the error of a learned reward model on a data distribution goes to zero, the worst-case regret of optimizing a policy according to that reward model also goes to zero.
2. For any ϵ > 0 there exists a reward model that achieves an expected error of ϵ and has a high-regret optimal policy.
3. When an MDP has a large number of independent bad policies, every data distribution is "unsafe."
4. Derive a set of linear constraints that precisely characterize the safe data distributions for a given MDP.
5. Regularized versions of Propositions 3.1 and 3.3.
6. Provide an analysis of RLHF in the contextual bandit case

*Evidence*

I checked for correctness of the proofs and found mixed support. My main issue has to do with the lack of connection between the paper's claims and the proofs in the Appendix. Currently several statements in the paper are not directly proved in the Appendix. The Appendix includes results to different claims which the reader is assumed to take as logically equivalent to the paper. This obscures the analysis and makes it difficult to both understand the result and verify their correctness.

For these cases, I suggest the authors either
(i.) directly prove the claims in the paper
(ii.) show the claims from the paper are logically equivalent to those in the appendix, or
(iii.) use the claims from the appendix in the main paper.

1.a (Valid) Propositions 3.1. Proof of Corollary D.7.
1.b (Invalid) Proposition 3.2. This result was not proved in the referenced result (Proof of Theorem D.11).
2. (Inconclusive) Proposition 3.3. Proof of Proposition C.5.
   - What makes this claim different than the definition?
   - Where in the proof must the policy's regret be high?
   - What fact guarantees that $\hat{\pi}$ is optimal for $\hat{R}$?
   - Minor point: $\epsilon$ should be no greater than one.
3. (Inconclusive) Corollary 3. Proof of Corollary C.6.
   - The proof correctly proves one of its assumptions, but includes no other explanation of how this follows from Proposition 3.3.
4. (Valid) Theorem 3.5. Proof of Theorem C.16.
   - The proof relies on Lemma C.15 from the Appendix. This asserts distributions are safe when there are no solutions to a linear system with non-safe vertices. The proof constructs a system of equations and an equivalent convex program then argues its solution is consistent with Lemma C.15. On the surface, this seems reasonable. However, I found the proof difficult to verify as several steps were skipped and others felt redundant.
   - Equation 34 needs further support; it is not clear why the logical equivalence holds between 1511 and 1512.
5.a (Valid) Proposition 4.1. Proof of Theorem D.21.
5.b (Valid) Theorem 4.2. Proof of Theorem C.41.
   - This proof relies on several lemmas which, if true, lead to a valid conclusion here.
6. (Invalid) Theorem 6. Proofs of Propositions C.34 and C.35.
   - The referenced results do not prove the stated claim. If these two proofs together support Theorem 6, then a proof needs to establish the logical connection between these claims.

---

> ### Author Rebuttal · Authors · 2025-04-01
>
> Thank you for your detailed feedback! Due to the 5000-character limit, we have to focus on your main concerns in this rebuttal. Please share any additional issues you'd like us to address.
> # Addressing your remarks about our proofs
>  We appreciate your thorough technical review. We apologize for inconsistencies between the appendix and main text, which occurred as the appendix was written before we unified results in a shared framework for the main paper. We're confident our core results are sound. **We'll thoroughly revise the appendix to improve proof exposition, clarity, and brevity, beyond the specific points addressed in this rebuttal**.
>
> > 1.b (Invalid) Proposition 3.2. This result was not proved in the referenced result (Proof of Theorem D.11).
>
> Proposition 3.2 follows from the third regret bound in Theorem D.11. We are adding a Corollary D.12 in the revision to make this connection explicit.
>
> In particular: To prove $D \in safe(R, \epsilon, L)$, we need to show that whenever $\mathbb{E}\Bigg[\frac{|R(s,a) - \hat R(s,a)|}{range(R)}\Bigg] \le \epsilon$ (A) and $\hat \pi$ is optimal for $\hat R$, then $Reg^R(\hat\pi) < L$. Using Theorem D.11:
>
> $\begin{eqnarray}
> Reg^R(\hat \pi) &\le& \frac{\sqrt{2} \cdot d^D(R, \hat{R})}{(1 - \gamma) \cdot (\max J^R - \min J^R) \cdot \min D(s,a)}\\\\
> &\le& \frac{\sqrt{2} \cdot \epsilon \cdot range(R)}{(1 - \gamma) \cdot (\max J^R - \min J^R) \cdot \min D(s,a)}\\\\
> &<& L\end{eqnarray}$
>
> The second inequality uses the definition of $d^D$ (see lines 3248 and 3259) and assumption (A), while the third uses the upper bound on $\epsilon$ from Proposition 3.2.
>
> > 2.  (Inconclusive) Proposition 3.3. Proof of Proposition C.5.
> > Where in the proof must the policy's regret be high?
>
> In the proof, we show that the assumptions imply the existence of $\hat{R}$ that is $\epsilon$-close to $R$ and for which $\hat{\pi}$ is optimal. If one additionally considers the assumption that $Reg^R(\hat{\pi}) \geq L$, then what we show implies that $D \in unsafe(R, \epsilon, L)$, by the definition of this set of distributions. Thus, the regret being high is implicitly used by our proof. We make this explicit in the revised version.
> > What fact guarantees that $\hat{\pi}$ is optimal for $\hat{R}$?
>
> The state-action pairs that $\hat{\pi}$ visits all lie in $supp D^{\hat{\pi}}$ , where $\hat{R}$ has maximal reward $max R$. This implies that $\hat{\pi}$ is optimal for $\hat{R}$.
> > 3. (Inconclusive) Corollary 3. Proof of Corollary C.6.
> The proof correctly proves one of its assumptions, but includes no other explanation of how this follows from Proposition 3.3.
>
> In the proof, we show $D(supp D^{\pi}) < \epsilon$. Implicitly, we also use that $Reg^{R}(\pi) \geq L$, which is due to $\pi \in \Pi_L$. These are the two assumptions from Proposition 3.3, which imply $D \in unsafe(R, \epsilon, L)$. Since $D$ was arbitrary, this implies $\Delta(S \times A) = unsafe(R, \epsilon, L)$. We make these last arguments explicit in an updated version.
> > (Invalid) Theorem 6. Proofs of Propositions C.34 and C.35. The referenced results do not prove the stated claim.
>
> Proposition C.34 demonstrates that reference policies $\pi_{ref}$ satisfying conditions a) and b) create unsafe data distributions $D^{ref}(s,a) := \mu_0(s) \cdot \pi_{ref}(a|s)$ (see Definition C.30 to verify this). In our revision, we'll streamline the proposition by referencing Def. C.30.
>
> Proposition C.35 then provides simpler conditions that imply those in Proposition C.34. Theorem 6.1 combines these results, showing that reference policies satisfying Proposition C.35's conditions create unsafe data distributions. In the revised version we will replace the statement of C.35 with the one of Thm. 6.1 to make this connection explicit.
> Lastly, the $2\cdot$ inside the $unsafe()$ statement is a typo that we will remove.
> # Regarding the assumptions
> > The analysis assumes the existence of an epsilon-accurate reward model over the full data distribution
>
> We focus our paper largely on negative results, which get stronger if they hold even under the assumption of an epsilon-accurate reward model over the full data distribution.
>
> > Regret is defined to be in the interval [0,1]. The definition considers a regret to be "low" if it falls with [0,1)
>
> Both, $\epsilon$, as well as $L$, are free variables. Hence, depending on the application, one can decide how to set these values, i.e., what constitutes a low regret. We plan to update our explanations to make this clearer.
>
> > the defining conditions of a "safe" policy are quite technical, and it is not obvious how prevalent such data distributions are in practice
>
> To clarify our definitions and negative results, we provide a detailed chatbot example in Appendix B.4. Furthermore, Figure 4 provides a simple example of data distributions, highlighting which are safe. We will better integrate these explanations into the main paper.
>
> ---
>
> Please let us know any remaining concerns we should address during the discussion phase!

---

### Official Review · Reviewer_CcB3 · 2025-04-04

**Overall Recommendation:** 2

**Summary:**

The paper states an important issue in RLHF, that is the error-regret mismatch, which is fundamental due to the distribution shift of the induced data by the fine-tuned policy. The core contribution of the paper is to theoretically analyze the possibility of error-regret mismatch, assuming accurate estimation of the reward function. A distribution that, with any accurate reward estimation, will create a low-regret policy is called safe, and it is unsafe otherwise.
The structure of the study is a set of theoretical claims as follows:
1. Full support distributions are safe for sufficiently accurate reward estimation (for both reg/unreg objectives)
2. The existence of a high regret policy with a small support intersection with a distribution makes it unsafe (for both reg/unreg objectives)
3. An equivalent linear condition is proposed for the safety of a distribution in case of the unregularized objective for policy optimization.
One interesting point of the analysis is the generalizability of the regularization term, which is not limited to KL or chi divergence.

**Claims And Evidence:**

The core claims are the same as stated in the summary, which are backed by theoretical proofs.

**Essential References Not Discussed:**

No essential reference is missed to the best of my knowledge.

**Experimental Designs Or Analyses:**

There are no experiments to practically validate the claims. There is an example of the computation of the matrix M on a simple MDP in App. C.3.4. But, I don't find any examples of the validation and application of the analysis on real-world scenarios on popular RLHF methods.

**Methods And Evaluation Criteria:**

Regret as the main measurement of the goodness of a policy is the standard criterion in the literature.
A point about the measurement of the accuracy of the reward estimation is that, as also stated by the paper, it is mostly an *upper bound* of the bounds provided on the reward estimation errors. Hence, it is mostly not possible to assert such a bound for the estimated rewards, which affects the application of the analysis for different algorithms and methods.

**Other Comments Or Suggestions:**

I don't have any other comments.

**Other Strengths And Weaknesses:**

The reward range is implicitly assumed to be bounded, which is not true in very popular preference-based policy optimization models such as Bradley-Terry.

**Questions For Authors:**

The final analysis on the RLHF setting is ambiguous for me. I don't get the main point of Theorem 6.1. Can the authors explain more about the significance and goal of Theorem 6.1 and provide an example of its application on a real, SOTA RLHF method?

**Relation To Broader Scientific Literature:**

Currently, the most popular application of the study is indeed RLHF. The paper compares itself with other analytical studies and reviews reward learning in offline RL literature. The main missing part is its relation w.r.t. the new RLHF methods such as RPO, SimPO, and IPO. It is true that the analytics is based on a specific approach in RLHF, however, the validity of the analysis on other RLHF methods can give very useful insights and directions for future work. Also, the general regularization term can be applied to some new studies in RLHF, as in the following paper, which can be validated empirically.
Huang, Audrey, et al. "Correcting the mythos of kl-regularization: Direct alignment without overoptimization via chi-squared preference optimization." arXiv preprint arXiv:2407.13399 (2024).

**Theoretical Claims:**

The paper is totally analytical and theoretical, hence, all the claims mentioned are justified by theoretical proof. I didn't get into the details of the proofs. But the statements of the intermediate lemmas and propositions are logical and indicate a correct deductive flow. Moreover, the theoretical claims are fairly intuitive.

---

### Decision · Program_Chairs · 2025-05-01

**Decision:**

Accept (poster)

**Comment:**

This paper provides a clear framework and theoretical contributions on a highly relevant topic, which has needed more such theoretical treatments to better guide the predominantly experimental and heuristic-guided approaches. Indeed, the results of this work apply significantly more broadly than the current trends in the field, even though they are arguably especially important at the present moment.

Arguably the strongest argument against the paper comes down to a potential mismatch between more informal or intuitively expressed claims and the precise theoretical result. Looking through these concerns, I think the author(s) have made a good faith effort to improve this alignment and do not think that this significantly detracts from the work.

Aside from these concerns, and a separate reviewer’s request for comparisons with relatively unrelated methods, the other two reviewers were significantly positive about this work and I share their general enthusiasm.